# Age-associated changes in lineage composition of the enteric nervous system regulate gut health and disease

Subhash Kulkarni[1,2]*, Monalee Saha[3†], Jared Slosberg[4†], Alpana Singh[3], Sushma Nagaraj[3], Laren Becker[5], Chengxiu Zhang[3], Alicia Bukowski[3], Zhuolun Wang[3], Guosheng Liu[3], Jenna M Leser[3], Mithra Kumar[3], Shriya Bakhshi[3], Matthew J Anderson[6], Mark Lewandoski[6], Elizabeth Vincent[4], Loyal A Goff[7,8], Pankaj Jay Pasricha[9]

[1]Division of Gastroenterology, Dept of Medicine, Beth Israel Deaconess Medical Center, Boston, United States; [2]Division of Medical Sciences, Harvard Medical School, Boston, United States; [3]Center for Neurogastroenterology, Department of Medicine, Johns Hopkins University – School of Medicine, Baltimore, United States; [4]Department of Genetic Medicine, Johns Hopkins University – School of Medicine, Baltimore, United States; [5]Division of Gastroenterology, Stanford University – School of Medicine, Stanford, United States; [6]Center for Cancer Research, National Cancer Institute, Frederick, United States; [7]Department of Neuroscience, Johns Hopkins University – School of Medicine, Baltimore, United States; [8]Kavli Neurodiscovery Institute, Johns Hopkins University – School of Medicine, Baltimore, United States; [9]Department of Medicine, Mayo Clinic, Scottsdale, United States

*For correspondence:
skulkar1@bidmc.harvard.edu

†These authors contributed equally to this work

Competing interest: The authors declare that no competing interests exist.

**Abstract** The enteric nervous system (ENS), a collection of neural cells contained in the wall of the gut, is of fundamental importance to gastrointestinal and systemic health. According to the prevailing paradigm, the ENS arises from progenitor cells migrating from the neural crest and remains largely unchanged thereafter. Here, we show that the lineage composition of maturing ENS changes with time, with a decline in the canonical lineage of neural-crest derived neurons and their replacement by a newly identified lineage of mesoderm-derived neurons. Single cell transcriptomics and immunochemical approaches establish a distinct expression profile of mesoderm-derived neurons. The dynamic balance between the proportions of neurons from these two different lineages in the post-natal gut is dependent on the availability of their respective trophic signals, GDNF-RET and HGF-MET. With increasing age, the mesoderm-derived neurons become the dominant form of neurons in the ENS, a change associated with significant functional effects on intestinal motility which can be reversed by GDNF supplementation. Transcriptomic analyses of human gut tissues show reduced GDNF-RET signaling in patients with intestinal dysmotility which is associated with reduction in neural crest-derived neuronal markers and concomitant increase in transcriptional patterns specific to mesoderm-derived neurons. Normal intestinal function in the adult gastrointestinal tract therefore appears to require an optimal balance between these two distinct lineages within the ENS.

## eLife assessment

This paper identifies a subset of neurons within adult mouse myenteric ganglia that are not labeled via canonical neural-crest labeling, and argues, based on extensive lineage tracing, imaging and genomic data that these neurons are derived from mesoderm. There is **convincing** evidence for the

existence of an unusual cell type in the gut that expresses neuronal markers, but which is derived from cells expressing markers of the mesoderm rather than the expected neural crest, which is an intriguing and **important** observation. While the data do not definitively establish the molecular taxonomy of this lineage, there is sufficient evidence to support the provocative and paradigm-shifting hypothesis of the non-ectodermal origin for enteric neurons to warrant further deeper investigation.

## Introduction

The enteric nervous system (ENS) is a large collection of neurons, glial, and precursor cells that resides within the gastrointestinal wall and regulates gut motility and secretion along with modulating epithelial and immune cell function (*Kulkarni et al., 2018*; *Jarret et al., 2020*). During fetal development, the mammalian ENS is populated by neurons and glia derived from neural crest (NC)-derived precursors (*Uesaka et al., 2016*; *Obermayr et al., 2013*; *Hao and Young, 2009*; *Anderson et al., 2006*; *Young and Newgreen, 2001*; *Young et al., 2000*; *Bergner et al., 2014*). These precursors follow diverse migratory routes to colonize and innervate various parts of the gut before birth (*Uesaka et al., 2015*; *Espinosa-Medina et al., 2017*; *Burns, 2005*). It is not clear, however, that this lineage persists in its entirety in the adult gut, as indicated by the observed lack of expression of fluorescent reporter protein in a subpopulation of adult enteric neurons in NC-lineage-traced mice (*Laranjeira et al., 2011*; *Brokhman et al., 2019*). Alternative sources of enteric neurons that have been proposed in the literature include the ventral neural tube (VENT) (*Sohal et al., 2002*), or the *Pdx1*-expressing pancreatic endoderm (*Brokhman et al., 2019*), but the interpretation of these studies has been limited by the lack of robust lineage markers for non-NC derived neurons (*Habeck, 2003*). In addition, while prior studies have documented cellular changes to the ageing ENS (*Saffrey, 2013*), the developmental mechanisms behind these changes are unknown. Thus, confirmation of a second, distinct lineage of enteric neurons in adults is important for our understanding of the healthy post-natal development and aging of the ENS, as well as for the pathogenesis of acquired disorders of the ENS.

In this study, we found that while the early post-natal ENS is derived from the canonical NC-lineage, this pattern changes rapidly as the ENS matures, due to the arrival and continual expansion of a novel population of *M*esoderm-derived *E*nteric *N*eurons (MENs) which represent an equal proportion of the ENS in young adulthood and with increasing age, eventually outnumber the NC-derived Enteric Neurons (NENs). We also found that, while the NEN population is regulated by glialderived neurotrophic factor (GDNF) signaling through its receptor RET, the MEN population is regulated by hepatocyte growth factor (HGF) signaling. Increasing HGF levels during maturation or by pharmacological dosing increase proportions of MENs. Similarly, decrease in GDNF with age decrease NENs; and increasing GDNF levels by pharmacological dosing increase NENs proportions in the adult ENS to impact intestinal motility.

These results indicate for the first time that the mesoderm is an important source of neurons in the second largest nervous system of the body. The increasing proportion of neurons of mesodermal lineage is a natural consequence of maturation and aging; further, this lineage can be expected to have vulnerabilities to disease that are distinct from those affecting the NEN population. These findings therefore provide a new paradigm for understanding the structure and function of the adult and aging ENS in health, age-related gut dysfunction and other acquired disorders of gastrointestinal motility.

## Results

### Only half of all mid-age adult enteric neurons are derived from the NC

We analyzed small intestinal longitudinal muscle-myenteric plexus (LM-MP) from adult (post-natal day 60; P60) Wnt1-Cre:*Rosa26*$^{lsl-tdTomato}$ mice, in which tdTomato is expressed by all derivatives of *Wnt1*$^+$ NC-cells (*Becker et al., 2012*). In these tissues, while GFAP, a glial marker, was always co-expressed with tdTomato (*Figure 1a*), tdTomato-expression was absent in many myenteric neurons (*Figure 1b and c*). By careful enumeration, almost half of all myenteric neurons expressing the pan-neuronal marker Hu were found to not express the reporter (percent tdTomato$^-$ neurons: 44.27±2.404 SEM; enumerated from 2216 neurons from six mice, *Figure 1d*). In these lineage-traced mice, myenteric

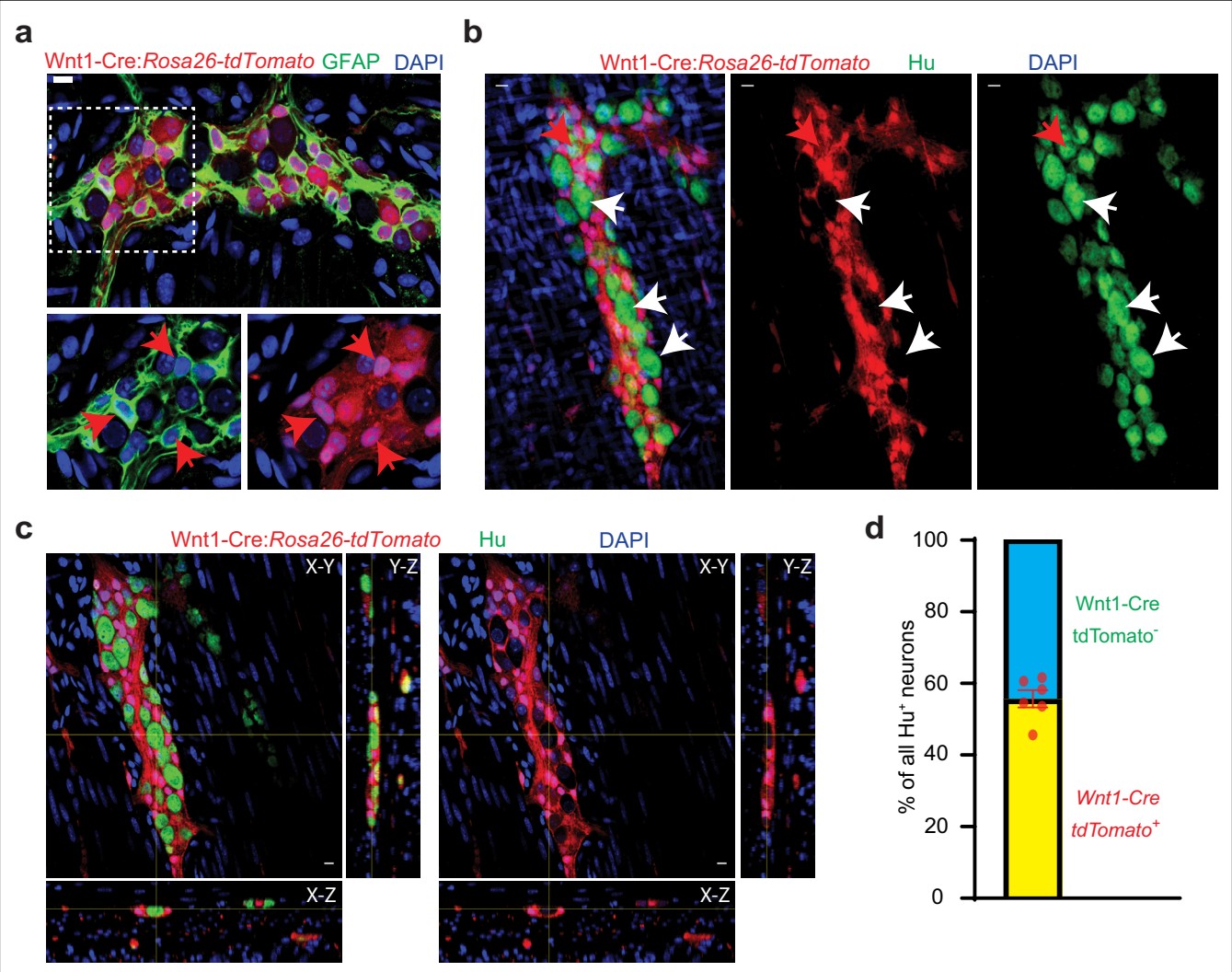

**Figure 1.** Half of all adult small intestinal myenteric neurons are derived from a non-neural crest lineage. (**a**) Enteric glia, labeled with GFAP (green) in the myenteric plexus from an adult Wnt1-Cre:*Rosa26*[lsl-tdTomato] mouse. Inset is magnified in color segregated panels, showing GFAP[+] glia co-expressing tdTomato (red, red arrows). Nuclei are labeled with DAPI (blue). Scale bar = 10 μm. (**b**) Two-dimensional representation of a three-dimensional stack of images shows enteric neurons, labeled with Hu (green), in the myenteric plexus from an adult Wnt1-Cre:*Rosa26*[lsl-tdTomato] mouse. Enteric neurons either express tdTomato (red arrow) or not (white arrows). Nuclei are labeled with DAPI (blue). Scale bar = 10 μm. (**c**) Orthogonal views of the image (**b**) shows the lack of tdTomato expression in a particular Hu-immunolabeled enteric neuron. (**d**) Quantification of tdTomato[+] and tdTomato[-] neurons in the myenteric ganglia of 6 P60 Wnt1-Cre:*Rosa26*[lsl-tdTomato] mice. Data represents Mean ± SEM.

The online version of this article includes the following source data and figure supplement(s) for figure 1:

**Figure supplement 1.** Presence and absence of tdTomato reporter expression in the myenteric ganglia of adult neural crest lineage-traced mice.

**Figure supplement 1—source data 1.** Antibodies against HuC and HuD also recognize HuB protein.

ganglia were found to contain tdTomato[high] and tdTomato[low] neurons and due care was taken to image subsets (*Figure 1—figure supplement 1a*). Both tdTomato[high] and tdTomato[low] neurons were classified as tdTomato[+] and only neurons that did not show any tdTomato-expression were classified as tdTomato[-] neurons. Immunofluorescence staining of small intestinal LM-MP tissue from adult Wnt1-Cre:*Rosa26*[lsl-tdTomato] mice using antibodies against tdTomato protein showed a consistent lack of reporter expression in a population of myenteric neurons (*Figure 1—figure supplement 1b*). This lack of reporter expression is not a function of tissue peeling or fixation, as the tdTomato[-] cells within the myenteric ganglia were also observed in freshly harvested unpeeled gut tissue, when imaged using a live tissue fluorescence microscope (*Figure 1—figure supplement 1c*). In addition, we noticed that myenteric ganglia of the Wnt1-Cre:*Rosa26*[lsl-tdTomato] transgenic mouse show similar tdTomato

aggregation in cells of the myenteric plexus from both freshly harvested and immediately fixed tissue (*Figure 1—figure supplement 1d, e*). Since we previously showed ongoing neuronal loss in healthy ENS (*Kulkarni et al., 2017*), we reasoned that cells that show hyper-aggregation of tdTomato are in advanced stages of cell death. Cells with hyper-aggregation of tdTomato showed a lack of staining for the nuclear marker DAPI (*Figure 1—figure supplement 1e*), suggesting that these are indeed cells in the advanced stages of death, and given that they are not labeled with antibodies against Hu, their presence does not alter our estimation of tdTomato-expressing neurons (*Figure 1—figure supplement 1f*). Thus, the absence of reporter expression in myenteric ganglia of freshly harvested tissue and the presence of tdTomato aggregation in both freshly harvested and fixed tissues shows that our observations are not caused due to any technical issues in tissue isolation and preservation.

We next sought additional confirmation that the lack of tdTomato expression in this often-used lineage fate mapping mouse model was not due to incorrect activity at the *Rosa26* locus where the floxed reporter transgene is located, due to aberrant Cre activity under the Wnt1 promoter, or due to issues with the antibodies used to detect the pan-neuronal protein Hu. This was provided by an analogous Wnt1-Cre:*Hprt*^lsl-tdTomato^ lineage-traced mouse line, in which tdTomato was expressed from the *Hprt* locus in a Wnt1-Cre-dependent manner (*Hprt* locus is X-inactivated in females, hence only adult male mice were used *Antal et al., 2014*); and by the *Pax3*^Cre^:*Rosa26*^lsl-tdTomato^ lineage-traced mouse line, where *Rosa26*^lsl-tdTomato^ driven by Cre expressed from the *Pax3* locus labels the derivatives of the neural tube and pre-migratory NC (*Figure 1—figure supplement 1g, h*; *Freyer et al., 2011*). Similar lack of reporter expression was previously observed by the Pachnis Lab in adult myenteric neurons from Sox10-Cre NC-lineage-traced mice, and more recently by the Heuckeroth Lab in significant numbers of myenteric neurons from Wnt1-Cre:*Rosa26*^H2B-mCherry^ mice, which expresses nuclear localized reporter mCherry in NC-lineage cells, further confirming our observations of the presence of non-NC-derived neurons in the adult ENS (*Laranjeira et al., 2011*; *Wright et al., 2021*). In addition to the ANNA-1 antisera, which is known to detect all neuronally significant Hu proteins (HuB, HuC, and HuD; *King et al., 1999*), we found that another often used antibody thought to be specific to HuC and HuD also detects HuB protein (*Figure 1—figure supplement 1i*) – showing that these antibodies do not detect the expression of a specific neuronal Hu antigen.

## Lineage-tracing confirms a mesodermal derivation for half of all adult myenteric neurons

Alternative sources of enteric neurons proposed previously include the ventral neural tube (*Sohal et al., 2002*), and the pancreatic endoderm (*Brokhman et al., 2019*), but the interpretation of these studies was limited by the lack of robust lineage markers (*Habeck, 2003*). *Brokhman et al., 2019* found evidence of labeled neurons in inducible Pdx1-Cre, *Foxa2*^CreESR1^, and *Sox17*^Cre^ lineage-traced mouse lines and inferred their derivation from pancreatic endoderm. However, in a Pdx1-Cre lineage-traced mouse line, many neuroectoderm-derived neurons of the central nervous system have also been previously shown to be derived from *Pdx1*-expressing cells (*Honig et al., 2010*), suggesting that *Pdx1* is not an exclusive endodermal marker. *Foxa2* is expressed by mesoderm-derived cells in the heart (*Bardot et al., 2017*) and *Sox17* also labels mesoderm-derived cells, especially cells of the intestinal vasculature (*Burtscher et al., 2012*; *Wilm et al., 2005*; *Kawaguchi et al., 2007*; *Herrick and Mutsaers, 2004*). We therefore hypothesized that the embryonic mesoderm may be the true developmental source of the non-NC enteric neurons.

Mesoderm-derived cells during embryogenesis express *Tek* (*Bardot et al., 2017*) and analysis of LM-MP tissues from adult male Tek-Cre:*Hprt*^lsl-tdTomato^ lineage-traced mice revealed the presence of a population of tdTomato^+^ neurons (*Figure 2a*). Since *Tek* is not expressed in the adult ENS (*Figure 2—figure supplement 1a*), the presence of *Tek*-derived adult myenteric neurons suggested their mesodermal origin. We then used *mesoderm posterior 1* or *Mesp1* as a definitive developmental marker for the embryonic mesoderm (*Chan et al., 2013*; *Devine et al., 2014*; *Lescroart et al., 2014*; *Klotz et al., 2015*) to confirm the mesodermal derivation of the non-NC derived adult enteric neurons. Reporter expression in this mouse line has been previously studied in cardiac development, which we confirmed by observing its expression in the adult heart tissue (*Figure 2b*). Expression of tdTomato in vascular cells of both Tek-Cre and *Mesp1*^Cre^ mice was more pronounced than in myenteric neurons (*Figure 2a, c and d*) and in cardiac myocytes of *Mesp1*^Cre^ mice (*Figure 2b*, *Figure 2—figure supplement 1b*), which reflects the variable expression of the reporter gene in both the NC-lineage

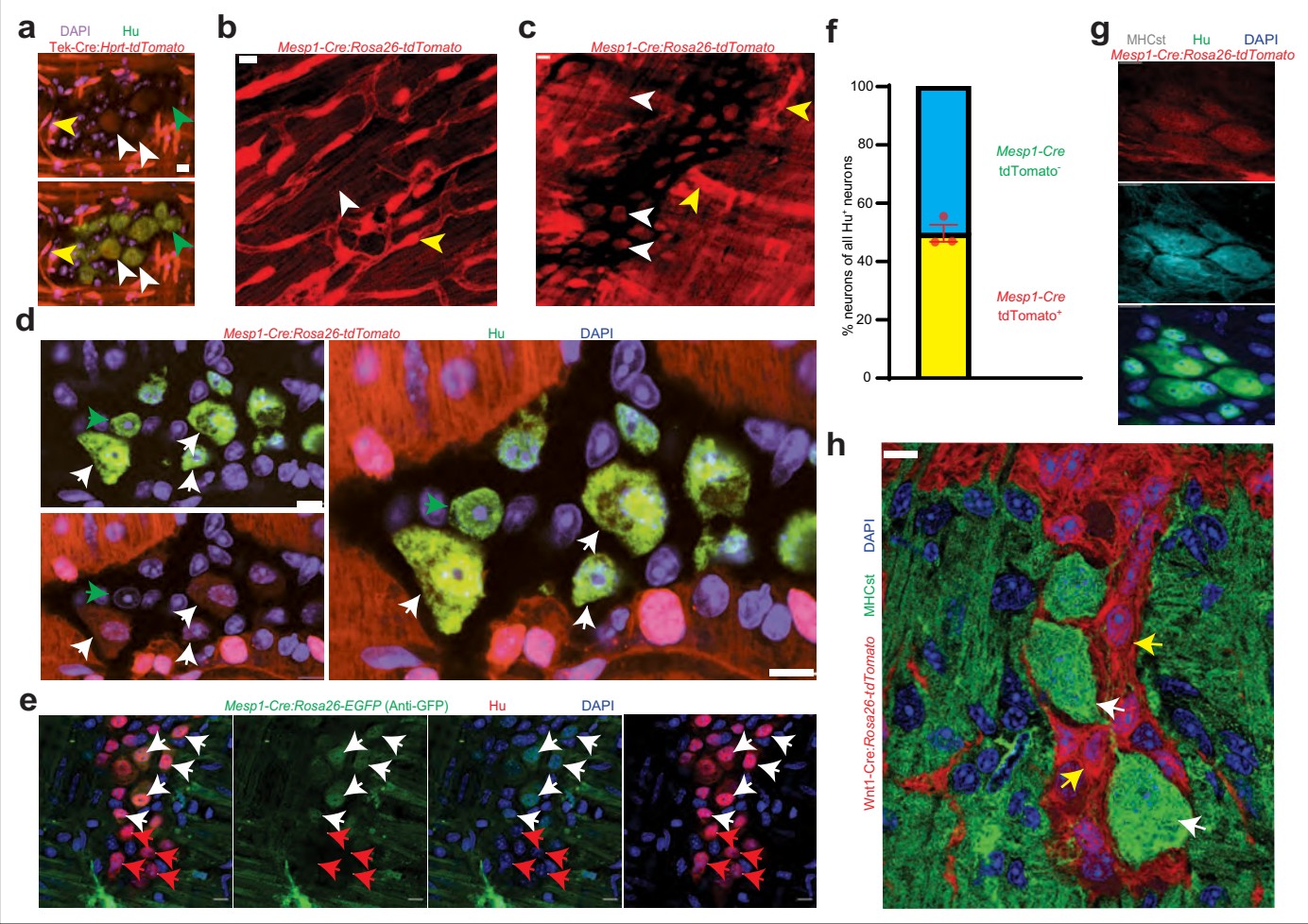

**Figure 2.** Mesoderm-lineage-tracing and marker expression provide evidence of the mesodermal-derivation of half of all adult small intestinal myenteric neurons. (**a**) Small intestinal LM-MP from adult male Tek-Cre:*Hprt*$^{lsl-tdTomato}$ shows the presence of *Tek*-derived and hence mesoderm-derived tdTomato$^+$ (red) Hu$^+$ (green) neurons (white arrows) and non-mesoderm-derived tdTomato$^-$ neurons (green arrows). Vascular cells depict higher fluorescence intensity (yellow arrow) than cells in the ganglia. Nuclei are labeled with DAPI. Scale bar = 10 μm. Cardiac and small intestinal tissues from (**b**) heart and (**c**) small intestinal LM-MP layer of adult male *Mesp1*$^{Cre}$:*Rosa26*$^{lsl-tdTomato}$ mice shows the presence of variable tdTomato expression in cells of these tissues. While cardiomyocytes in heart and cells within the myenteric ganglia (white arrows) exhibit lower tdTomato fluorescence, similar to the tissues from Tek-cre:*Hprt*$^{lsl-tdTomato}$ mice in (**a**), the vascular cells exhibit higher fluorescence intensity (yellow arrows). Scale bar = 10 μm. (**d**) Small intestinal LM-MP from adult male *Mesp1*$^{Cre}$:*Rosa26*$^{lsl-tdTomato}$ mouse shows the presence of mesoderm-derived tdTomato$^+$ (red) Hu$^+$ (green) neurons (white arrows) and non-mesoderm-derived tdTomato$^-$ neurons (green arrows). Nuclei are labeled with DAPI. Scale bar = 10 μm. (**e**) Small intestinal LM-MP from adult male *Mesp1*$^{Cre}$:*Rosa26*$^{lsl-EGFP}$ mouse, when immunostained with antibodies against GFP and against Hu, shows the presence of mesoderm-derived EGFP$^+$ (green) Hu$^+$ (red) neurons (white arrows) and non-mesoderm-derived EGFP$^-$ neurons (red arrows). Nuclei are labeled with DAPI. Scale bar = 10 μm. (**f**) Quantification of tdTomato-expressing and non-expressing neurons in the myenteric ganglia of 3 P60 *Mesp1*$^{Cre}$:*Rosa26*$^{lsl-tdTomato}$ mice. Data represents Mean ± SEM. MHCst immunolabeling exclusively marks all the (**g**) mesoderm-derived adult Hu$^+$ (green) neurons in the small intestinal LM-MP of *Mesp1*$^{Cre}$:*Rosa26*$^{lsl-tdTomato}$ mice; and (**h**) all the non-NC-derived neurons in the small intestinal LM-MP of Wnt1-Cre: *Rosa26*$^{lsl-tdTomato}$ mice (white arrows). MHCst does not label tdTomato$^+$ NC-derived cells (yellow arrows). Nuclei are stained with DAPI (blue). Scale bar = 10 μm.

The online version of this article includes the following figure supplement(s) for figure 2:

**Figure supplement 1.** *Tek*-expressing and *Mesp1*-expressing mesodermal lineage contributes to adult enteric neurons.

**Figure supplement 2.** Validation of MENs-specific markers by immunohistochemistry and confocal microscopy.

and mesoderm-lineage-specific transgenic mouse lines (***Figure 2b and c***; ***Figure 1—figure supplement 1a***, ***Figure 2—figure supplement 1c***). Using an analogous *Mesp1*$^{cre}$:*Rosa26*$^{lsl-EGFP}$ lineage-traced mouse line, in which EGFP was expressed from the *Rosa26* locus in a *Mesp1*$^{Cre}$-dependent manner, we next confirmed the derivation of a population of myenteric neurons from *Mesp1*-expressing mesoderm (***Figure 2e***, ***Figure 2—figure supplement 1d, e***). Similar to our observations with the tdTomato

reporter (*Figure 2a–d*), we observed lower expression of the reporter EGFP in myenteric neurons as compared to vascular cells (*Figure 2e*; *Figure 2—figure supplement 1d, e*). Variable expression of CAG and other pan-cellular promoters in various cell-types and during processes of maturation have previously been reported (*Baup et al., 2009*; *Alexopoulou et al., 2008*; *Hu et al., 2021*; *Akagi et al., 1997*). In addition, an earlier study by *Agah et al., 1997* showed differing degrees of CAG-driven LacZ reporter activation in a cardiac-specific transgenic Cre mouse line, which were unrelated to copy number, suggesting insertional and positional effects or, potentially, differential methylation (). These reports are consistent with our observation and could potentially help explain the observed variable expression of CAG and CMV promoter-driven reporter genes in our study.

Analysis of small intestinal LM-MP from P60 *Mesp1$^{Cre}$:Rosa26$^{lsl-tdTomato}$* lineage-traced mice showed that tdTomato-expression was observed in about half of all myenteric neurons (*Figure 2f*; percent tdTomato$^+$ neurons: 50.28±2.89 SEM; enumerated from 484 neurons from 3 mice). Because of the variable expression of the reporter gene in the transgenic mice described above, we also labeled tdTomato$^+$ neurons in the myenteric plexus of *Mesp1$^{Cre}$:Rosa26$^{lsl-tdTomato}$* mice with the S46 antibody, which was raised against the slow tonic myosin heavy chain protein MHCst derived from avian embryonic upper leg muscle, and is expressed exclusively by mesoderm-derived cell populations (*Figure 2g*; *Sokoloff et al., 2007*; *Stockdale and Miller, 1987*; *Miller et al., 1985*; *Vaughan et al., 2017*). MHCst immunostaining was exclusively observed in all Tek-Cre:*Rosa26$^{lsl-tdTomato+}$* MENs (*Figure 2—figure supplement 1f*). None of the 107 MHCst$^+$ intra-ganglionic cells observed (35.67±6.17 SEM MHCst$^+$ cells per mouse) across three Wnt1-Cre:*Rosa26$^{lsl-tdTomato}$* mice expressed tdTomato, suggesting that MHCst immunostaining was exclusive to non-NC lineage neurons (*Figure 2h*, *Figure 2—figure supplement 2a* shows the same image where the MHCst immunostained ganglia can be observed with and without the tdTomato channel for better visualization). MHCst antibody was also found to label extra-ganglionic smooth muscle cells in the LM-MP tissue (*Figure 2—figure supplement 2a*). Thus, both lineage tracing and protein biomarker expression provides strong support for their mesodermal origin (*Figure 2g*, *Figure 2—figure supplement 2a*). Along with our observations on Wnt1-Cre:*Rosa26$^{lsl-tdTomato}$* mice (*Figure 1c*), our results indicate that these two distinct lineages together account for all the adult small intestinal myenteric neurons.

The proteins RET, a receptor tyrosine kinase that transduces GDNF signaling in NC-derived enteric neuronal precursors, and MET, a receptor for hepatocyte growth factor (HGF), are expressed by different subsets of adult myenteric neurons (*Avetisyan et al., 2015*). MET is classically expressed by mesoderm-derived cells (*Rappolee et al., 1996*), and by using immunostaining of small intestinal LM-MP tissues from Wnt1-Cre:*Rosa26$^{lsl-tdTomato}$* and *Mesp1$^{Cre}$:Rosa26$^{lsl-tdTomato}$* mice, we found that the expression of MET was restricted to a sub-population of adult MENs (*Figure 3a and b*). By contrast, RET expression was confined to NENs (*Figure 3c*).

We then studied whether MENs and NENs differed phenotypically. In the current paradigm regarding the neurochemical basis for the functional classification of adult enteric neurons, inhibitory enteric motor neurons express the nitric oxide-producing enzyme nitric oxide synthase 1 (NOS1), excitatory motor neurons express the acetylcholine-producing enzyme choline acetyl transferase (ChAT), and enteric sensory neurons, called intrinsic primary afferent neurons (IPANs) express the neuropeptide calcitonin gene related peptide (CGRP) (*Bergner et al., 2014*; *Avetisyan et al., 2015*; *Hao et al., 2013*). Immunostaining using antibodies against NOS1, ChAT, and CGRP in the LM-MP from P60 Wnt1-Cre:*Rosa26$^{lsl-tdTomato}$* mice showed that both tdTomato$^+$ NENs and tdTomato$^-$ MENs express these neuronal markers (*Figure 3d, f and h*). Quantification of proportions in these neuronal lineages shows that the majority of NOS1$^+$ inhibitory neurons and ChAT$^+$ excitatory neurons are NENs (*Figure 3e*; of the 616 NOS1$^+$ neurons observed across three mice, NOS1$^+$ tdTomato$^-$: 27.06%±4.46 SEM; p<0.0001; Fisher's exact test); (*Figure 3g*; of the 912 ChAT +neurons observed across three mice, ChAT$^+$ tdTomato$^-$: 26.34%±2.54 SEM; p<0.0001; Fisher's exact test). By contrast, the majority (~75%) of CGRP$^+$ neurons were found to be MENs (*Figure 3i*; of the 146 CGRP$^+$ neurons observed across three mice, CGRP$^+$ tdTomato$^-$: 75.76%±0.63 SEM; p<0.0001; Fisher's exact test). These results are in keeping with a previous report by *Avetisyan et al., 2015* that shows low expression of NOS1 (0% of MET$^+$ neurons were NOS$^+$) and ChAT (<8% of MET$^+$ neurons were ChAT$^+$) and abundant expression of CGRP by MET$^+$ neurons in the adult ENS, previously not known to be derived from a different lineage. In addition, MENs also express Cadherin-3 (CDH3, *Figure 2—figure supplement 2b*), which is known to also mark a sub-population of mechanosensory spinal neurons (*Abraira et al., 2017*). Apart from

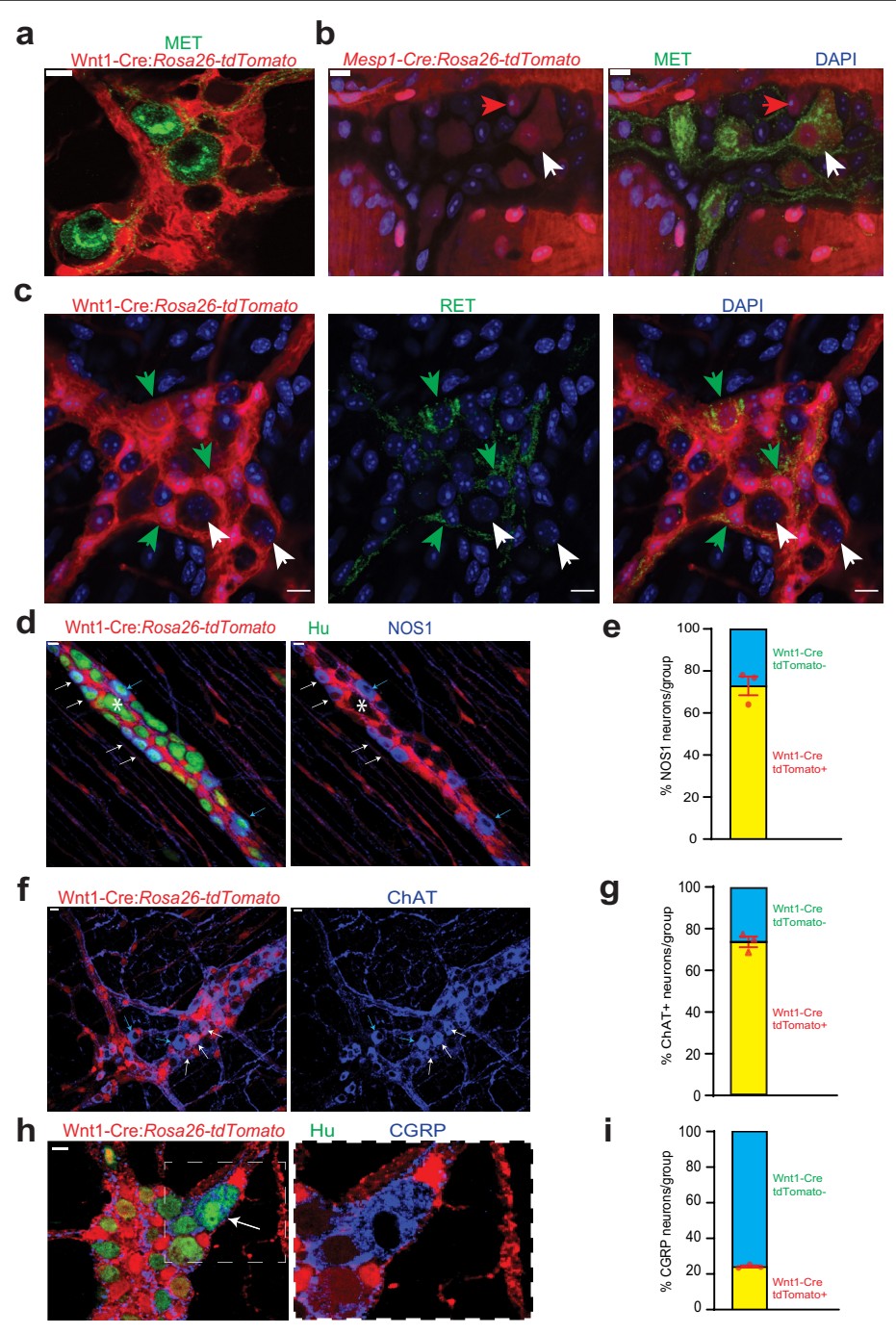

**Figure 3.** Cellular and molecular phenotyping of MENs and NENs. MET immunostaining (green) labels (**a**) Wnt1-Cre:*Rosa26* [lsl-tdTomato-] MENs and (**b**) a population of *Mesp1*[Cre]: *Rosa26* [lsl-tdTomato+] MENs (white arrow) while not labeling all MENs (red arrow). (**c**) RET immunostaining (green) only labels Wnt1-Cre:*Rosa26*[lsl-tdTomato+] NENs (green arrow) and not tdTomato- MENs (white arrows). (**d**) NOS1 is expressed by both tdTomato+ NENs (red, white arrows) and tdTomato- MENs (blue arrows) in an adult Wnt1-Cre:*Rosa26* [lsl-tdTomato] mouse, but most MENs do not express NOS1 (marked by *). (**e**) NEN lineage contains significantly higher proportions of NOS1+ neurons compared to MEN lineage in Wnt1-Cre:*Rosa26* [lsl-tdTomato] mice. Data represent mean ± S.E.M. p<0.0001; Fisher's exact test. (**f**) ChAT is expressed by both tdTomato+ NENs (red, white arrows) and tdTomato- MENs (blue arrows) in an adult Wnt1-Cre:*Rosa26* [lsl-tdTomato] mouse. (**g**) NEN lineage contains significantly higher proportions of ChAT+ neurons compared to MEN lineage in Wnt1-Cre:*Rosa26* [lsl-tdTomato] mice. Data represent mean ± S.E.M. p<0.0001; Fisher's exact test. (**h**) Both tdTomato+ (red) and tdTomato- neurons in the myenteric plexus of an adult Wnt1-Cre:*Rosa26* [lsl-tdTomato] mouse

*Figure 3 continued on next page*

*Figure 3 continued*

(Hu, green) express CGRP (blue) Inset showing a tdTomato⁻ CGRP⁺ neuron (white arrow) is magnified on the right. (**i**) MEN lineage contains significantly higher proportions of CGRP⁺ neurons compared to NEN lineage in Wnt1-Cre:*Rosa26* $^{lsl-tdTomato}$ male mice. Data represent mean ± S.E.M. *P*<0.0001; Fisher's exact test. Nuclei in (**b**) and (**c**) are labeled with DAPI (blue). Scale bar for all images denotes 10 μm.

their derivation from a distinct developmental lineage, the mean cell size of MENs was significantly larger than that of NENs (*Figure 4—figure supplement 1a*; n=143 neurons/group observed across three mice; Feret diameter (μm) MENs: 17.47±0.50 SEM, NENs: 13.03±0.36 SEM, p=0.002; Student's t-test).

## An expanded molecular characterization of MENs using unbiased single-cell RNA sequencing (scRNAseq)-based analyses

Recent studies have used single cell or single nuclei-based transcriptomic analyses to query molecular diversity of enteric neurons. These studies have used either used enrichment methods based on FACS sorting NC-derived cells (using reporter-expression from NC-specific lineage fate mapping mice *Wright et al., 2021*; *Zeisel et al., 2018*; *Drokhlyansky et al., 2020*), *Phox2b*-expressing cells (*May-Zhang et al., 2021*), Baf53-derived cells (*Morarach et al., 2021*) or used classical NENs markers such as *Snap25* to drive identification and/or enrichment of enteric neurons (*Drokhlyansky et al., 2020*). Instead of using these strategies, we relied upon another accepted use of scRNAseq, which allows for sequencing of diverse un-enriched cell populations from a tissue, to discover any novel cell types in that tissue (*Pollen et al., 2014*). We performed truly unbiased and agnostic clustering of the scRNAseq data from tissues of two 6-month-old adult male C57BL/6 wildtype mice (*Figure 4a*; *Figure 4—figure supplement 1b, c*). We identified clusters of NC-derived cells by exclusive expression of canonical NC markers *Ret* and *Sox10* (which include the clusters of *Ret*-expressing NENs and *Sox10*-expressing Neuroglia); and the MENs cluster by its co-expression of the genes *Calcb* (CGRP), *Met*, and *Cdh3* (*Figure 4a and b*). With cells from both samples pooled together, we compared 1,737 NC-derived cells across the NENs (77 cells) and Neuroglia (1660 cells) clusters with 2223 cells in the MENs cluster. A list of top marker genes for every cluster are provided in *Supplementary file 2*. It is important to note here that it is well known that enzyme-based dissociation methods are unequally tolerated by diverse cell types, which is known to cause over- or under-representation of several cell types in scRNAseq (*Uniken Venema et al., 2022*; *Wu et al., 2018*). The same is true for neurons, where certain neuronal subtypes may be significantly over- or under-represented in scRNAseq databases, compared to their true representation in tissue (*Tiklová et al., 2019*). Thus, the relative sizes of the NENs and MENs clusters in the scRNAseq data should not be taken as a reliable indicator of their actual proportions in tissue.

The neuronal nature of MENs was first established by testing our scRNAseq dataset for expression of genes known to be expressed by enteric or other neurons. We found detectable expression of genes *Elavl2* (HuB), *Hand2, Pde10a, Vsnl1, Tubb2b, Stmn2, Stx3*, and *Gpr88* in the MENs as well as in the NC-derived cell clusters (*Figure 4b*). While expression *of Elavl2, Stmn2* and *Hand2* has been previously observed in enteric neurons (*Heanue and Pachnis, 2006*; *D'Autréaux et al., 2007*), the expression of neuronal marker genes *Pde10a* (expressed by medium spiny neurons in the striatum *Strick et al., 2010*), *Vsnl1* (expressed by hippocampal neurons *Bernstein et al., 2002*), *Gpr88* (expressed by striatal neurons *Massart et al., 2009*), *Stx3* (expressed by hippocampal neurons *Soo Hoo et al., 2016*), and *Tubb2b* (expressed by adult retinal neurons *Breuss et al., 2015*) has not been studied. Using antibodies against these markers along with co-staining with anti-Hu ANNA1 antisera to label neurons, we visualized the expression of these neuronal markers in both tdTomato⁺ NENs and tdTomato⁻ MENs of the small intestinal myenteric plexus tissue from adult Wnt1-Cre:*Rosa26*$^{lsl-tdTomato}$ mice (*Figure 4c*, *Figure 2—figure supplement 2l–q*). In addition, the MENs cluster was also enriched in previously characterized ENS markers, such as *Ntf3* and *Il18*[2,63] (*Figure 4b*). We validated the MEN-specificity of these markers using immunochemical analyses (*Figure 4c*; *Figure 2—figure supplement 2b–k*). In addition, we also detected >40 neuronally significant genes whose expression was unique to or enriched in MENs scRNAseq cluster, when compared with other larger cell clusters of macrophages, smooth muscle cells, neuroglia, and vascular endothelium (*Figure 4—figure supplement 2*). These genes include neurotransmitter receptor genes (such as *Gabra1, Gabra3, Gria3,*

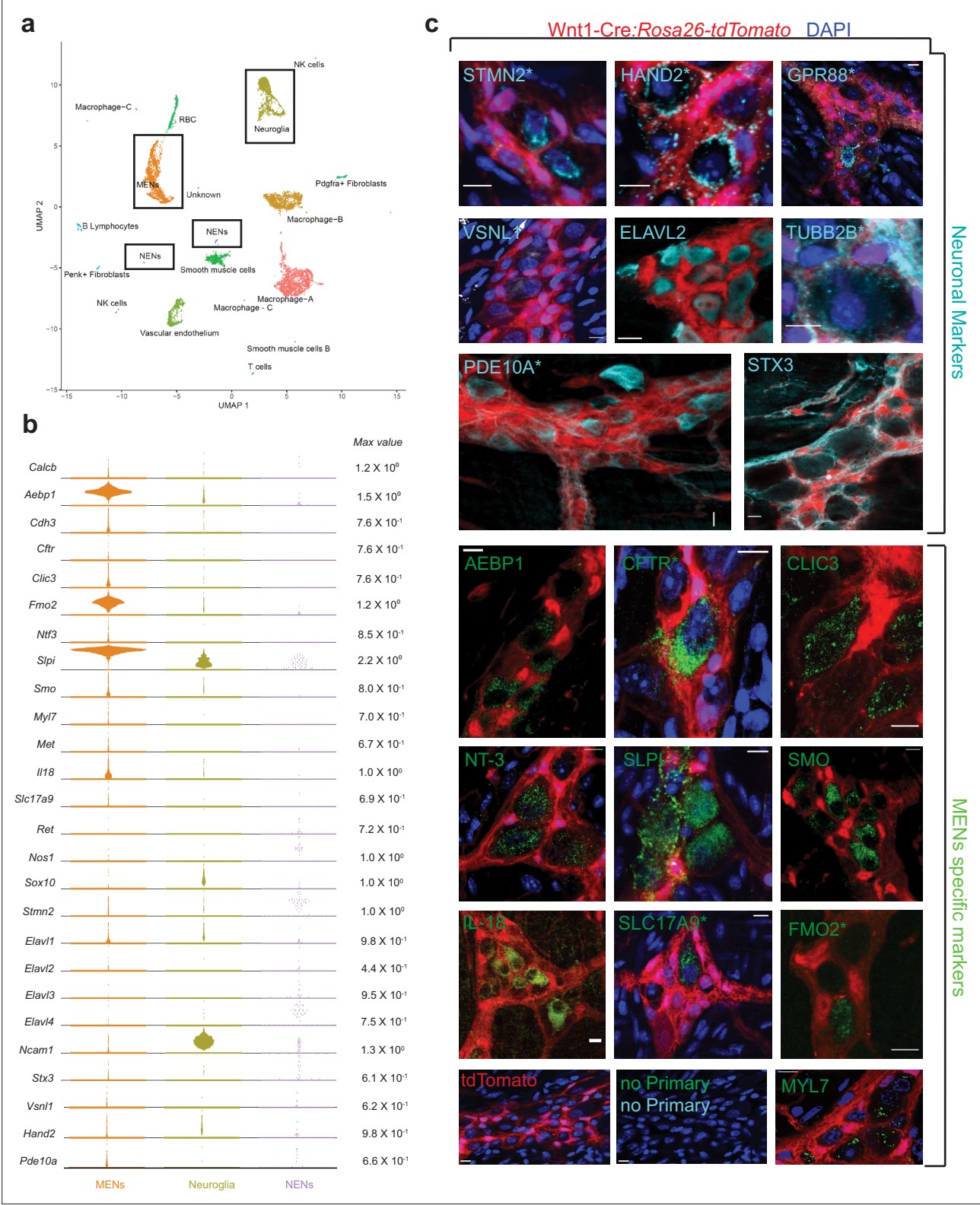

**Figure 4.** scRNAseq-analyses identifies the distinct transcriptomic profile of the MENs. (**a**) UMAP representation of 11,123 sequenced high-quality cells that were identified as meeting a 200 UMI minimum threshold with a mitochondrial read ratio of less than 20%. Clusters were annotated by markers that were found to be cell-type specific by searching UniProt, Allen Cell Atlas and Pubmed databases. Cells of the neural crest lineage were then identified as NENs by their expression of neural crest marker gene *Ret* and Neuroglia by their expression of *Ncam1* and *Sox10*, or as MENs by co-expression of *Calcb* (CGRP), *Cdh3*, and *Met* genes. (**b**) Visualization of expression of select MEN-specific and neuronal markers using quasi-violin

*Figure 4 continued on next page*

*Figure 4 continued*

plots. "Max value" represents the scale for the log-normalized expression of each gene. (**c**) Validation of the MENs-specific marker genes discovered in the scRNAseq analyses by immunohistochemistry and confocal microscopy of small intestinal LM-MP from adult male Wnt1-Cre:*Rosa26* ^lsl-tdTomato^ mice. Expression of neuronal markers STX3, VSNL1, STMN2, HAND2, GPR88, PDE10A, ELAVL2, and TUBB2B (Gray, green arrows) was found in tdTomato⁻ MENs. Immunostaining of the proteins AEBP1, CFTR, CLIC3, NT-3, SLPI, SMO, IL-18, SLC17A9, FMO2 and MYL7 (green; green arrows) was found to be localized to tdTomato⁻ MENs. tdTomato⁺ (red) NC-cells did not immunostain for these markers (red arrows). Panel also shows tissue from Wnt1-Cre:*Rosa26* ^lsl-tdTomato^ mouse with no primary controls (stained with AlexaFluor 488 and AlexaFluor 647 antibodies). Figures with * annotations are 2D representation of 3D stacks of images. Nuclei in some panels were labeled with DAPI (blue). Scale bar denotes 10 µm.

The online version of this article includes the following source data and figure supplement(s) for figure 4:

**Source data 1.** Top expressed genes by the Calcb-expressing clusters and the putative MENs cluster in data from May-Zhang et al, along with top-expressed genes in the MENs cluster in our data.

**Figure supplement 1.** scRNAseq metrics.

**Figure supplement 2.** Expression of neuronally significant genes by MENs.

**Figure supplement 3.** Sub-clustering MENs in the scRNAseq dataset generated from 6-month-old mice.

*Grik5, Grind2d, Npy1r*); ion channels genes (such as *Cacna1a, Cacna1g, Cacnb3, Clcn3*); gap junction genes (*Gjc1, Gjb5*); transient receptor potential channel genes (*Trpv4, Trpc1, Trpc6*); potassium channel genes (such as *Kcnn1, Kcnj8, Kcnd3, Kcnh3, Kcns3*); hormone encoding genes (*Ghrh, Gnrh1, Nucb2*) along with other neuronal genes such as, *Prss12* (encoding for Neurotrypsin *Mitsui et al., 2007*), *Uchl1* (encoding for pan-neuronal marker PGP9.5 *Krammer et al., 1993*), *Cplx2* (encoding for Complexin 2 *Ono et al., 1998*), *Gpm6a* (encoding for neuronal membrane glycoprotein M6-A *León et al., 2021*), and *Vamp2* (encoding for Synaptobrevin 2 *Hoogstraaten et al., 2020*; *Figure 4—figure supplement 2*). Owing to the small size of the NENs cluster, a comparative analysis between these two neuronal lineages for these neuronally significant genes was not possible. However, the discovery of these neuronally significant genes in the MENs cluster, and the subsequent validation of a set of them provides evidence for the neuronal nature of MENs. Further examination of the MENs cluster yielded additional MENs-specific marker genes *Slpi, Aebp1, Clic3, Fmo2, Smo, Myl7*, and *Slc17a9*, whose expression by adult enteric neurons has not been previously described and which we also validated (*Figure 4b*).

Next, we examined the MENs cluster for expression of a mesenchymal gene that would underscore the mesodermal nature of MENs. *Decorin* (*Dcn*) is a gene coding for a chondroitin-dermatan sulphate proteoglycan, which is expressed by mesoderm-derived cells, including fibroblasts and smooth muscle cells (*Neill et al., 2012*). *Decorin* was found to be highly expressed in the MENs cluster in our data (*Figure 5a*), and not by the neurons in the Zeisel et al. database (*Figure 5b*; *Zeisel et al., 2018*). We used a validated antibody against DCN (*Scott et al., 1993*) and found that this protein, known to be specifically expressed by mesoderm-derived cells, is expressed specifically by tdTomato-negative small intestinal myenteric neurons of the Wnt1-Cre:*Rosa26*^lsl-tdTomato^ lineage fate mapping mouse (*Figure 5c*).

These data show that the MENs scRNAseq cluster, which can be identified by its co-expression of neuronal markers (*Elavl2, Pde10a, Hand2*, etc), validated cell surface marker (*Cdh3*), and the mesodermal marker *Dcn* is the same as the population of Hu-expressing neurons that do not express tdTomato in the Wnt1-Cre:*Rosa26*^lsl-tdTomato^ NC lineage fate mapping mouse. We thus establish the distinct nature of MENs based on the co-expression of many neuronally significant genes and of mesodermal genes by this cell population of tdTomato-negative neurons in Wnt1-Cre:*Rosa26*^lsl-tdTomato^ mice. Our experimental strategy of performing scRNAseq on LM-MP cells without any pre-enrichment helps detects this newly characterized cell type. However, it does not allow us to compare and contrast the transcriptomic profiles of MENs and NENs – especially given that we do not know the true size of NENs represented in our dataset. This is exemplified by the fact that cells within the Neuroglia cluster showed expression of glial genes such as *Sox10* as well as neuronal genes such as *Ncam1, Hand2* and *Stmn2*, suggesting that NC-derived neurons may be present within both the smaller NENs cluster as well as the larger Neuroglia cluster (*Figure 4b*).

## Detection of MENs in other murine and human datasets

Since Phox2b, the transcription factor expressed by all ENS cells, shows significant higher expression in some neurons than in all ENS glial cells (*Corpening et al., 2008*, *May-Zhang et al., 2021*)

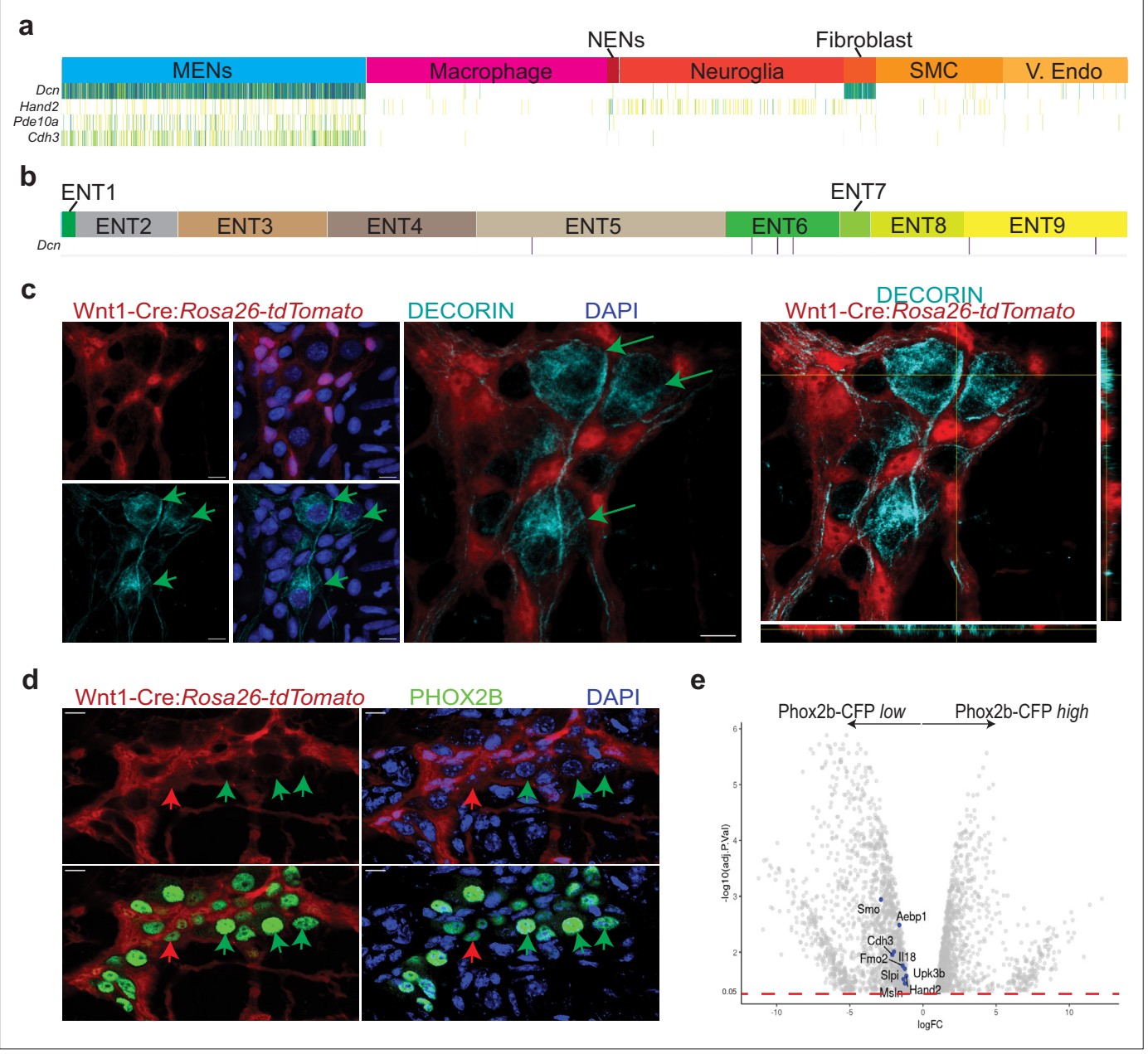

**Figure 5.** MENs express the mesenchymal gene *Decorin* and the ENS-specific gene *Phox2b*. (**a**) Sparkline representation plot of the top 90 percentile of expressed genes in the various scRNAseq cell clusters from the adult murine LM-MP tissue of a 6-month-old mouse shows that the MENs cluster, which expresses the genes *Cdh3*, *Pde10a*, and *Hand2*, also expresses significant amounts of the mesenchymal gene *Decorin* (*Dcn*). Darker colors of the sparkline plot represent higher expression. (**b**) Sparkline representation plot of the neural crest-derived enteric neurons from the Zeisel et al. dataset shows that the *Decorin* gene is not found expressed by most neural crest-derived enteric neurons. (**c**) Two-dimensional representation views and orthogonal view of a myenteric ganglion from the small intestinal LM-MP of adult male Wnt1-Cre:*Rosa26* [lsl-tdTomato] mouse, where tdTomato (red) is expressed by neural crest-derived cells, when immunostained with antibodies against DECORIN (grey) shows that the DECORIN-expressing myenteric cells (green arrows) do not express tdTomato and hence are non-neural crest-derived MENs. Nuclei are labeled with DAPI. Scale bar = 10 μm. (**d**) Representative image of a myenteric ganglion from the small intestinal LM-MP of adult male Wnt1-Cre:*Rosa26* [lsl-tdTomato] mouse, where tdTomato (red) is expressed by neural crest-derived cells, when immunostained with antibodies against PHOX2b (green) shows that the PHOX2b-expression is found in myenteric cells that do not express tdTomato (green arrows, and hence are not neural crest-derived cells, or MENs) as well as in tdTomato-expressing neural crest-derived NENs (red arrow). Nuclei are labeled with DAPI. Scale bar = 10 μm. (**e**) Volcano plot of gene expression profiles of *Phox2b*-expressing cells, which were sorted based on their CFP expression level, shows that the expression of MENs marker genes *Smo*, *Aebp1*, *Cdh3*, *Fmo2*, *Il18*, *Slpi*, *Upk3b*, *Msln*, and *Hand2* is significantly enriched in the Phox2b-CFP[low] fraction. Red dotted line shows the p[adjusted] value of 0.05.

The online version of this article includes the following figure supplement(s) for figure 5:

*Figure 5 continued on next page*

**Figure supplement 1.** Expression of mesodermal marker MYH11, and reporter expression under the Baf53b-Cre transgenic line in post-natal myenteric neurons.

reasoned that flow sorted nuclei of Phox2b-CFP^high cells from a Phox2b-CFP reporter mouse would provide for an enriched population of ENS neurons that can be used for single nuclei RNA sequencing (snRNAseq). They first performed bulk-RNAseq on Phox2b-CFP⁺ nuclei that were flow sorted based on CFP-expression level into CFP^low and CFP^high fractions, of which the CFP^low fraction was found enriched with glial-specific gene expression profile. *May-Zhang et al., 2021* used this to provide a rationale for performing snRNAseq on Phox2b-CFP^high nuclei, which they presumed to consist of all enteric neurons. Since, MENs make up roughly half of all adult ENS neurons, we hypothesized that they should also express Phox2b. By using a PHOX2b-specific antibody, we validated the expression of this important transcription factor in MENs (*Figure 5d*). Next, we tested whether we could detect some of the genes that show enriched expression in MENs (namely, *Smo, Aebp1, Cdh3, Fmo2, Il18, Slpi, Upk3b, Msln, Hand2*) in the Phox2b⁺ bulkRNAseq data from *May-Zhang et al., 2021*, and found that not only were these genes expressed by their Phox2b⁺ cells, but expression of these genes was also significantly enriched in the Phox2b-CFP^low fraction (*Figure 5e*) that was not used for subsequent snRNAseq experiments. These data provide evidence that MENs express *Phox2b*, both transcriptomically, as well as at the protein level. We next queried whether any MENs were represented in the May-Zhang et al.'s snRNAseq data generated on Phox2b-CFP^high nuclei, given that they reported on the presence of a cluster of adult enteric neurons that expressed mesenchymal markers. We again tested whether the MENs-expressed genes, such as *Decorin*, etc were also expressed by the 'mesenchymal' neurons in the *May-Zhang et al., 2021* dataset. We found a high degree of similarity between the top genes expressed by the MENs cluster in our scRNAseq dataset and the 'mesenchymal' neurons in the May-Zhang et al.'s snRNAseq dataset (*Figure 4—figure supplement 1d*, *Figure 4—source data 1*). Furthermore, examination of the top genes shows that this *Decorin*-expressing neuronal sub-population is significantly different from the other *Calcb*-expressing neuronal populations, which we presume to be the CGRP-expressing NENs (*Figure 4—source data 1*).

To confirm that the 'mesenchymal' neurons identified by *May-Zhang et al., 2021* are MENs, we used an agnostic bioinformatics projection-based approach which we have previously published (*Stein-O'Brien et al., 2019*). This approach allows us to learn latent space representations of gene expression patterns using single-cell transcriptomic data, generating patterns that correspond to cell-type-specific gene expression signatures (*Stein-O'Brien et al., 2019*). We can then project other single cell and bulk RNA transcriptomic datasets into these learned patterns to accurately quantify the differential use of these transcriptional signatures in target data across platforms, tissues, and species (*Stein-O'Brien et al., 2019*). In this instance, gene expression patterns were learned on our murine single cell transcriptomic data using non-negative matrix factorization (NMF) dimensionality reduction and the data was decomposed into 50 distinct NMF-patterns (*Stein-O'Brien et al., 2019*; *Stein-O'Brien et al., 2018*; *Figure 6a*). Four NMF patterns were identified that were specific to MENs (*Figure 6a and b*; *Figure 4—figure supplement 1e*). We next projected the snRNAseq dataset from *May-Zhang et al., 2021* into the four MEN-specific patterns and found that all these patterns specifically labeled the cluster of 'mesenchymal' neurons in the May-Zhang et al. dataset of the adult ileal ENS (*Figure 6b*). Further, upon clustering the nuclei based on their projection weights from the four MEN-specific patterns, we identified a cluster of nuclei that showed higher usage of all four patterns, when compared to all the other nuclei sequenced in the dataset (*Figure 4—figure supplement 1f*). The genes that define this cluster of cells are similar to the top-expressed genes that define our MENs cluster (*Figure 4—figure supplement 1g*, *Figure 4—source data 1*). This approach, together with the high similarities in the top-expressed genes, establishes that cluster of neurons annotated as 'mesenchymal' neurons by May-Zhang et al. are indeed MENs. Further, one of the markers used by May-Zhang et al. to describe their mesenchymal neuronal cluster was the gene *Myh11* (*May-Zhang et al., 2021*), which is known to be expressed by mesoderm-derived smooth muscle cells in the adult gut (*Lee et al., 2015*). To test whether *Myh11* is expressed by MENs, we immunolabeled the small intestinal LM-MP tissue from an adult Wnt1-Cre:*Rosa26^lsl-tdTomato* mouse with a monoclonal antibody against MYH11 (Invitrogen) and found that tdTomato-negative and Hu⁺ neurons exclusively immunolabeled for MYH11 (*Figure 5—figure supplement 1a*). We confirmed that a subpopulation

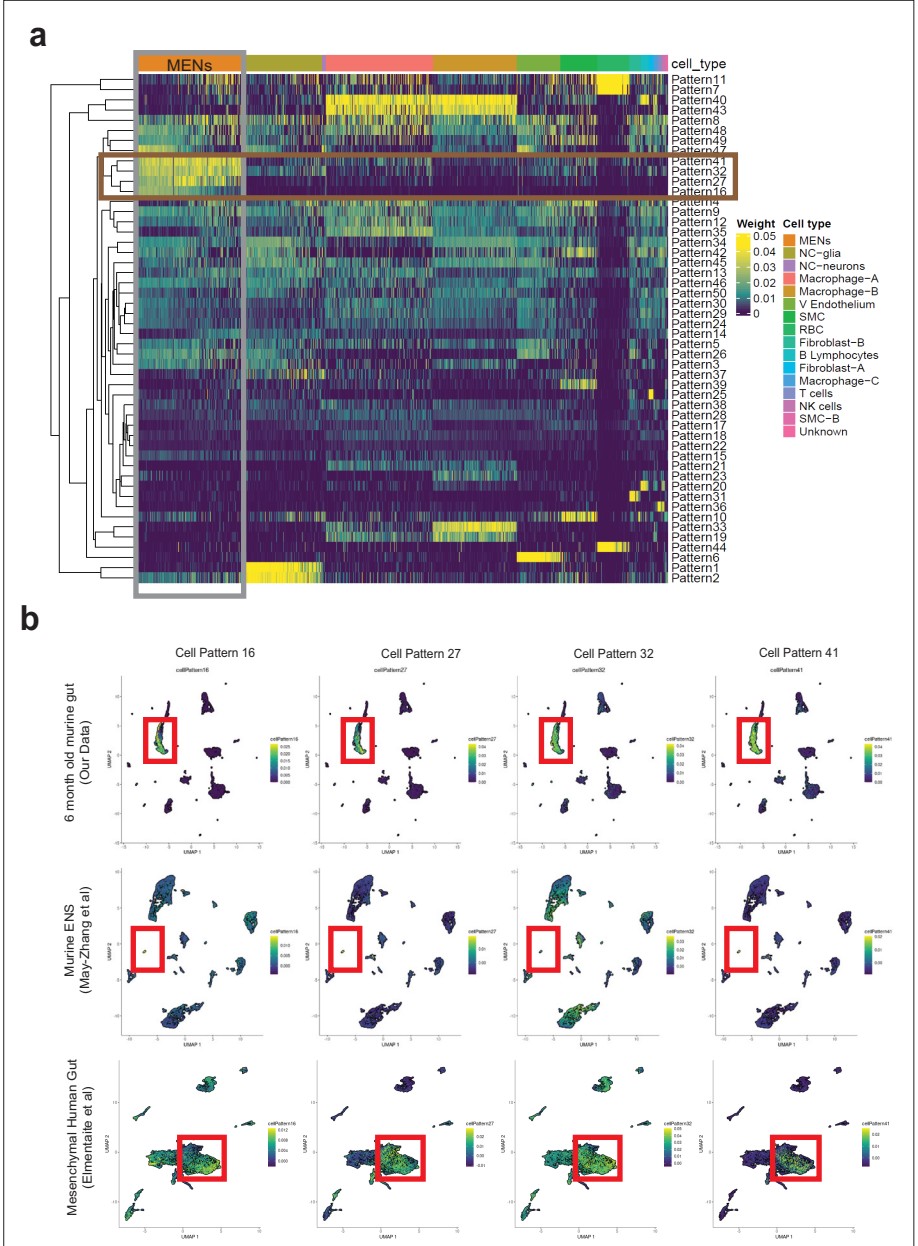

**Figure 6.** Computational analyses using *projectR* identifies MENs in publicly available murine and human transcriptomic datasets. (**a**) Heatmap of cell weights for the patterns learned by NMF (k=50). Hierarchical clustering was calculated using Euclidean distance. Multiple clusters annotated as the same cell type are merged. The four most specific MENs patterns (16, 27, 32, and 41) are selected. (**b**) In addition to using the four MEN-specific NMF patterns to label the MENs cluster in our scRNAseq dataset (plots in top row), two other transcriptomic datasets: May-Zhang et al.'s murine ileal ENS snRNAseq dataset *May-Zhang et al., 2021* (plots in second row), and *Elmentaite et al., 2021* gut mesenchymal scRNAseq dataset (plots in third row), were projected into the defined four MEN-specific pattern space using *ProjectR*. Raw projected cell weights were visualized using previously learned UMAP embedding. The cell clusters that show high pattern usage are shown bounded by the red square. Plots in the final row shows projection of healthy post-natal mesenchymal cells from *Elmentaite et al., 2021* into the four MEN-specific NMF patterns, which again show a population of cells showing high MEN-specific pattern utilization.

of adult enteric neurons expressed Myh11 by immunolabeling small intestinal LM-MP tissues from a tamoxifen treated Myh11^Cre-ERT2^:Rosa26^lsl-YFP^ transgenic mouse line with anti-YFP/GFP and anti-Hu antibodies (*Figure 5—figure supplement 1b*). Anti-MYH11 antibodies labeled circular smooth muscle in the human duodenum, where it did not immunolabel nerve fibers and some submucosal neurons immunostained with antibodies against the pan-neuronal marker PGP9.5 but showed immunolabeling against PGP9.5-expressing myenteric neurons in the adult healthy small intestinal tissue (*Figure 5—figure supplement 1c, d*). Thus, we confirm that the MENs identified by us correspond to the cluster of 'mesenchymal neurons' observed by May-Zhang et al. As compared to our scRNAseq dataset, where MENs comprise of a significantly large numbers of cells, the MENs cluster in the May-Zhang et al. dataset is significantly smaller, possibly due to the fact that a large number of MENs were left unsequenced in their Phox2b-CFP^low^ fraction.

Similarly, *Drokhlyansky et al., 2020* using MIRACL-seq found a small cluster of cells that express the MENs-signature genes (*Cdh3, Dcn, Slpi, Aebp1, Wt1, Msln, Fmo2*). However, the neuronal nature of these cells was not correctly identified as the study used a gene signature profile specific for NC-derived neurons (derived from their experiment with enriched Wnt1-Cre2:Rosa26^lsl-tdTomato+^ NC-derived cells). As a result, this cluster of cells – which shows similarity to the MENs transcriptomic profile, was instead annotated as *Mesothelial*. This 'mesothelial' cluster in the *Drokhlyansky et al., 2020* dataset shows similar gene expression profile as that of the cluster of 'mesenchymal neurons' from the *May-Zhang et al., 2021* dataset (*Figure 4—source data 1*), especially given that many of the marker genes for both clusters are similar (*Lrrn4, Rspo1, Msln, Upk3b, Upk1b, Gpm6a, Wnt5a, Gpc3, Sulf1, Muc16*). The 'mesothelial' cluster also expresses neuronally significant genes such as *Gabra3* (GABA receptor A subunit 3 *Purves-Tyson et al., 2021*), *Prss12* (Neurotrypsin *Mitsui et al., 2007*; *Mitsui et al., 2009*), *Synpr* (Synaptoporin *Singec et al., 2002*), *Trpv4* (Transient receptor potential cation channel subfamily V member 4 *Fichna et al., 2015*), and *Nxph1* (Neurexophillin-1 *Born et al., 2014*). Unfortunately, a deeper interrogation of this dataset was not possible due to the size and the manner in which the original data was processed *Drokhlyansky et al., 2020*. In addition, MENs were not represented in the snRNAseq carried out in *Wright et al., 2021*, which was based solely on NC-derived cells.

In contrast to these snRNAseq-based studies, the Marklund Lab in two studies performed single cell RNA-sequencing (scRNAseq) on juvenile ENS cells, first on tdTomato^+^ cells from P21 Wnt1-Cre:Rosa26^lsl-tdTomato^ lineage fate mapping mice in *Zeisel et al., 2018* and then from the newly characterized Baf53b-Cre:Rosa26^lsl-tdTomato^ BAC-transgenic mouse line *Zhan et al., 2015* in *Morarach et al., 2021*. Barring 2 small clusters (Clusters 5 and 11), the other neuronal clusters identified in *Morarach et al., 2021* mapped to the NC-derived clusters identified earlier by them in *Zeisel et al., 2018*. The gene expression signature of these two clusters (Cluster 5: *Sst, Calb2*; Cluster 11: *Npy, Th, Dbh*) did not match the signature of our MENs cluster or that of the cluster of mesenchymal neurons in the *May-Zhang et al., 2021* study (*Figure 4—source data 1*). We reasoned that the lack of MENs in the *Morarach et al., 2021* data may be driven by the non-comprehensive nature of Baf53b-Cre in this BAC transgenic mouse line labeling all neurons. On immunolabeling LM-MP tissues from two P21 Baf53b-Cre:Rosa26^lsl-tdTomato^ mice with antibodies against Hu, we enumerated 1312 Hu^+^ cells in the ileum (759 from mouse 1 and 553 from mouse 2), 5758 Hu^+^ cells in the proximal colon (3411 from mouse 1 and 2347 from mouse 2), and 3352 Hu^+^ cells in the distal colon (1894 from mouse 1 and 1458 from mouse 2), and found that a significant population of myenteric neurons did not express tdTomato (*Figure 5—figure supplement 1e, f*). Given this non-comprehensive nature of the Baf53b-Cre BAC transgenic line in labeling all myenteric neurons, and the congruence between the transcriptomic profiles of cells analyzed using the Baf53b-Cre and the NC-specific Wnt1-Cre lines, we infer that molecular taxonomy performed on enteric neurons in the *Morarach et al., 2021* dataset is restricted to NENs.

In a recent report, *Elmentaite et al., 2021* performed scRNAseq on dissociated single cells from the human gut at fetal, juvenile, and adult ages and catalogued multiple intestinal cell-types to generate a comprehensive gut cell atlas. Canonical enteric neurons, while adequately represented in the fetal ages, were not found in the juvenile and adult ages which were dominated by mesenchymal and other cell types (*Elmentaite et al., 2021*). Given that our scRNAseq approach of sequencing diverse cell types from the murine gut wall without any marker-based pre-selection was similar to this approach in human tissue, we reasoned that MENs would also be represented in their description of mesenchymal cell clusters, especially in the juvenile and adult ages. To detect putative MENs in this human gut cell atlas data, we projected the entire human mesenchymal cell scRNAseq dataset into

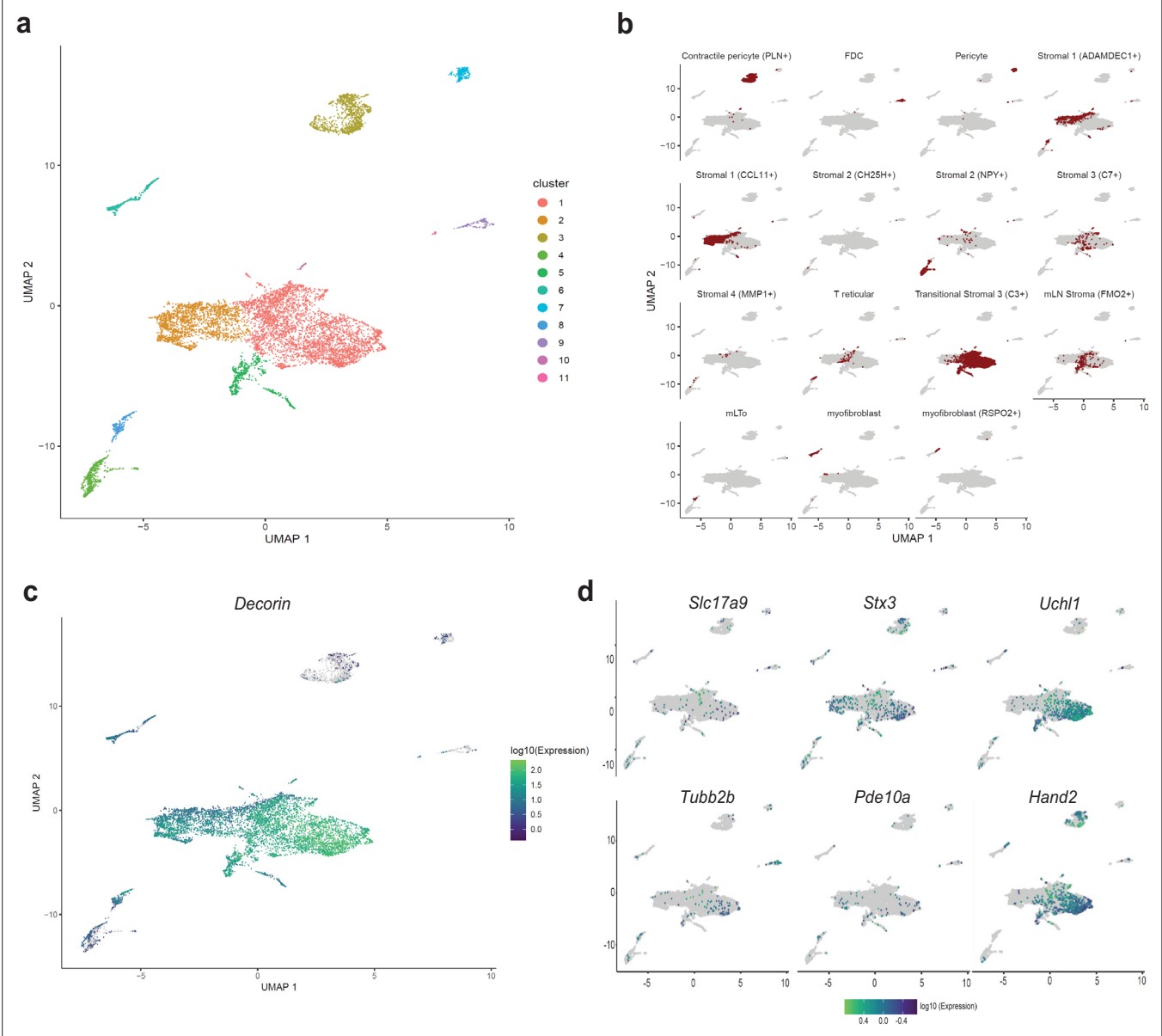

**Figure 7.** Detection of MENs in human single cell RNA sequencing data. (**a**) UMAP representation of scRNAseq data from *Elmentaite et al., 2021*'s healthy post-natal mesenchymal cells from the human gut . (**b**) Breakdown for previously published annotated features of the post-natal health subset of the gut cell atlas as presented in *Elmentaite et al., 2021*. (**c**) Expression of the MENs marker gene *Dcn* in the UMAP representation of healthy post-natal data from *Elmentaite et al., 2021*., which was annotated as clusters of mesenchymal cells. (**d**) Expression of neuronal marker genes *Slc17a9*, *Stx3*, *Uchl1*, *Tubb2b*, *Pde10a*, and *Hand2* in the UMAP representation of healthy post-natal data from *Elmentaite et al., 2021*, which was annotated as clusters of mesenchymal cells.

the four murine MEN-specific NMF patterns and detected putative human MENs clusters (*Figure 4—figure supplement 1h*). Since our identification of MENs and the generation of MEN-specific NMF patterns was based on post-natal healthy murine tissue, we next projected data from only post-natal healthy mesenchymal clusters into the four MEN-specific NMF patterns (*Figure 6b*). The four patterns identified a population of cells within the post-natal intestinal mesenchymal cell clusters suggesting that these cells are putative MENs in the human gut (*Figure 6b*). These putative MENs, which map to Cluster 1 in our representation of the intestinal mesenchymal cells from the post-natal gut cell atlas data (*Figure 7a*) were annotated previously by Elmentaite et al. as transitional stromal cells (*Figure 7b*). We next tested and found that the MENs marker *Dcn* (*Figure 7c*) along with pan-neuronal *Uchl1* among other neuronal markers *Hand2*, *Stx3*, *Slc17a9*, *Pde10a*, and *Tubb2b* (*Figure 7d*) that we

earlier showed to be expressed by adult murine MENs, were expressed by the putative MENs in the adult human gut.

We hypothesized that the reason these cells were not annotated as enteric neurons was due to the use of a NENs-restricted gene list. Apart from the NENs marker *Ret*, such gene lists often contain the pre-synaptic gene *Snap25*, a component of the SNARE complex, which is widely assumed to be a pan-neuronal marker in the adult ENS (*Zeisel et al., 2018*; *Drokhlyansky et al., 2020*; *Matheis et al., 2020*; *Barrenschee et al., 2015*). While we have previously observed the NENs-specific nature of RET, our transcriptomic data suggests that MENs either do not express *Snap25* or do so in very small amounts (*Figure 4—figure supplement 1d*). We tested the expression of SNAP-25 by NENs

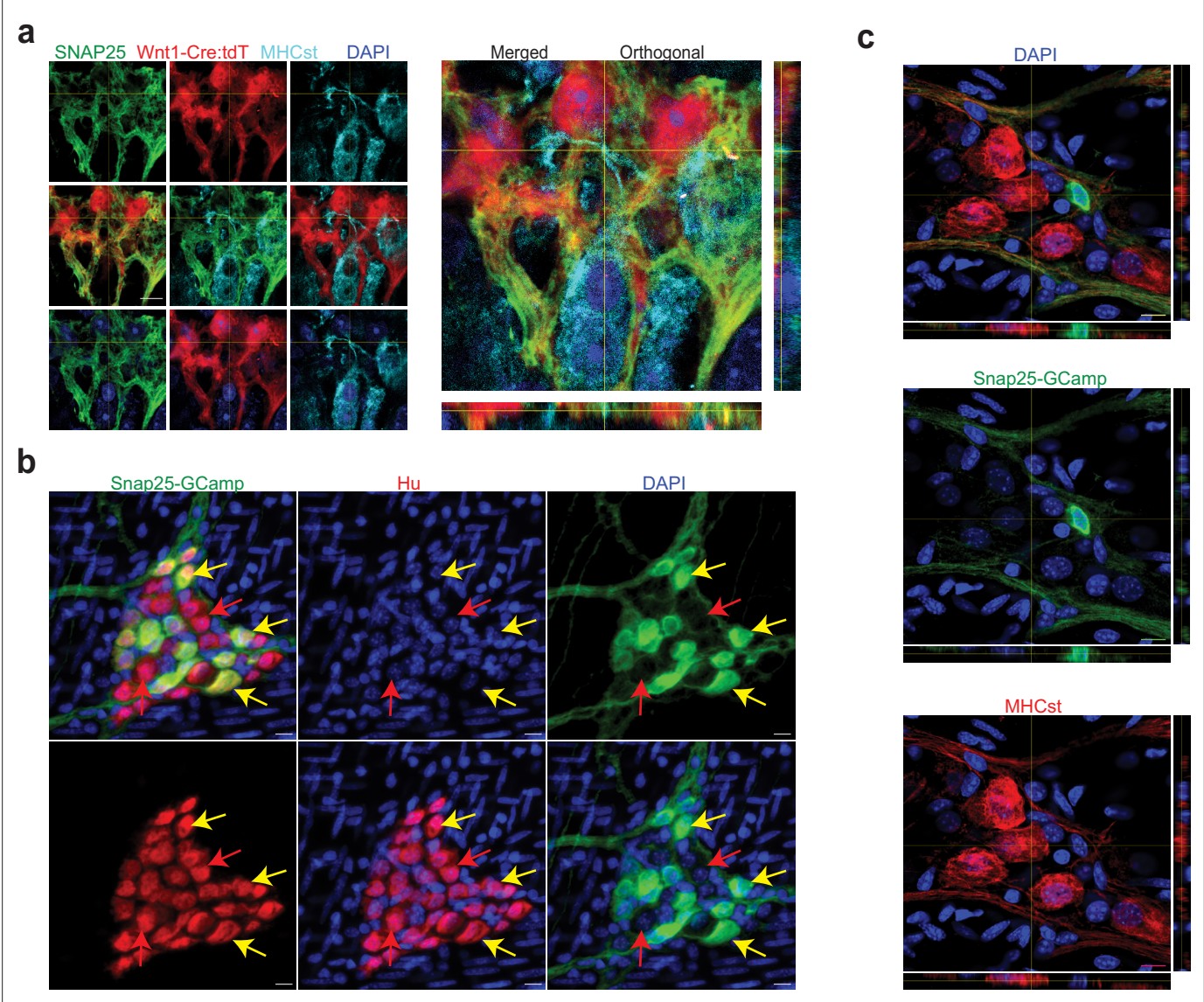

**Figure 8.** Validation of SNAP-25 as a NENs marker. (**a**) SNAP-25 expression is restricted to the neural crest lineage in the adult myenteric ganglia. SNAP-25 expression (green) co-localizes with tdTomato (red) but not with the MENs marker MHCst (cyan) as observed in 2D and orthogonal views of a myenteric ganglia from a Wnt1-Cre:*Rosa26* ^*lsl-tdTomato*^ mouse that was immunolabeled with antibodies against MHCst and SNAP-25. Nuclei were labeled with DAPI (blue). Scale bar denotes 10 μm. (**b**) In the adult male *Snap25*^*GcaMP*^ knock-in mice, the expression of Snap25-GCaMP/GFP (green) is restricted to a subset of Hu-expressing (red) myenteric neurons in the adult murine small intestinal myenteric ganglia (yellow arrows), while many neurons (red arrows) do not show any detectable expression of *Snap25*. Nuclei were labeled with DAPI (blue). Scale bar denotes 10 μm. (**c**) Orthogonal views of z-stack of an image of the myenteric ganglion from the small intestinal LM-MP tissue from an adult male *Snap25*^*GcaMP*^ knock-in mice shows that the expression of Snap25-GCaMP/GFP (green) is exclusive of the expression of the MENs marker MHCst (red). Nuclei were labeled with DAPI (blue). Scale bar denotes 10 μm.

and MENs in the myenteric ganglia of an adult Wnt1-Cre:*Rosa26*[lsl-tdTomato] mouse and found that SNAP-25 expressing neurons expressed tdTomato, but not the MENs-marker MHCst, suggesting the NEN-specific expression of this canonical marker for synaptic neurons (*Figure 8a*). Next, we used a validated *Snap25*[GCaMP] knock-in mouse line (*Shore et al., 2020*; *Steinmetz et al., 2017*; *Zatka-Haas et al., 2021*), where GCaMP/GFP is knocked in at the *Snap25* locus and hence is expressed by *Snap25*-expressing cells, to confirm that the expression of *Snap25* is indeed restricted to a subset of myenteric neurons in the adult LM-MP layer. By using antibodies against Hu and against GFP, we found that the expression of *Snap25*[GCaMP] is indeed restricted to a subset of adult myenteric neurons (*Figure 8b*). Finally, by immunolabeling adult small intestinal LM-MP of *Snap25*[GCaMP] mice

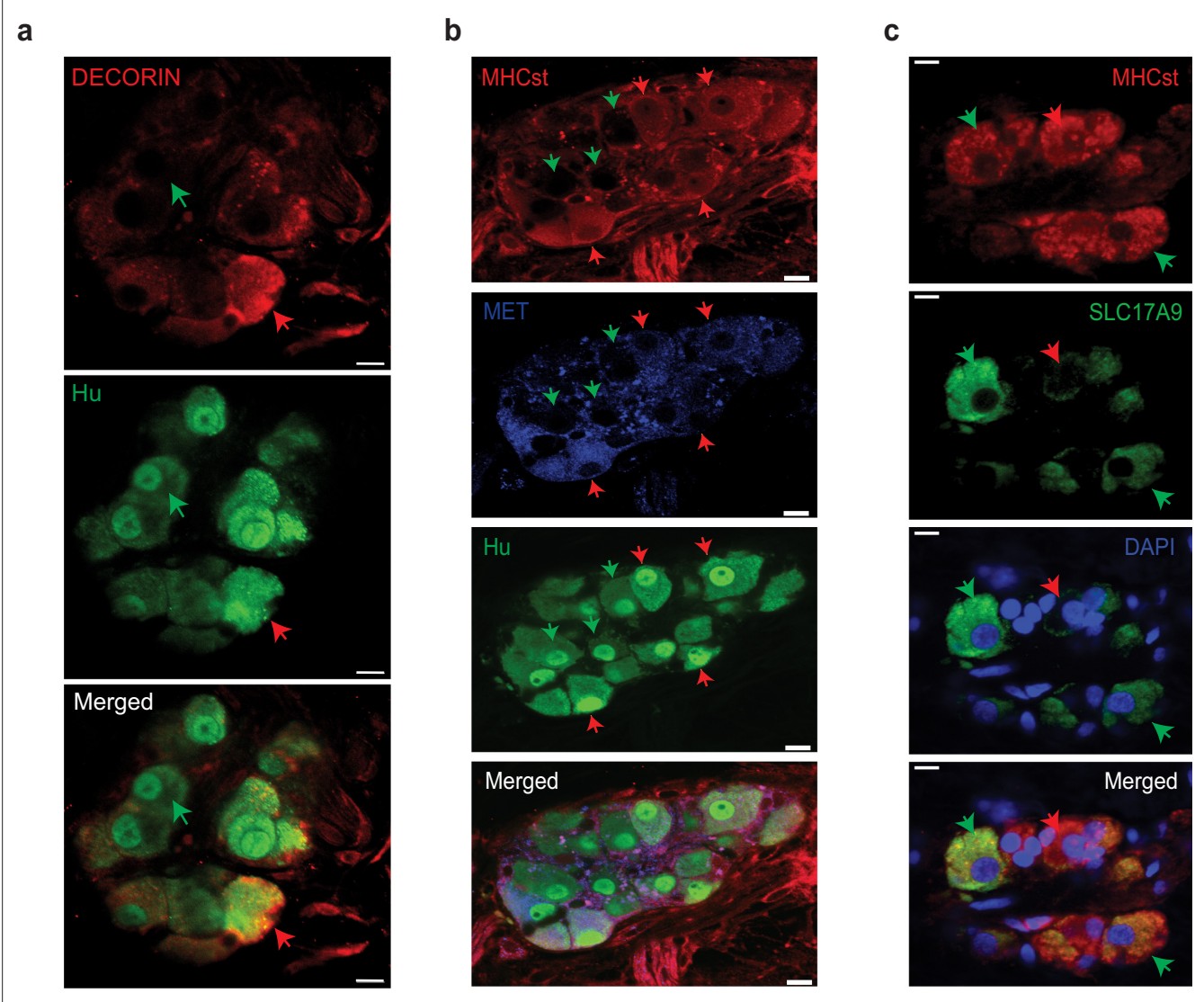

**Figure 9.** Observation of MENs marker expression in the adult human myenteric ganglia. (**a**) Hu-expressing small intestinal myenteric neurons (green) from the normal human duodenal tissue, when immunolabeled with antibodies against the MENs marker DECORIN (red) identifies putative human MENs (red arrows) and NENs (green arrows) by presence or absence of DECORIN immunolabeling. Scale bar denotes 10 μm. (**b**) Hu-expressing small intestinal myenteric neurons (green) from the normal human duodenal tissue, when immunolabeled with antibodies against the MENs marker MHCst (red) and MET (blue) identifies putative human MENs (red arrows) and NENs (green arrows) by presence or absence of these MENs markers. Scale bar denotes 10 μm. (**c**) Immunolabeling myenteric tissue with antibodies against the MENs markers SLC17A9 (green, green arrows) and MHCst (red) shows SLC17A9 expression in a subset of MHCst-expressing (red) neurons. Subset of MHCst-expressing cells do not express SLC17A9 (red arrow). Nuclei are labeled with DAPI (blue) in the normal human duodenal tissue. Scale bar denotes 10 μm.

The online version of this article includes the following figure supplement(s) for figure 9:

**Figure supplement 1.** Putative mesoderm-derived enteric neurons (MENs) are present in adult human myenteric ganglia.

with antibodies against MHCst and against GFP, we confirmed the lack of *Snap25* expression by a population of MENs (*Figure 8c*). The low or lack of expression of *Ret, Snap-25, Elavl3* and *Elavl4* in MENs as observed both by us and *May-Zhang et al., 2021* may help explain why these canonical gene lists missed the correct identification and annotation of MENs' neuronal nature in human data-sets (*Elmentaite et al., 2021*). Next, we examined the expression of MENs markers MHCst, MET, SLC17A9, and DECORIN in LM-MP tissues from adult humans with no known gut motility disorder and found them to be expressed by a population of myenteric neurons in normal adult human ENS (*Figure 9*; *Figure 9—figure supplement 1*). Thus, our bioinformatic approaches and immunofluo-rescence analyses of MEN-specific markers in adult human gut together shows that the human gut similarly contains a population of MENs.

We next tried sub-clustering the MENs cell cluster in our scRNAseq data to study whether indi-vidual subclusters could be discriminated on the basis of MENs markers or other genes. However, at the current sequencing depth, these clusters did not yield meaningful information on specific marker genes that could be used to define them (*Figure 4—figure supplement 3*).

## The proportion of mesoderm-derived neurons expands with age to become the dominant population in the aging ENS

Since the ENS at birth consists solely of NENs (*Laranjeira et al., 2011*), we next studied the birth-date and eventual expansion of the MEN lineage in the post-natal gut. Using Wnt1-Cre:*Rosa26*[lsl-tdTomato] mouse line, we enumerated the tdTomato[-] MENs in LM-MP at different ages and found a significant age-associated increase in MENs proportions (*Figure 10a, b and c*; One-way ANOVA, $F$=117.6, DFn, DFd = 5, 12; p<0.0001). At P11, MENs were found only in few isolated myenteric ganglia (*Figure 10a*) and together represented only ~5% of all myenteric neurons (tdTomato[-] neurons: 4.12%±2.98 SEM; n=1327 Hu[+] neurons counted from three mice), suggesting that they originate shortly before this age. The proportion of MENs in myenteric ganglia rises sharply thereafter: by P22, they account for ~30% of all neurons (tdTomato[-] neurons: 29.63%±1.229 SEM; n=742 neurons counted from three mice); and at P60 they represent ~45% of all neurons (tdTomato[-] neurons: 46.38%±4.62 SEM; n=700 neurons counted from three mice). During the adult ages, the proportions of MENs in the adult ENS remains relatively stable as at P180, MENs continue to represent roughly half of all myenteric neurons (tdTomato[-] neurons: 57.29%±3.62 SEM; n=586 neurons counted from three mice). However, MENs dominate the proportions of the aging ENS as at the very old age of 17 months (P510), the ENS is populated almost exclusively by MENs (*Figure 10b*; tdTomato[-] neurons: 95.99%±1.62 SEM; n=996 neurons counted from three mice). Thus, the proportions of NENs and MENs in the myenteric ganglia can be used as a biomarker for deducing ENS age, as a healthy ENS dominant in NENs would be juve-nile, one with roughly equal proportions would be adult, and an aging ENS is dominated by MENs.

We next tested whether the representation of MEN-specific transcriptomic signatures show a similar increase during the maturation and aging of the human gut. We tested whether the repre-sentation of the four MEN-specific NMF patterns (*Figure 5a*) increased with age in the human transcriptomic data from *Elmentaite et al., 2021*. We grouped the transcriptomic data from the wide-range of healthy post-natal specimens of the gut cell atlas into three groups: Juvenile (containing data on 2910 cells represented by 10 samples within the age range of 4–12 years), Adult (containing data on 3696 cells represented by 36 samples within the age range of 20–50 years), and Aging (containing data on 2848 cells represented by 37 samples within age range of 60–75 years) and tested whether the representation of the four MEN-specific patterns changed significantly between these age groups. Barring pattern 16 which showed a non-significant increase in projec-tion weights between the Juvenile and Aging samples (One-way ANOVA, $F$=2.10, DFn, DFd = 2, 8, p=0.18), projection weights of MEN-specific pattern 27 (One-way ANOVA, $F$=16.08, DFn, DFd = 2, 8, p=0.0016), pattern 32 (One-way ANOVA, $F$=8.13, DFn, DFd = 2, 8, p=0.01) and pattern 41 (One-way ANOVA, $F$=9.818, DFn, DFd = 2, 8, p=0.007) all show significant age-associated increase in the human gut tissue (*Figure 10d*, every datapoint refers to mean projected pattern weight for cells within a defined age or age range). This suggests that analogous to the age-associated increase in MENs we observed in murine gut (*Figure 10b*), the proportions of MENs in the human gut also might increase with age. We also found that the representation of MENs cluster (Cluster 1) in the post-natal human gut cell atlas data expanded with maturation and age (*Figure 9—figure supplement 1d*).

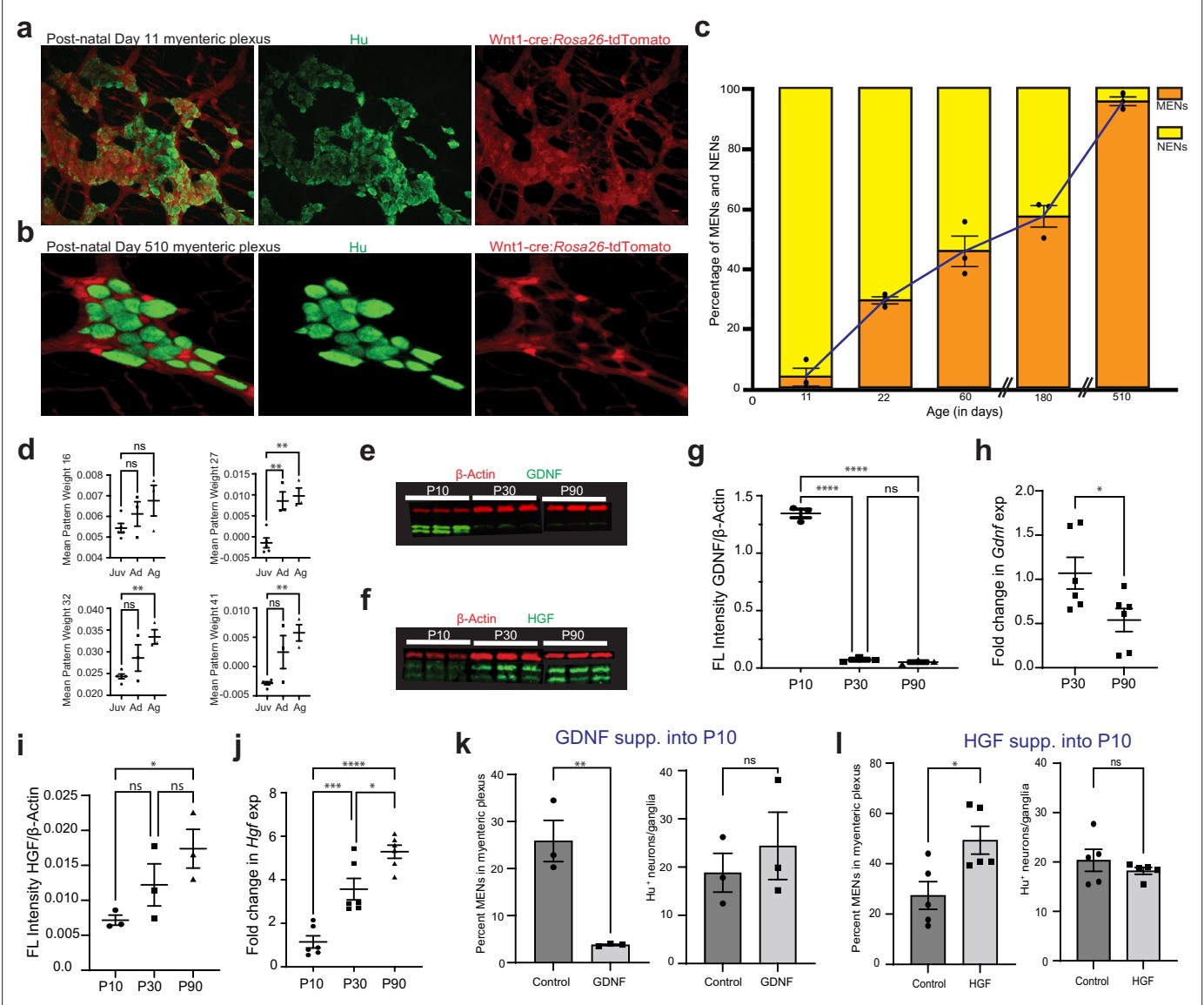

**Figure 10.** GDNF and HGF signaling regulate age-dependent changes in NENs and MENs proportions. (**a, b**) Immunostaining myenteric plexus tissue from juvenile and mature Wnt1-Cre:*Rosa26* [lsl-tdTomato] mice with antibodies against the pan-neuronal marker Hu (green). (**c**) Age-associated loss of NENs and gain of MENs in the small intestinal LM-MP of maturing and aging Wnt1-Cre:*Rosa26* [lsl-tdTomato] mice. Data represent mean ± S.E.M. (**d**) Mean ± SEM of the four MEN-specific pattern weights in the human mesenchymal cell populations from *Elmentaite et al., 2021*, wherein data from age ranges of 4–12 years was clubbed together as Juvenile (Juv), data from age ranges of 20–50 years was clubbed together as Adult (Ad), and data from age ranges of 60–75 years was clubbed together as Aged (Ag). Every datapoint refers to mean projected pattern weight for cells within a defined age or age range, where Juv data comprise of ages 4, 6, 9, 10, 12 years; Ad data comprise of age ranges 20–25, 25–30, and 45–50 years; and finally Ag data comprise of age ranges 60–65, 65–70, and 70–75 years. One-way (ANOVA **=p < 0.01). (**e**) Western blot analyses of GDNF (green) and the house-keeping protein β-actin (red) expression in LM-MP tissues from mice of ages P10, P30, and P90. (n=3 mice per group; each sample is a biological replicate). Fluorescent intensities of the two bands of GDNF (that correspond to ~25 kD bands of protein marker) were quantified together. The lower band of GDNF is present only in the P10 tissues and disappears in P30 and P90 adult murine tissues. (**f**) Western blot analyses of HGF (green) and the house-keeping protein β-Actin (red) expression in LM-MP tissues from mice of ages P10, P30, and P90. (n=3 mice per group; each sample is a biological replicate). Fluorescent intensities of the two bands of HGF (that are between 50 kD and 37 kD bands of the protein marker) were quantified together. (**g**) The normalized fluorescent intensity of GDNF protein to house-keeping protein β-Actin compared between the three age groups. GDNF presence was highest in P10 group and was significantly reduced in P30 and P90 groups. Data represent mean ± S.E.M. One-way ANOVA **** p<0.001. (**h**) Age-dependent decrease in *Gdnf* mRNA transcript expression (normalized to the house-keeping gene *Hprt*) in the myenteric plexuses of P30 and P90 old mice. Data represent mean ± S.E.M. Student's t-test *p<0.05. (**i**) The normalized fluorescent intensity of HGF protein to house-keeping protein β-Actin was compared between the three age groups. HGF expression significantly increased from P10 through P90. Data represent mean ± S.E.M. One-way ANOVA *p<0.05. (**j**) Age-dependent increase in *Hgf* mRNA transcript expression (normalized to the house-keeping gene *Hprt*) in the myenteric plexuses of P10, P30, and

*Figure 10 continued*

P90 old mice. Data represent mean ± S.E.M. One-way ANOVA * p<0.05, *** p<0.01, **** p<0.001. (**k**) Percent proportions of tdTomato⁻ MENs and mean Hu⁺ neurons/ganglia in LM-MP of cohorts of Wnt1-Cre:*Rosa26* ^lsl-tdTomato^ mice that were dosed with GDNF or Saline from P10 to P20 age. Data represent mean ± S.E.M. Student's t-test ** p<0.01. (**l**) Percent proportions of tdTomato⁻ MENs and mean Hu⁺ neurons/ganglia in LM-MP of cohorts of Wnt1-Cre:*Rosa26* ^lsl-tdTomato^ mice that were dosed with HGF or Saline from P10 to P20 age. Data represent mean ± S.E.M. Student's t-test * p<0.05.

The online version of this article includes the following source data and figure supplement(s) for figure 10:

**Source data 1.** Western blot data of RET and GDNF proteins.

**Source data 2.** Western blot data of HGF protein.

**Figure supplement 1.** HGF and GDNF ratios in maturing and adult murine gut.

## GDNF and HGF levels regulate the populations of the neural crest-derived and the mesoderm-derived neurons, respectively, in the maturing ENS

GDNF-RET signaling is responsible for proliferation and neurogenesis from NC-derived ENS precursor cells during development as well as for the survival of *Ret*-expressing enteric neurons (*Gianino et al., 2003*; *Natarajan et al., 2002*; *Taraviras et al., 1999*; *Rodrigues et al., 2011*). Similarly, HGF signaling has been shown to be essential for the proliferation of mesoderm-derived cells (*Amano et al., 2002*). Since the expression of the HGF receptor MET and the GDNF receptor RET is exclusive to MENs and NENs respectively, we studied the correlation between age and levels of HGF and GDNF in LM-MP (*Figure 10e and f*). Given that both GDNF, as well as HGF expression is found in the LM-MP layer (*Avetisyan et al., 2015*; *Brun et al., 2013*; *Jangphattananont et al., 2019*), we used LM-MP tissue to study how the levels of these two trophic factors change in and around the ENS tissue during maturation. We found that in agreement with a previous report (*Amano et al., 2002*), GDNF protein levels are markedly reduced between the P10 and the P30 ages and remains reduced thereafter up to the P90 age (*Figure 10g*; n=3 mice/age-group, $F$=6.821, DFn, DFd = 1, 7, p=0.0348, One-way ANOVA). In addition, expression of Gdnf transcripts show significant reduction between the ages of P30 and P90 (*Figure 10h*; n=6 mice/age-group, P30: 1.070±0.179 SEM, P90: 0.539±0.129 SEM, p=0.037, Student's t-test). On the other hand, HGF expression increases progressively between P10 and P90 ages (*Figure 10i and j*: n=3 mice/age-group, Protein levels: $F$=8.820, DFn, DFd = 1, 7, p=0.02; mRNA levels: $F$=36.98, DFn, DFd = 1, 16, p<0.0001, One-way ANOVA). The ratio of HGF to GDNF expression in ileal LM-MP tissue shows significant increase from the age of P10 to P90 (*Figure 10—figure supplement 1a*, $F$=48.60, DFn, DFd = 1, 7, p=0.0002, One-way ANOVA), and HGF expression is consistently higher than GDNF expression in the full thickness small intestinal tissue from adult and aging mice in the Tabula muris data (*Almanzar et al., 2020*; *Figure 10—figure supplement 1b*). We then queried the plasma proteome levels from the LonGenity cohort of 1,025 aging individuals and found that HGF levels correlated positively, while GDNF and RET levels correlated negatively with age (*Figure 10—figure supplement 1c*; *Sathyan et al., 2020*), suggesting parallels between our data from murine intestine and human plasma proteome data.

Since GDNF tissue levels are correlated with NENs proportions, we hypothesized that GDNF signaling regulates NENs proportions in maturing and adult ENS. On administration of GDNF or saline to cohorts of P10 Wnt1-Cre:*Rosa26*^lsl-tdTomato^ mice over 10 days 87,96, we found that GDNF treatment promoted the juvenile phenotype by enhancing the proportions of tdTomato⁺ NENs and correspondingly reduced the proportion of tdTomato⁻ MENs in P20 mice to a level similar to that seen at the P10 age, while retaining the MENs proportions in saline-treated control mice remained at a level expected of its age (*Figure 10k*; Controls: %MENs: 25.87±4.37 SEM of 1350 neurons from three mice; GDNF: %MENs: 3.86±0.07 SEM of 1301 neurons from three mice; p=0.0072; Student's t-test). Consistent with a previous report *Gianino et al., 2003*, the GDNF-driven expansion of NENs and associated contraction of MENs conserved the total neuronal numbers (*Figure 10k*; Controls: neurons/ganglia = 20.60 ± 4.00 SEM; GDNF: neurons/ganglia = 24.39 ± 6.96 SEM; p=0.52; Student's t-test).

Since HGF tissue levels increase with age, we hypothesized that increasing HGF signaling drives the expansion of MENs in the maturing gut. HGF administration to cohorts of P10 Wnt1-Cre:*Rosa26*^lsl-tdTomato^ mice over 10 days promoted an increase in the tdTomato⁻ MENs population in P20 mice to levels previously observed in P60 mice, while tissues from saline-treated control mice exhibited a MENs:NENs ratio that is expected at P20 (*Figure 10l*; Controls: %MENs: 27.40 ± 5.49 SEM of 1970

neurons from 5 mice; HGF: %MENs: 49.37 ± 5.52 SEM of 1704 neurons from five mice; p=0.02, Student's t-test). Similar to GDNF treatment, HGF treatment also did not cause any significant change in total neuronal numbers (*Figure 10l*; Controls: neurons/ganglia = 20.35 ± 2.22 SEM; HGF: neurons/ganglia = 18.21 ± 0.69 SEM; p=0.38, Student's t-test).

## Transcriptomic evidence of MENs genesis

As the proportions of MENs in the myenteric plexus rise significantly between the ages of P10 and P30, we reasoned that this phase was accompanied by significant neurogenesis of MENs. To provide transcriptomic evidence that MENs populations indeed expand during this phase, using 10 X Genomics v3.1, we again performed unbiased scRNAseq and agnostic clustering, this time on unsorted cells from the myenteric plexus layer of two male mice of P21 age, when the MENs population is still expanding (*Figure 11—figure supplement 1a*). Using NMF-generated MENs patterns to run projectR-based analyses (*Figure 11—figure supplement 1b*) and by using the NENs and Neuroglia-specific markers, we again annotated the neuroglia, the NENs, and the MENs as before and found that at this age, we were able to sequence a similar number of MENs and NENs (~500 cells) along with a large number (~800) of neuroglia cells (*Figure 11a*). The expression of Met was again detected in the MENs cluster, as well as in a small subset of Ret-expressing NENs in the P21 cluster (*Figure 11—figure supplement 1c*). MENs showed significantly higher UMI per cell, when compared to neural crest-derived cells (NENs and neuroglial cells) (*Supplementary file 4*, One-way ANOVA; $F=187.4$, DFn, DFd = 2, 1877, p<0.0001), which we expected given the larger size of MENs (*Nadal-Ribelles et al., 2019*; *Figure 4—figure supplement 1a*). By immunolabeling LM-MP tissue from three P21 Wnt1-Cre:*Rosa26*[lsl-tdTomato] male mice with antibodies against MET, we found that proportions of MET-expressing neurons at this age were significantly higher in the population of tdTomato-negative MENs than in the population of tdTomato⁺ NENs (of 152 MET-immunolabeled neurons across three mice, 81.51±1.55 S.E.M. were tdTomato-negative MENs). The *Met* and *Ret* co-expressing NENs cluster in our data were described as populations of NC-derived *Chat⁺ Calcb⁺* neurons by *Zeisel et al., 2018* at the P21 age (*Figure 11—figure supplement 1d*), where they were annotated as ENT6, 8, and 9 clusters (and which correspond to clusters 1, 3, 4, 7, 6, and 7 of the database). These *Met*- and *Ret*-co-expressing neurons were also described by as fetal-born NC-derived neurons that respond to both GDNF and HGF. By contrast, MENs do not express *Ret* and hence are the only cell population that can respond to solely to HGF. This data provides further evidence on the HGF-induced expansion of MENs during the juvenile phase.

We next performed in silico analyses on the MENs cluster using our recently published Tricycle software (*Zheng et al., 2022*), which is capable of inferring continuous cell-cycle position and can be applied to datasets with multiple cell types, across species and experimental conditions, including sparse data and shallow sequenced droplet-based dataset - thus allowing us to find evidence of cell cycling in the MENs subset. P21 MENs scRNAseq data was projected into universal cell cycle principal components (PCs) defined by the expression of 500 cell-cycle correlated genes and the continuous cell cycle position (theta) was measured as the angle from the origin (*Figure 11b*). Based on expression of 500 cell-cycle correlated genes, cells between 0.5π and 1.5π show hallmarks of being cycling cells (*Figure 11b*). Our analyses showed that a significant number of cells in the MENs cluster (213 out of 510) were present between 0.5 and 1.5π, which represents the mitotic position in the continuous cell-cycle phase (*Figure 11c*). We confirmed our analyses by observing that the expression of the cell cycle gene Top2a in the cells of the MENs cluster was highly correlated with this phase (*Figure 11d*). Next, we found that a key cell cycle regulator gene *Ect2* (epithelial cell transforming 2), which encodes for the guanine nucleotide exchange factor protein ECT2, which activates RhoA in a narrow zone at the cell equator in anaphase during cell division (*Schneid et al., 2021*), was expressed in the MENs cluster and its expression was highly correlated with the cell's mitotic position on the Tricycle plot (*Figure 11e*). We then used antibodies against ECT2 to immunolabel the LM-MP from P21 mice and found expression of this key cell cycle regulator protein in DECORIN-expressing and MHCst-expressing MENs at this age (*Figure 11f and g*). Thus, the Tricycle-based computational analyses found evidence of cell cycling in the MENs cluster and identified a key cell cycle gene *Ect2* as a marker for putative MENs precursor cells.

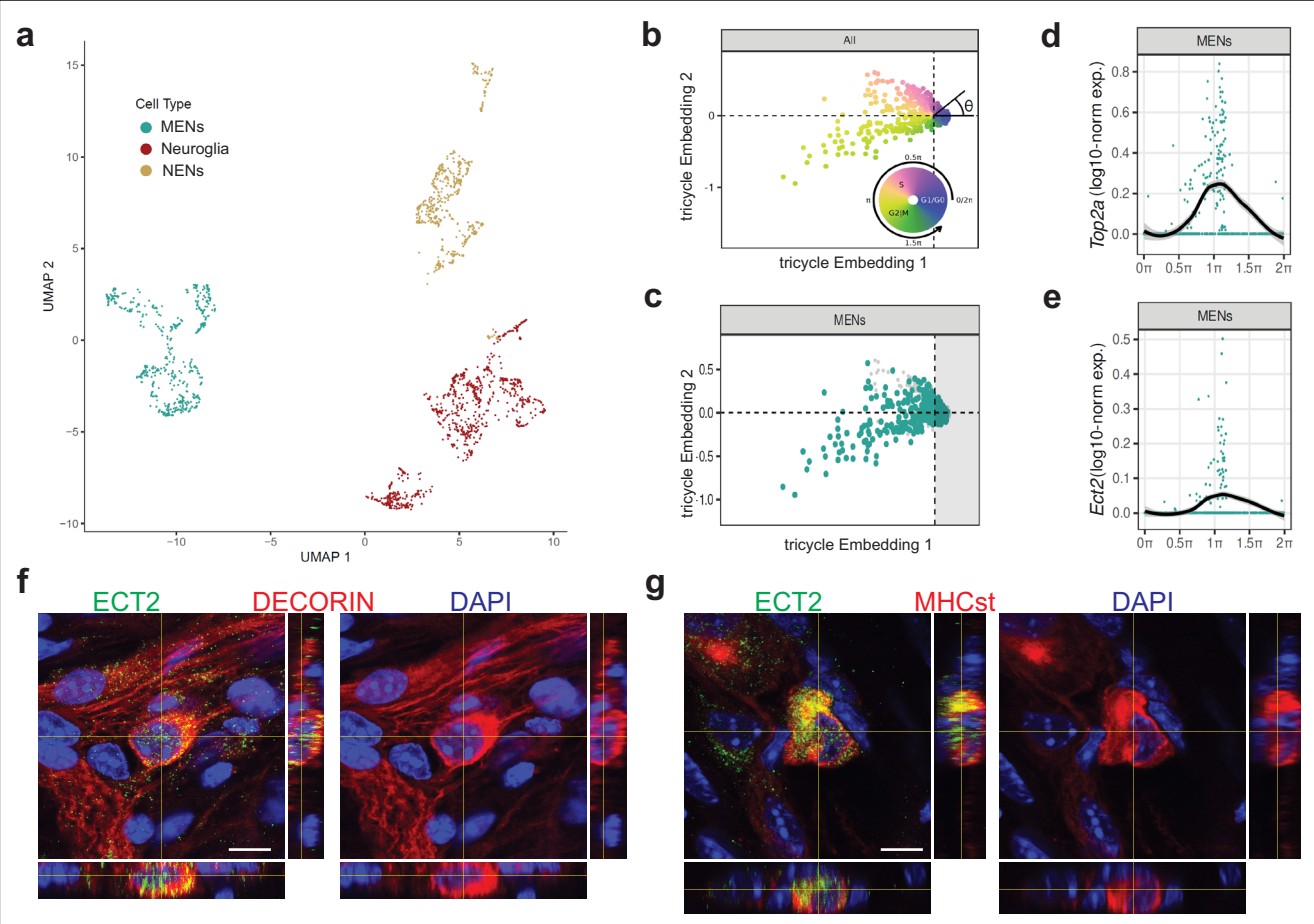

**Figure 11.** *Ect2*-expression labels a population of cycling MENs. (**a**) UMAP representation of the cells of the MENs, NENs, and Neuroglia clusters from scRNAseq of cells from the small intestinal LM-MP tissue of two post-natal day 21 (**P21**) mice. (**b**) Tricycle analyses of ENS cell-types with their cell cycle positions. The continuous cell cycle position (theta) is measured as the angle from the origin. Anti-clockwise representation of cells that are represented between 1.5π and 0.5π space on the embedding are increasingly present within the quiescent G1/G0 cell cycle phase. In contrast, cells that are represented between the 0.5π and 1π space on the embedding are present within the S phase, and those within the 1π and 1.5π space are present within the G2M phase. (**c**) Cells within the P21 MENs cluster were projected into Tricycle software. 213 of the 504 P21 MENs were present between the 0.5π and 1.5π space on the embedding and hence were inferred to be undergoing cell cycling. (**d, e**) Expression of cell-cycle correlated genes *Top2a* and *Ect2* in MENs shows significant expression of these two genes in cells of the P21 MENs cluster that are present between the values of 0.5π and 1.5π in the Tricycle embedding. Loess curve fittings (black) represent the dynamics of gene expression over the cell cycle (theta). Orthogonal views of small intestinal myenteric ganglia from a P21 wildtype mouse when immunostained with antibodies against ECT (green) and against MENs markers (**f**) DECORIN (red) and (**g**) MHCst (red), shows that ECT2 is expressed in a subset of MENs at this age. Nuclei are labeled with DAPI (blue). Scale bar denotes 10 μm.

The online version of this article includes the following figure supplement(s) for figure 11:

**Figure supplement 1.** Identification of the MENs cluster in the P21 scRNAseq data and expression of *Ret* in NENs and in the *Ret* heterozygous mice.

## Reduced GDNF-RET signaling accelerates ENS aging to cause intestinal dysfunction

Since reduced GDNF or RET levels are associated with intestinal dysfunction in patients (*Barrenschee et al., 2015*; *Barrenschee et al., 2017*; *Rossi et al., 2016*), we hypothesized that alterations in GDNF-RET signaling unrelated to those seen with normal aging, would cause dysfunction. To test this hypothesis, we studied lineage proportions and intestinal function in a mouse model of reduced RET signaling. Ret-null heterozygous mice, which have been previously used to study the effect of reduced RET signaling in the adult ENS, have normal ENS structure but altered gut physiology (*Gianino et al., 2003*). A similar mouse model with a *Ret^CFP* allele has a *Cfp* reporter gene inserted at its *Ret* locus rendering it null (*Uesaka et al., 2008*). *Ret^+/CFP* (or *Ret^+/-*) mice carrying a single copy of the *Ret*

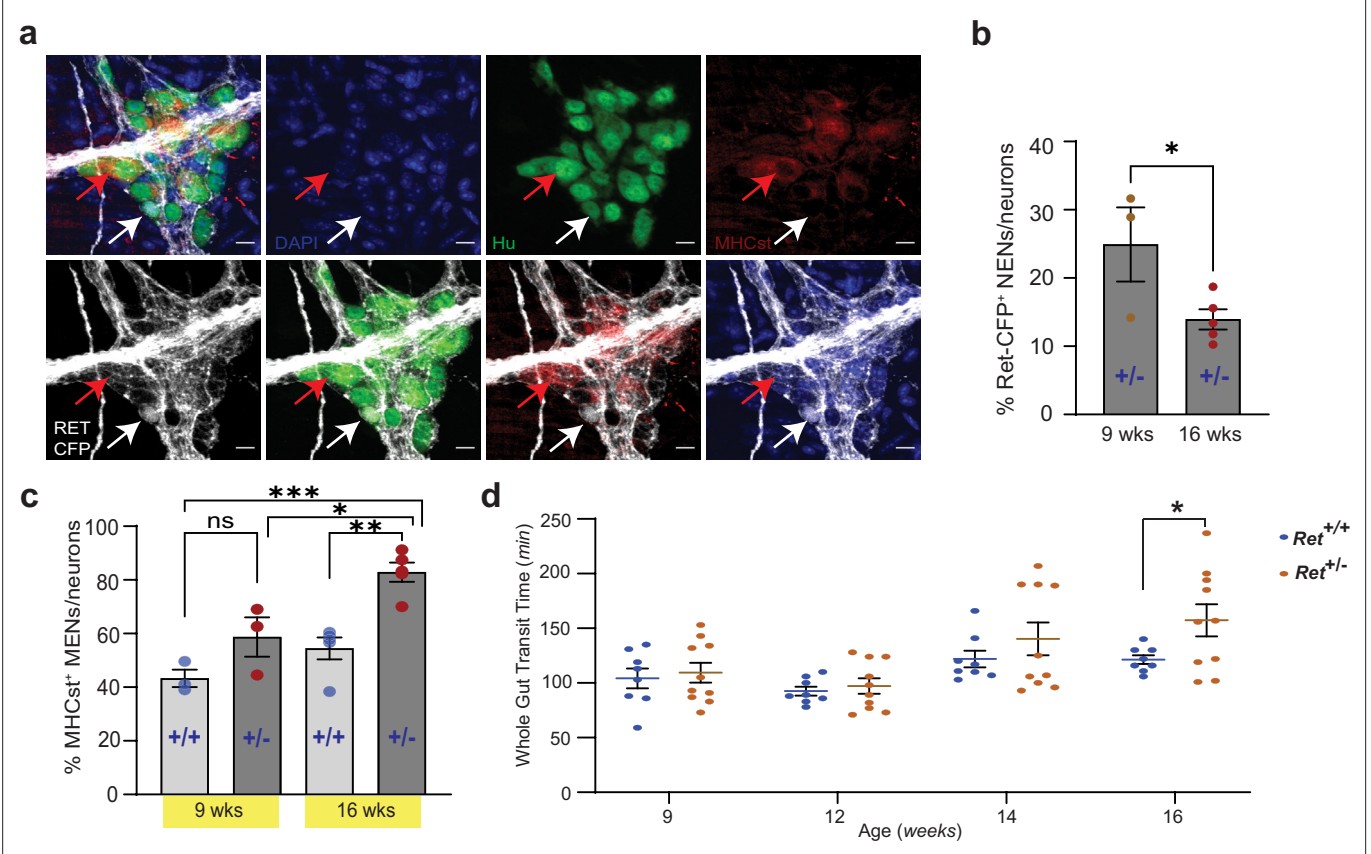

**Figure 12.** Reduced RET signaling accelerates ENS aging to cause pathology. (**a**) Hu immunostaining (green) LM-MP tissues from 16-week-old *Ret⁺/CFP* (*Ret⁺/⁻*) mouse shows mutually exclusive expression of Ret-CFP (cyan, white arrow) and MHCst (red, red arrow) MENs. Nuclei are stained with DAPI (blue). Scale bar = 10 μm. (**b**) Quantification of Ret-CFP⁺ neurons from 9- and 16-week-old *Ret⁺/⁻* +/- show age-associated loss of Ret-CFP⁺ neurons. Data represent mean ± S.E.M. Student's t-test * p<0.05. (**c**) Quantification of MHCst⁺ MENs shows significant increase in their proportions in *Ret⁺/⁻* mice but not in *Ret⁺/⁺* mice with age. Data represent mean ± S.E.M. One-way (ANOVA * p<0.05, ** p<0.01, *** p<0.001). (**d**) Measures of whole gut transit time (WGTT) in cohorts of *Ret⁺/⁻* and *Ret⁺/⁺* mice MENs show significant slowing of whole gut transit of *Ret⁺/⁻* +/- not *Ret⁺/⁺* mice with age. Data represent mean ± S.E.M. One-way ANOVA *=p < 0.05.

gene showed significant reduction in *Ret* transcript and RET protein expression in the early post-natal murine gut (*Uesaka et al., 2008*). Similarly, we found significantly reduced Ret transcript expression in the adult LM-MP of *Ret⁺/⁻* mice compared to age-matched wildtype *Ret⁺/⁺* mice (*Figure 11—figure supplement 1e*). In these mice, using antibodies against CFP/GFP, Hu, and MHCst, we confirmed that the NENs marker Ret-CFP, and the MENs marker MHCst were expressed by different neuronal subpopulations (*Figure 12a*). Using the adult *Ret⁺/⁻* mice, we tested the effect of partial *Ret* loss on ENS lineages at two adult ages: 9 weeks (~P60) and 16 weeks (~P110). Using antibody against CFP/GFP to detect CFP⁺ RET-expressing neurons, we found that *Ret⁺/⁻* mice show a significant reduction in the proportions of Ret-CFP⁺ NENs (*Figure 12b*; 9 weeks: %CFP⁺ neurons: 24.91±5.42 SEM of 837 neurons from three mice; 16 weeks: %CFP⁺ neurons: 13.13±0.98 SEM of 1227 neurons from five mice; p=0.03, Student's t-test). We observed a corresponding significant increase in the proportions of MHCst⁺ MENs with age in *Ret⁺/⁻* mice (*Figure 12c*; 9 weeks: %MENs: 58.74±7.33 SEM of 644 neurons from three mice; 16 weeks: %MENs: 82.84±3.58 SEM of 935 neurons from five mice, One-way ANOVA, p=0.014), while control *Ret⁺/⁺* mice showed no significant age-associated change in the proportions of MENs (*Figure 12c*; 9 weeks: %MENs: 43.27±3.24 SEM of 780 neurons from three mice; 16 weeks: %MENs: 54.48±4.07 SEM of 1276 neurons from five mice, One-way ANOVA, p=0.36), which is consistent with our previous results that MENs proportions in wildtype animals are relatively stable in this time window. The expedited loss of NENs in *Ret⁺/⁻* mice confirms that depletion of endogenous RET signaling in the adult ENS accelerates the aging-associated loss of NENs.

Having previously shown that aging mice have intestinal dysmotility (*Kim et al., 2019*), we tested whether the increased loss of NENs in the *Ret* $^{+/-}$ ENS, concomitant with the expansion of MENs accelerated ENS aging, causes an early onset of aging-associated intestinal dysmotility. We studied whole gut transit time (WGTT) in a cohort (n=8) of adult *Ret* $^{+/-}$ mice and their littermate control (n=10) *Ret*$^{+/+}$ mice over 7 weeks, between 9 and 16 weeks of age. While 9 week adult *Ret* $^{+/-}$ mice were similar to control *Ret*$^{+/+}$ mice, WGTT between the two genotypes diverged with age. Consistent with a prior report (*Gianino et al., 2003*), 16 week old *Ret* $^{+/-}$ mice displayed significantly delayed intestinal transit compared to age-matched control *Ret*$^{+/+}$ mice (*Figure 12d*; WGTT (in min) *Ret*$^{+/+}$: 121.4±4.01 SEM; *Ret*$^{+/-}$: 157.3±14.62 SEM, p=0.048; Student's t-test).

## GDNF reverts aging in the ENS to normalize intestinal motility

Along with others, we have previously shown that aging is associated with slowing of intestinal motility (*Sun et al., 2018*; *Becker et al., 2018*). We hypothesized that this may be a consequence of the replacement of the juvenile NENs population by MENs and therefore GDNF supplementation, by restoring a more balanced MENs:NENs ratio, may prevent age related changes in motility. We studied 17-month-old male mice (at an age where NENs constitute only ~5% of all myenteric neurons; *Figure 10c*) before and after they received 10 days of intraperitoneal injection of GDNF or saline (n=5 for each cohort). While the two cohorts showed no significant difference in their intestinal transit times at baseline (WGTT (in min) Control: 192.8±11.55 SEM; GDNF: 202.4±7.60 SEM, p=0.50, Student's t-test), GDNF treatment caused significant improvement in intestinal transit (*Figure 13a*, WGTT (in min) Control: 175.0±8.89 SEM; GDNF: 101.0±8.91 SEM, p=0.0004, Student's t-test), reaching levels previously observed in healthy mice (*Figure 12d*).

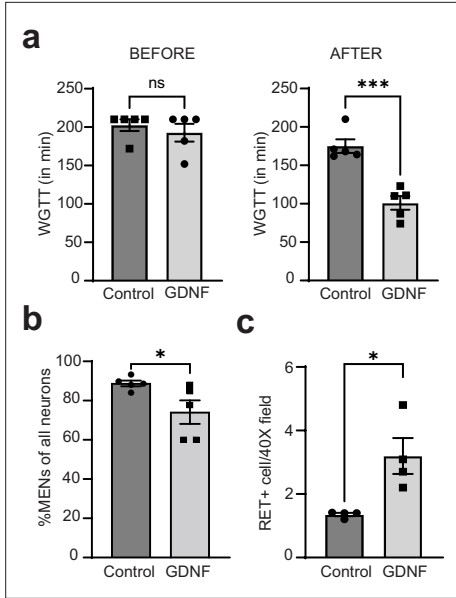

**Figure 13.** GDNF normalizes altered intestinal motility by increasing NENs proportions in the aging gut. (**a**) Measures of whole gut transit time (WGTT) in GDNF (treated with GDNF) and Control (treated with Saline) cohorts of 17-month-old mice taken before the start of treatments and after the end of 10 consecutive days of treatment shows that the two groups are matched in their transit times before treatment, but GDNF treatment significant decreases average transit times when compared to the control cohort. Data represent mean ± S.E.M. Student's t-test ***=p < 0.001. (**b**) Quantification of percent MHCst$^+$ MENs per Hu-labeled neurons in myenteric ganglia in the GDNF and Control cohorts shows significant decrease in their proportions in GDNF-treated cohort but not in saline-treated control cohort. Data represent mean ± S.E.M. Student's t-test ** p<0.01. (**c**) Quantification of numbers of RET$^+$ NENs per 40 X field views of myenteric ganglia shows significant increase in their numbers in GDNF cohort mice when compared with Control cohort mice. Data represent mean ± S.E.M. Student's t-test * p<0.05.

The online version of this article includes the following figure supplement(s) for figure 13:

**Figure supplement 1.** Effect of GDNF treatment on MHCst-expressing MENs and RET-expressing NENs in aging mice.

GDNF treatment significantly reduced the proportions of MHCst$^+$ MENs (*Figure 13b*; *Figure 13—figure supplement 1a, b*; Control: %MENs: 88.76±1.48 SEM of 909 neurons from five mice; GDNF: %MENs: 74.12±5.98 SEM of 799 neurons from five mice; p=0.045; Student's t-test) while increasing the numbers of RET$^+$ NENs (*Figure 13c*; *Figure 13—figure supplement 1c, d*; Control: RET$^+$ neurons: 1.35±0.05 SEM in forty-two 40 X fields from four mice; GDNF: RET$^+$ neurons: 3.19±0.56 SEM in forty-one 40 X fields from four mice, p=0.017; Student's t-test). Consistent with our earlier observation, GDNF treatment did not change the average neuronal numbers in the ileal myenteric ganglia (Control: 17.56±1.82 SEM neurons per ganglia, GDNF: 16.01±1.22 SEM neurons per ganglia, p=0.49, Student's t-test).

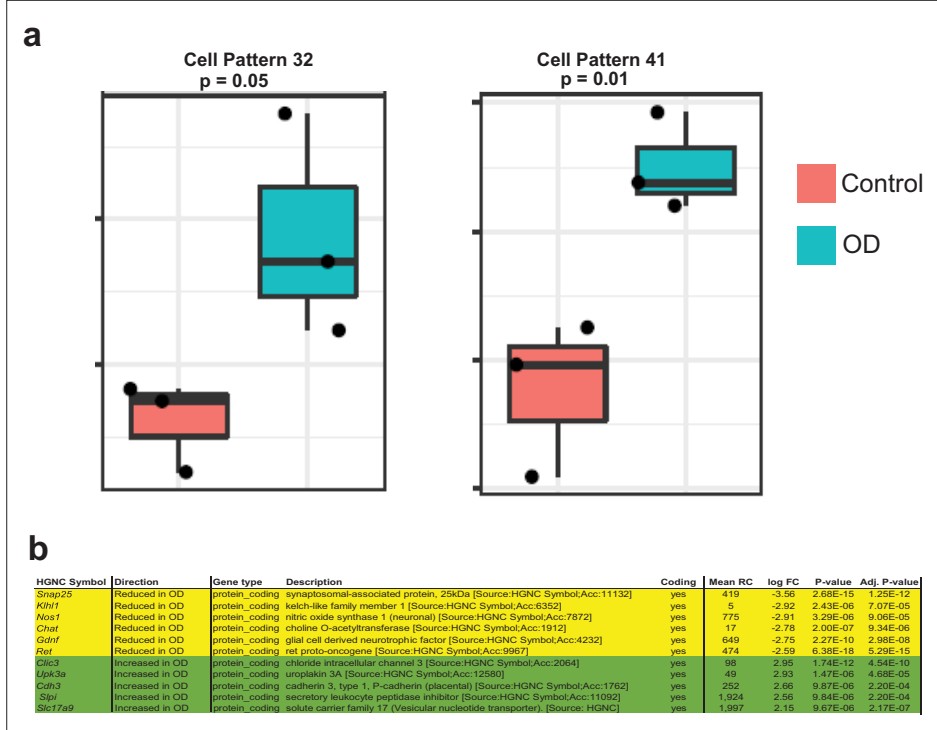

**Figure 14.** Patients with chronic gut dysmotility show significant shifts in their normal proportions of the two neuronal lineages. (**a**) ProjectR-based projection of bulkRNAseq data from intestinal specimens of patients with normal motility and patients with obstructive defecation (OD), a chronic condition of intestinal dysmotility, into the 50 different NMF patterns learnt earlier, shows that the MEN-specific NMF patterns 32 and 41 were significantly upregulated in bulk-RNAseq of OD patients compared to controls. Data represent mean ± S.E.M. One-way ANOVA. Data mined from raw data generated by *Kim et al., 2019* (**b**) OD patients show a significant decrease in the expression of important NENs-associated genes such as *Ret, Gdnf, Snap-25, Nos1, Klhl1,* and *Chat*, while showing a significant increase in the expression of important MENs-specific genes such as *Clic3, Upk3a, Cdh3, Slpi,* and *Slc17a9*. Data taken from *Kim et al., 2019*.

The online version of this article includes the following figure supplement(s) for figure 14:

**Figure supplement 1.** Projection of the bulk RNA sequencing data of intestinal tissue from Control and Patients with Obstructed Defecation into our murine scRNAseq-derived NMF patterns using *projectR*.

## The two neuronal lineages in human disease

In addition to the detection of MENs in the adult healthy gut by bioinformatic and immunofluorescence means (*Figures 4–7*), we next examined the relevance of the two neuronal lineages NENs and MENs in human disease. For this, we tested whether gut dysfunction associated with pathological reductions in NENs-signaling mechanisms are also associated with increased abundance of MENs. We mined previously generated and publicly available transcriptomic data from intestinal tissues from humans with normal gut motility (controls) and from patients with obstructed defecation (OD) associated with enteric neuropathy (*Kim et al., 2019*). In this dataset, Kim et al. have previously shown that OD patients have significantly reduced intestinal expression of *Gdnf* and *Ret* (*Kim et al., 2019*). Again, using our transfer learning tool *ProjectR* (*Stein-O'Brien et al., 2019*; *Sharma et al., 2020*), which allowed us to query bulk-RNAseq data across species to estimate relative use of MENs-specific NMF-patterns generated earlier, we tested whether the usage of MENs-specific NMF-patterns was significantly altered between transcriptomes of OD patients and healthy humans (n=3/group) (*Kim et al., 2019*). This projection analysis indicated that two of the four MENs-specific NMF-patterns showed significantly higher usage in OD samples compared to controls (*Figure 14a*, *Figure 14— figure supplement 1*; Student's t-test, p<0.05), providing evidence that the enteric neuropathy in these patients is associated with a relative increase in MENs-specific transcriptional signatures. We further queried the differential gene expression analysis performed by Kim et al and found that in addition to *Gdnf* and *Ret* (as reported by them) other important NEN genes (*Snap25* and *Nos1*) were

also significantly downregulated while the MEN marker genes identified in this study (*Clic3*, *Cdh3*, *Slc17a9*) (*Figure 14b*) were significantly upregulated.

## Discussion

Current dogma states that the adult ENS is exclusively derived from NC precursors that populate the gut during fetal development *Young and Newgreen, 2001*. The results of this study indicate a much more complex system, one in which the fetal and early life 'juvenile' ENS consisting of NENs is incrementally replaced during maturation by (MENs). Eventually, the aging ENS consists almost exclusively of the neurons of the MEN lineage. Using a combination of floxed transgenic reporters bred into multiple lineage fate mapping transgenic mice, this study also provides the first definitive evidence of a significant mesodermal contribution to any neural tissue. Previously, the mesoderm and especially the Lateral Plate Mesoderm (LPM) was known to give rise to diverse cell-types within the viscera, including several in the gut (*Prummel et al., 2020*), and our study shows that cells of this embryonic derivation also give rise to ENS neurons. A previous report on a dual origin of the ENS described adult enteric neurons from *Foxa2⁺* or *Sox17⁺* precursors (*Brokhman et al., 2019*) and inferred that these were endodermal in origin. However, *Foxa2* and *Sox17* are also expressed by mesoderm-derived cells (*Bardot et al., 2017*; *Burtscher et al., 2012*; *Wilm et al., 2005*; *Kawaguchi et al., 2007*). By contrast, using two NC-lineage-traced mouse lines (Wnt1-Cre and *Pax3^Cre*), two lineage-traced mouse lines marking mesodermal derivatives (Tek-Cre and *Mesp1^Cre*), and robust adult mesoderm markers, we identify a population of *Mesp1*-derived MHCst- and MET-expressing adult enteric neurons, which makes up the entire non-NC-derived population of myenteric neurons. These results provide robust evidence that the second source of adult enteric neurons is the mesoderm, and not the endoderm.

### Distinct neuronal nature of MENs

While the developmental origins of neurons have been thought to be restricted to the neural tube and the NC, many neurons – such as placode-derived neurons, have been known to be derived from non-neural tube, non-NC developmental lineages (*Carlson, 2014*). Similarly, while supposed pan-neuronal markers, such as *Snap25*, are expressed specifically by neurons, not all neurons express *Snap25* (*Verderio et al., 2004*). Indeed, release of neurotransmitters may occur independently of *Snap25* (*Nouvian et al., 2011*; *Grabs et al., 1996*; *Hellström et al., 1999*; *Morris et al., 2000*; *Greenlee et al., 2001*; *Gibbins et al., 2003*), suggesting that the expression of this protein is not central to neuronal functions. These present us with important context based on which, one can study the data generated using multiple lines of evidence to infer the neuronal nature of MENs. *Firstly*, we show that MENs are intra-ganglionic cells, which conditionally express reporters under mesoderm-specific lineage fate mapping models and are labeled with antibodies against the neuronal marker Hu. *Secondly*, MENs show expression of important ENS neurotransmitter-encoding or generating proteins CGRP, NOS1, and ChAT. *Thirdly*, with our single-cell transcriptomic analysis, where we used the markers *Calcb*, *Met*, and *Cdh3* that we had validated to identify and annotate the MEN cell cluster, we discovered an additional set of neuronal-specific marker genes (*Pde10a*, *Vsnl1*, *Stmn2*, *Stx3*, *Tubb2b*, *Slc17a9*, *Hand2*, *Gpr88*, *Elavl2*, *Ntf3*) which were found to be expressed widely or selectively within MENs. These markers include both novel as well as established enteric neuronal markers (*Jarret et al., 2020*; *De Giorgio et al., 2000*; *Xue et al., 2016*). *Fourthly*, further transcriptomic analyses of the MENs scRNAseq cluster identifies the expression of many other neuronally significant genes – including those encoding for ion channels, neurotransmitter receptors, hormones, calcium channels, etc. – which provide a further glimpse into the diverse neuronal functions of MENs. *Finally*, our analyses shows that MENs express the pan-ENS-expressed gene *Phox2b*, and that our MENs scRNAseq cluster maps to a neuronal cluster in the *May-Zhang et al., 2021* database, which we shows expression of 'mesenchymal' and 'neuronal' markers (*May-Zhang et al., 2021*). Our data, together with our analyses of publicly available data, thus provides evidence on the neuronal nature of MENs. While the co-expression of aforementioned pan-neuronal markers establishes the neuronal nature of MENs, the expression of set of key genes in MENs suggests putative specialized functional roles of these subpopulations when they co-exist with NENs in adults. These include the expression of genes associated with putative sensory (*Calcb*) (*Wang et al., 2021a*), neuromodulatory (*Ntf3*) (*Tender et al., 2011*), secretory (*Cftr*) (*de Jonge et al., 2020*), immunomodulatory (*Il18*) (*Jarret et al., 2020*),

hormone regulating (*Ghrh*) (*de Lima et al., 2021*), extracellular matrix (ECM)-modulating (*Dcn*), and motility-regulating roles (*Slc17a9* or VNUT) (*Chaudhury et al., 2016*). These MEN-enriched or MEN-specific expression profiles suggest that MENs perform various roles in the regulation of diverse gastrointestinal functions. While the current scRNAseq data is restricted in providing meaningful cell clusters that can inform the true functional diversity of MENs, future work will provide detailed analyses and functional characterization of MENs.

Beyond establishing the neuronal nature of MENs, our analyses also help establish the distinct neuronal nature of MENs. Based on the lineage fate mapping experiments with multiple combinations of NC and mesoderm lineage fate mapping mice models and based on the expression of mesoderm-specific markers such as MHCst and DCN (*Decorin*), we identified that MENs are a population of mesoderm-derived cells. MHCst is a subset of myosin heavy chain isoforms that has been previously shown to be specifically expressed by a subset of mesoderm-derived muscle cells (*Daugherty et al., 2012*). While prior reports do not clarify the nature of the exact epitope and the gene responsible for MHCst, the expression of the smooth muscle myosin heavy chain protein MYH11 specifically in MENs suggests that the epitope detected by the anti-MHCst antibody in MENs is of the mesoderm specific protein MYH11 protein.

Further, our bioinformatic analyses of the human gut cell atlas data generated by *Elmentaite et al., 2021* also shows the existence of a putative MENs-like cell cluster in the human gut, which shows a similar co-expression of mesenchymal and neuronal markers as observed by us in our MENs cluster and by *May-Zhang et al., 2021* in their 'mesenchymal neuronal' cluster. Indeed, *Elmentaite et al., 2021* identified this supposed mesenchymal cluster by the expression of the mesoderm-specific gene *Decorin*. Further, by observing the co-expression of neuronal and mesoderm-specific markers in human gut tissue, we validate the presence of neurons that express mesoderm-specific markers or MENs in adult human tissue.

## Detection and identification of MENs in other datasets

Recent studies have significantly increased our understanding of the molecular nature of various classes of enteric neurons. However, these studies differ significantly in their methods and tools used, which have precluded them from detecting or correctly identifying the developmental origins of MENs. While *May-Zhang et al., 2021* observed the presence of a small neuronal cluster that expressed mesenchymal marker genes, their study did not further interrogate the developmental origins of the 'mesenchymal' enteric neurons. *Drokhlyansky et al., 2020* in their unbiased MIRACL-seq analyses and *Elmentaite et al., 2021* in their human gut cell atlas datasets detected the MENs cluster but could not correctly identify their neuronal nature, possibly owing to the expression of mesenchymal genes in this cluster, and also due to the absent or lower than expected expression of canonical neural markers such as *Ret* and *Snap25*. In contrast to these studies that performed single cell or nuclear transcriptomic studies that were unbiased by lineage markers, other transcriptomic datasets from *Drokhlyansky et al., 2020*, *Wright et al., 2021*, and *Zeisel et al., 2018* were generated using enriched NC-derived cells, which contributed to the absence of MENs in these datasets. An important outlier to these studies is *Morarach et al., 2021* from the Marklund group, which utilized the newly characterized BAC-transgenic line Baf53b-Cre to label and flow sort enteric neuronal cells for subsequent single-cell RNA sequencing. Our analysis of ileal and colonic tissue from this line shows that it does not comprehensively label all enteric neurons. While this may seem at odds from the *Morarach et al., 2021* report, this apparent difference in the behavior of this transgenic line can be explained by the fact that in contrast to the established and accepted practice of counting all Hu-immunolabeled neurons employed by us and others (*Wright et al., 2021*; *May-Zhang et al., 2021*; *Parathan et al., 2020*), the Morarach et al. study restricted the enumeration to only include neurons that showed only cytoplasmic and not nuclear expression of Hu. Neuronal Hu proteins (HuB, HuC, and HuD) show both nuclear and cytoplasmic localization (*Zhu et al., 2006*), and since the *Morarach et al., 2021* study only enumerated cytoplasmic Hu-expressing neurons, their characterization of this transgenic mouse line was performed using only a subset of all neurons. In addition, compared to *Drokhlyansky et al., 2020*, *Wright et al., 2021*, and *May-Zhang et al., 2021* whose protocols for neuronal enrichment were based on flow sorting neuronal nuclei, the Marklund group (in both *Zeisel et al., 2018* and *Morarach et al., 2021*) performed flow sorting of late post-natal enteric neuronal cells, which can cause significant stress on neuronal cells. These factors, together with the BAC transgenic nature of

the Baf53b-Cre line, may have contributed to the lack of MENs in the *Morarach et al., 2021* data. Further, all neuronal cells described in the *Morarach et al., 2021* study express *Snap25* (which we show is not expressed by all adult enteric neurons), and mostly map to known NC-derived cell clusters identified in *Zeisel et al., 2018*. This suggests that the rich transcriptomic atlas of enteric neurons generated by deep transcriptomic profiling of Baf53b-Cre-derived neurons carried out by *Morarach et al., 2021* is restricted only to the NENs, which comprise of half of all adult enteric neurons. Thus, prior experimental data could not establish the true developmental identity, correct annotation, and transcriptomic definition of a population of mesoderm-derived neurons or MENs in murine and human tissue.

By contrast, we were able to correctly identify the MENs cluster in unbiased scRNAseq of adult murine LM-MP cells and use it to both identify and validate the expression of MENs-specific markers in the adult murine LM-MP tissue, as well as use bioinformatic analyses to identify the MENs in publicly available murine data from *May-Zhang et al., 2021* and human data from *Elmentaite et al., 2021*. This allowed us to show the expression of MENs markers MHCst, MET, DCN, and SLC17A9 in many enteric neurons in adult humans, suggesting that the mesoderm-derived ENS may be a feature common to mice and humans alike. MENs in both species can readily be identified by their expression of MENs-specific markers, thus providing a convenient tool to further identify and study these neurons.

## Development of MENs and MENgenesis

The presence of small population of MENs in the gut at the post-natal age day 10 (P10) and their expansion during maturation suggests that key timepoints in their development correspond with the post-natal development of the ENS. Importantly, since MENs populations are expanding at this age, performing scRNAseq-based bioinformatic analyses of the juvenile LM-MP cells not only provided us with the computational evidence of significant cell cycling in the MENs cluster at this age, but also provided us with the information on *Ect2*, a cell-cycle marker expressed by cycling putative MENs precursor cells. *Ect2* transcripts were found expressed by cycling cells within the MENs cluster at the P21 age, which also showed significant enrichment in the expression of the cell-cycle gene *Top2a*. Except for a cluster of cells annotated as proliferating glial cells that show expression of the cell cycle gene *Top2a, Ect2* gene expression is not detected in other murine myenteric NC-derived cells in the scRNAseq study performed by *Zeisel et al., 2018* at the P21 juvenile age. Our anatomical observation on the expansion of the MEN population in the adolescent gut along with the computational evidence of cell cycling in the MENs cluster at this age, and the detection and validation of *Ect2* gene expression in a subset of MENs provide evidence of significant MENgenesis during the juvenile phase. ECT2 expression was found in MHCst- and DCN-expressing myenteric cells at this age, suggesting that many of these are proliferating MEN-precursor cells, and may not be terminally differentiated functional neurons.

The rapidly expanding MEN population is 10–11 days old in the P21 post-natal gut (as they originate in the gut just before P10 post-natal age) and thus, the age at which these cells partake in regulation of diverse gastrointestinal functions is yet unknown. This important issue precludes us from using the two MENs transcriptomic datasets to compare how MENs functions change with age. Prior studies have reported on the differing birthdates of various NEN sub-populations in the developing fetal gut (*Bergner et al., 2014*; *Morarach et al., 2021*; *Epstein et al., 1992*). Similar studies need to be performed in the future to understand the birthdates of the various subpopulations that comprise MENs, and to ascertain when do they become functional. A recent study by *Parathan et al., 2020* highlighted important neuronal and neurochemical changes in the adolescent gut. Further studies on how the ENS matures and ages thus need to be performed in the context of NENs and MENs to truly understand how shared and distinct functions of these two lineages change with maturation and age. A crucial aspect of this work will require further identifying the molecular markers for the *Ect2*-expressing proliferating MEN progenitor cells. Our earlier work showed that *Nestin*-expressing neural progenitor cells are NC-derived cells that contribute to steady state neurogenesis in the adult ENS (*Kulkarni et al., 2017*), thus suggesting that MENs are derived from *Ect2*-expressing non-*Nestin*[+] precursor cells.

## Use of scRNAseq to identify MENs

The use of single cell and single nucleus transcriptomic approaches provides an important tool that has been recently used to perform molecular taxonomy of fetal and post-natal enteric neurons and glial cells (*Wright et al., 2021*; *Zeisel et al., 2018*; *May-Zhang et al., 2021*; *Morarach et al., 2021*). In addition to performing deep sequencing that allows for cluster-based molecular taxonomy, these tools can also be utilized for identification of novel cell types, where prior information on their presence and marker expression is unknown (*Pollen et al., 2014*; *Zheng et al., 2022*; *Zanini et al., 2020*; *Wang et al., 2021b*; *Zhang et al., 2022*). Compared to the other studies that utilized this tool to study the ENS, we used the single-cell transcriptomics to both confirm the presence of MENs and identify more MEN-specific markers as well as to use the MEN-specific transcriptomic signature to query their presence in other murine and human datasets. In contrast to its use for performing molecular taxonomy, which requires deep sequencing of an enriched cell population, our strategy of opting for a lower read depth (shallow sequencing) is efficient, practical, and economical for discovering and identifying novel celltypes (*Pollen et al., 2014*). In addition, even if a gene is being expressed at a medium - low level, there is a certain probability that it will not be detected by scRNAseq in both deep and shallow sequencing approaches. This is important since the older v2.0 10 X Genomics chemistry (used for our 6-month dataset) captures only 10–14% of the cellular transcriptome, and that from the newer v3.1 chemistry (for our P21 data) captures only a third of the transcriptome from a suspension of homogenous single cells (*Zheng et al., 2017*; *Genomics, 2023*). While a partial solution to this issue is to increase sequencing depth, beyond a certain point, this strategy leads to diminishing returns as the fraction of PCR duplicates increases with deeper sequencing. While getting insights into cellular heterogeneity of MENs is important for furthering our understanding of how they regulate diverse gut functions, the transgenic mice that can specifically label only MENs or all enteric neurons to allow for their enrichment are currently not available (*Drokhlyansky et al., 2020*; *Zeisel et al., 2018*; *Wright et al., 2021*; *Morarach et al., 2021*; *Corpening et al., 2008*; *May-Zhang et al., 2021*). Without the availability of MENs-specific transgenic models, the scRNAseq strategy employed by us is the only currently available tool for querying the transcriptome of MENs. These reflect significant bottlenecks that have disallowed the comprehensive and an in-depth characterization of all enteric neurons in all the current studies on the adult ENS.

## Expression of MET and RET by MENs and NENs

Since our scRNAseq data highlighted the lineage-specific nature of the expression of *Ret* and *Met* in the adult ENS, we studied whether these genes regulated the origin, expansion, and maintenance of the neuronal populations that express them. The expression of MET by MENs in the mature adult ENS is consistent with an earlier study by Avetisyan et al. that reported that a deletion of *Met* in the Wnt1-Cre expressing NC-lineage did not alter the abundance of MET+ neurons in the adult ENS (*Avetisyan et al., 2015*). If *Met* was expressed solely by NC-derived cells, then the loss of MET by enteric neurons would have been expected since the conditional deletion of *Met* gene in this floxed *Met* transgenic animal has been observed to cause a significant or comprehensive loss of MET protein expression in other organs (*Song et al., 2015*; *Dai et al., 2005*; *Ishibe et al., 2009*; *Arechederra et al., 2013*). While at first glance, observations of the mutually exclusive nature of RET and MET provided by us and by *Avetisyan et al., 2015* in the adult gut might not agree with *Zeisel et al., 2018* observed co-expression of *Ret* and *Met* transcripts in NENs at the P21 developmental stage, our observation that a small proportion of NENs in the P21 gut express MET helps provide an explanation. Our P21 data and the data from *Zeisel et al., 2018* are consistent with one of the observations made by *Avetisyan et al., 2015*, who showed the existence of a population of fetal NENs that are co-dependent on GDNF and HGF and hence would express both RET and MET. In contrast to these observations made in the fetal and juvenile ENS, the anatomical observations made by *Avetisyan et al., 2015* in the adult ENS (where MET immunoreactivity in the adult enteric neurons is exclusive of *Ret* expression) suggests that the population of fetal-derived Ret and Met co-expressing neurons is lost after maturation. Compared to the P21 scRNAseq data from *Zeisel et al., 2018* that shows that all *Met*-expressing NC-derived neurons co-express *Calcb* (CGRP), *Ret*, and *Chat*, Avetisyan et al.'s anatomical data in the adult gut shows that despite accounting for a third of all enteric neurons and expressing CGRP, MET-expressing adult enteric neurons do not co-express *Ret* and only a small proportion of these MET+ neurons co-express ChAT. Thus, Avetisyan et al.'s data agrees with our data

on the adult ENS which shows exclusive expression of RET and MET by NENs and MENs, respectively, and that while MENs consist of a large proportion of CGRP-expressing MET[+] neurons, they contain only a small population of ChAT-expressing neurons. Along with Avetisyan et al.'s data, our data suggests that the adult small intestinal myenteric plexus contains a significant proportion of neurons that do not immunolabel with antibodies against NOS1 or ChAT. This observation is consistent with earlier data from *Parathan et al., 2020* who showed that ~35–40% of small intestinal myenteric neurons from 6-week-old mice did not immunolabel with antibodies against NOS1 or ChAT.

Since they do not express the NC-marker RET, MENs are the only neuronal population that respond to HGF alone. In this study, we show that MENs are dependent on HGF-MET signaling in a manner analogous to the requirement of GDNF-RET signaling by NENs. While our study does not discount that the exogenous HGF supplementation in juvenile mice may alter proportions of RET and MET-co-expressing neurons within the NENs subpopulation, our report focuses on HGF's ability to drive the expansion of MENs population in the ENS. Similarly, *Ret* haploinsufficiency-mediated loss of the NEN lineage causes a proportional increase in the MEN lineage, while maintaining overall neuronal numbers. Conservation of neuronal number, despite significant shifts in the proportions of the two neuronal lineages suggests the presence of yet unknown quorum sensing signaling mechanisms between the two lineages that reciprocally regulate their populations to maintain the structure of the post-natal ENS. This again implies the existence of a yet undefined precursor cell responsible for the expansion of the MEN population in a manner analogous to what we have previously described for NENs (*Kulkarni et al., 2017*).

## Relevance to aging and human disease

While it is known that *Gdnf* expression in the mature gut is significantly downregulated (*Avetisyan et al., 2015*; *Gianino et al., 2003*; *Wang et al., 2010*; *Golden et al., 1999*), the functional consequences of this loss have been unclear (*Gianino et al., 2003*). We found that reduced GDNF-RET signaling drives the age-dependent loss of NENs as this loss can be slowed or reversed by GDNF supplementation and accelerated by further reduction of *Ret* expression. In aging animals, GDNF-driven resuscitation of NENs was associated with a functional normalization of age-associated delay in intestinal transit. Our results identify a novel role for GDNF in maintaining or resuscitating the canonical NEN population, while preserving overall enteric neuronal numbers. In the last few years, studies have focused on identifying juvenile protective factors (JPFs), the loss of which correlates with or drives maturation- and aging-associated phenotypes (*Papaconstantinou et al., 2015*). In this context, GDNF may therefore qualify as a JPF or a *senolytic* as its presence maintains the dominance of NENs in maturing ENS, corresponding to a juvenile phenotype; and its re-introduction promotes and resuscitates the genesis of NENs in adult and aging gut to restore healthy gut function. The exact nature of the cells that respond to GDNF re-introduction and generate NENs is yet unknown, but it can be hypothesized that these may include *Nestin*[+] enteric neural stem cells and/or GDNF-responsive Schwann cells (*Kulkarni et al., 2017*; *Soret et al., 2020*). In addition to its effect on resuscitating NENs, acute administration of GDNF also augments peristaltic motility by increasing neuronal activity (*Wright et al., 2021*; *Grider et al., 2010*). Since the aging gut has a very small number of RET-expressing GDNF-responsive NENs, we assume that GDNF-driven normalization of intestinal motility would first increase the proportions of RET-expressing NENs and then augment their activation. This further emphasizes the importance of using GDNF or similar RET agonists in the treatment of age-associated intestinal dysmotility.

While Avetisyan et al. showed that MET expression highly correlated with CGRP-expression in the adult murine myenteric plexus (*Avetisyan et al., 2015*), MET co-expression by all MHCst[+] human enteric neurons in our data and by all PGP9.5[+] neurons in human enteric neurons by *Avetisyan et al., 2015* suggests that MET-expression in the human ENS may not be restricted to CGRP-expressing IPANs. In addition, recent data shows that our understanding of the nature of markers that can correctly identify IPANs in the murine and human ENS is still evolving (*May-Zhang et al., 2021*; *de Melo et al., 2020*), suggesting that ascribing function to neurons using a single marker may be incorrect.

The detection of MENs cluster in the post-natal and adult scRNAseq data in the gut cell atlas generated by *Elmentaite et al., 2021*, their increasing representation with age, and the subsequent validation of the MEN-specific markers MHCst, MET, and SLC17A9 in a subset of adult human ENS neurons shows that our findings may be of clinical importance. Many gastrointestinal motility functions are

compromised with advancing age, resulting in clinically significant disorders (*Saffrey, 2014*). Although the exact mechanisms will need to be worked out, our results indicate that a MENs-predominant ENS is associated with significant differences in gut motility. The progressive nature of the change in enteric neuronal lineages with age may have pathological implications for the elderly gut when the balance is overwhelmingly in favor of MENs. Understanding the forces that regulate parity between these two different sources of neurogenesis therefore holds the promise of arresting or reverting progressive loss of gut motility with increasing age. Our results may also have implications for the pathogenesis of disordered motility unrelated to aging, as downregulation of *Gdnf* and *Ret* expression has been associated with diverticular disease and obstructed defecation in adult patients (*Barrenschee et al., 2017*; *Kim et al., 2019*; *Cossais et al., 2019*). GWAS analyses further showed that *Gdnf* gene was in the eQTL associated with increased incidence of diverticular disease (*Schafmayer et al., 2019*). It is in this context that the identification of SNAP-25 as a NENs marker gene and the transcriptomic pattern analyses of patients with chronic obstructed defecation are significant. Expression of *Snap25* is upregulated by GDNF and is significantly downregulated in gut tissues of patients with diseases associated with significant reduction in GDNF-RET signaling (diverticular disease and obstructed defecation *Barrenschee et al., 2015*; *Barrenschee et al., 2017*; *Kim et al., 2019*). Thus, while earlier thought to be a pan-neuronal (*Drokhlyansky et al., 2020*; *Muller et al., 2020*), establishing the identity of SNAP-25 as a NEN lineage-restricted marker provides us not only with an important tool to query proportions of NENs in murine and human tissue but also suggests the NENs-limited nature of prior observations based on *Snap25* transgenic animals (*Matheis et al., 2020*; *Muller et al., 2020*). Gut dysmotility disorders that present with conserved ENS neuronal numbers *Barrenschee et al., 2017* have puzzled investigators on the etiology of the disease. Our results suggest an explanation based on NENs-to-MENs lineage shifts that conserve overall enteric neuronal numbers with variable effects on function.

In conclusion, using multiple lines of evidence, we show that the ENS of the juvenile and mature mammalian gut is dominated by two different classes of neurons that have distinct ontogenies and trophic factors, associated with functional differences. The shift in neuronal lineage may represent a functional adaptation to changes in nutrient handling, microbiota or other physiological factors associated with different ages. Further research in the functional differences between the neuronal lineages and factors that regulate the parity between these two nervous systems in humans during the lifetime will be important to advance our understanding of the adult ENS and the treatment of age-related and other pathological disorders of gut motility.

## Methods

### Animals

Experimental protocols (Protocol number M021M105) were approved by The Johns Hopkins University's Animal Care and Use Committee in accordance with the guidelines provided by the National Institutes of Health. Presence of vaginal plug was ascertained as 0.5 days post-fertilization and this metric was used to calculate age of mice. Only male mice were used for the studies detailed in this report. The Wnt1-Cre:*Rosa26*$^{lsl-tdTomato}$ lineage-traced line was generated as detailed before by breeding the B6.Cg-Tg(Wnt1-Cre) with the Ai14 transgenic mouse line (Jax #: 007914) containing the *Rosa26*$^{lsl-tdTomato}$ transgene (*Becker et al., 2012*; *Kulkarni et al., 2017*; *Becker et al., 2013*). The Wnt1-Cre line used in this study is the well validated line for studying NC derivatives previously used by us and others (*Becker et al., 2012*; *Kulkarni et al., 2017*; *Wright et al., 2021*; *Zeisel et al., 2018*; *Becker et al., 2013*), and not the Wnt1-Cre2 line used by *Drokhlyansky et al., 2020*. We again validated the fidelity of this line by observing whether any aberrant recombination occurred in our Wnt1-Cre:*Rosa26*$^{lsl-tdTomato}$ line by studying the intestinal mucosal tissue and found that the line behaved expectedly to label NC-derivatives and epithelial cells (Suppl. Fig 1i). *Pax3*$^{Cre}$:*Rosa26*$^{lsl-tdTomato}$ lineage-traced line was generated by breeding the Ai9 transgenic mouse line (Jax #: 007909) with the *Pax3*$^{Cre}$ knock-in mouse (Jax #: 005549). The Wnt1-Cre:*Hprt*$^{lsl-tdTomato}$ mouse was generated by breeding our aforementioned Wnt1-Cre transgenic mouse line with the *Hprt*$^{lsl-tdTomato}$ transgenic mouse line (Jax #: 021428, kind gift from Prof. Jeremy Nathans). *Mesp1*$^{Cre}$:*Rosa26*$^{lsl-tdTomato}$ mice were generated by breeding the *Mesp1*$^{Cre}$ mice (*Shenje et al., 2014*) (gift from Dr. Chulan Kwon, JHU) with the Ai14 transgenic mice. *Mesp1*$^{Cre}$:*Rosa26*$^{lsl-EGFP}$ mice were generated by breeding the *Mesp1*$^{Cre}$ mice with the

*Rosa26*<sup></sup>*^lsl-EGFP* transgenic mouse line (gift from Dr. Akira Sawa, JHU). Neither the *Rosa26^lsl-tdTomato* (Ai14) line nor the *Rosa26^lsl-EGFP* line showed any reporter expression in the absence of a Cre driver (Suppl. Fig 1j, k). *Ret^+/CFP* mice (MGI:3777556) were inter-bred to get a colony of adult *Ret^+/+* and *Ret^+/CFP* mice. *Ret^CFP/CFP* mice died at or before term. Tek-Cre:*Hprt^lsl-tdTomato* mice were generated by breeding Tek-Cre transgenic mice (also known as Tie2-Cre; Jax #: 004128) with the *Hprt^lsl-tdTomato* transgenic mouse line. Seventeen-month-old male C57BL/6 mice from the aging mouse colony of the National Institute of Aging were procured for the GDNF-treatment experiment.

## Human tissues

Human tissues were obtained under IRB protocol IRB00181108 that was approved by Institutional Review Board at the Johns Hopkins University. Pathological normal specimens of human duodenum and colon were obtained post-resection. Tissues were obtained from adult donors and were de-identified such that the exact age, gender, and ethnicity of the donors was unknown.

## Tissue preparation

Mice were anesthetized with isoflurane and sacrificed by cervical dislocation. A laparotomy was performed, and the ileum was removed and lavaged with PBS containing penicillin-streptomycin (PS; Invitrogen), then cut into 1-cm-long segments and placed over a sterile plastic rod. A superficial longitudinal incision was made along the serosal surface and the LM-MP was peeled off from the underlying tissue using a wet sterile cotton swab *Kulkarni et al., 2017* and placed in Opti-MEM medium (Invitrogen) containing Pen-Strep (Invitrogen). The tissue was then laid flat and fixed within 30 min of isolation with freshly prepared ice cold 4% paraformaldehyde (PFA) solution for 4-5 min in the dark to preserve fluorescence intensity and prevent photo-bleaching. All LM-MP tissues post-isolation were fixed within 30 min of their isolation. After the fixation, the tissue was removed and stored in ice cold sterile PBS with Pen-Strep for immunofluorescence staining and subsequent microscopy.

For human tissues, duodenal tissue from adult human patients (n=3 patients), who did not have any prior history of chronic intestinal dysmotility, that was removed by Whipple procedure was obtained. A colonic sample from a pathologically normal colonic resection from an adult donor suffering from colon carcinoma who similarly did not have prior history of chronic intestinal dysmotility was also obtained. The resected tissue was placed in ice cold Opti-MEM medium (Invitrogen) containing Pen-Strep (Invitrogen). The mucosal and sub-mucosal tissue was dissected out in the medium under light microscope and the muscularis layer containing myenteric plexus tissue was obtained. The tissue was laid out between two glass slides and fixed overnight in ice cold 4% PFA after which it was removed and stored in ice cold sterile PBS with Pen-Strep for immunofluorescence staining, optical clarification, and subsequent microscopy.

## Immunohistochemistry

For murine tissue: The fixed LM-MP tissue was washed twice in ice-cold PBS in the dark at 16 °C. The tissue was then incubated in blocking-permeabilizing buffer (BPB; 5% normal goat serum with 0.3% Triton-X) for 1 hr. While staining for antibodies that were mouse monoclonal, 5% normal mouse serum was added to the BPB. The tissue was then removed from the BPB and was incubated with the appropriate primary antibody at the listed concentration (*Supplementary file 1*) for 48 hr at 16 °C in the dark with shaking at 55 rpm. Following incubation with primary antibody, the tissue was washed three times (15 min wash each) in PBS at room temperature in the dark. The tissue was then incubated in the appropriate secondary antibody at room temperature for 1 hr while on a rotary shaker (65 rpm). The tissue was again washed three times in PBS at room temperature, counterstained with DAPI to stain the nuclei, overlaid with Prolong Antifade Gold mounting medium, cover-slipped, and imaged.

Immunostaining for ChAT was performed using fixed and unpeeled murine tissues from adult male Wnt1-Cre:*Rosa26^lsl-tdTomato* mice. The intestinal segments were obtained by perfuse fixing the mouse with ice-cold PBS and then freshly prepared 4% PFA solution. Once the intestine was harvested, the ileum was flushed with cold HBSS with 0.4 mol/L N-acetyl-l-cysteine and then PBS to remove luminal contents. After, the tissue was fixed in 4% PFA overnight. The fixed tissue was then immersed in 2% Triton-X 100 solution for 2 d at 15 °C for permeabilization. The tissue was immunostained with antibodies against Hu (ANNA1) and against ChAT (rabbit, Abnova). The primary antibody was then diluted (1:100) in the dilution buffer (0.25% Triton X-100, 1% normal goat serum, and 0.02% sodium azide

in PBS) and incubated for 1 day at 15 °C. An Alexa Fluor 647-conjugated goat anti-rabbit secondary antibody (1:200, Invitrogen) and an Alexa Fluor 488-conjugated goat anti-human secondary antibody (1:200, Invitrogen) were then used to reveal the immunopositive structure. Finally, the labeled specimens were immersed in FocusClear solution (CelExplorer) for optical clearing before being imaged by laser confocal microscopy using a Zeiss LSM 510 META. The LSM 510 software and the Avizo 6.2 image reconstruction software (VSG) were used for the 2D and 3D projection of the confocal micrographs. The Avizo software was operated under a Dell T7500 workstation and the "Gaussian Filter" function of Avizo was used for noise reduction of the micrographs.

Colchicine treatment: For CGRP immunostaining, mice were injected with Colchicine at a concentration of 5 mg/kg body weight 16 hr (overnight) before they were sacrificed. The mice were housed singly during this time and adequate gel packs were provided. Food and water were provided ad libitum. On the following day, the mice were sacrificed, and their LM-MP tissues were harvested as detailed above.

For human tissue: The fixed muscularis layer containing myenteric plexus tissue was removed from ice cold PBS and incubated in blocking-permeabilizing buffer (BPB; 5% normal goat serum, 5% normal mouse serum with 0.3% Triton-X) for 4 hr. The tissue was then removed from the BPB and was incubated with the appropriate primary antibody at the listed concentration (*Supplementary file 1*) for 5 days at 16 °C in the dark with shaking at 55 rpm. Following incubation with primary antibody, the tissue was washed five times (15 min wash each) in PBS at room temperature in the dark. The tissue was then incubated in the appropriate secondary antibody at 16 °C in the dark with shaking at 55 rpm for 2 days. The tissue was again washed in dark for five times in PBS that contained DAPI at room temperature. After the final wash, the tissue was suspended in tissue clarification buffer CUBIC *Susaki et al., 2014* for 1 hr at 4 °C in the dark after which it was overlaid with Prolong Antifade Gold mounting medium, cover-slipped, and imaged. Briefly, the CUBIC optical clarification buffer was made by mixing 2.5 g of urea (25% by wt), 2.5 g of N, N, N´, N´-tetrakis (2-hydroxy-propyl) ethylenediamine (25% by wt), 1.5 g of Triton X-100 (15% by wt) in 35 ml of Distilled Water. The solution was shaken till the ingredients were dissolved and yielded a clear viscous solution.

## Microscopy

Except for ChAT microscopy, all other imaging was done by using the oil immersion 63 X objective on the Leica SP8 confocal microscope and by using the oil immersion 40 X objective on the Olympus Fluoview 3000rs confocal microscope with resonance scanning mode. For thick tissues, such as human tissues, the Galvano mode of the Olympus Fluoview 3000rs microscope that enabled higher resolution imaging and averaging was used. Images obtained were then analyzed using Fiji (https://fiji.sc/).

Live tissue imaging of the mouse gut tissue was performed by harvesting small intestinal tissue from an adult Wnt1-Cre:*Rosa26*^lsl-tdTomato mouse and immediately putting it in OptiMEM solution. The tissue was then immediately put on a chamber slide containing OptiMEM and imaged in live tissue culture conditions of the EVOS M7000 microscope under the 20 X objective.

## Enumeration of neurons

Enumeration of tdTomato⁺ and tdTomato⁻ neurons was performed on tissue with native tdTomato fluorescence. The native tdTomato fluorescence in fixed tissue was intense, thus there was no need to increase fluorescent signal using anti-RFP/tdTomato antibodies.

Enumeration of enteric neurons to study alterations to ENS neuronal numbers was performed by following the well-established protocol *Young et al., 1993*; *Zhou et al., 2013*; *Hosie et al., 2019*; *Lefèvre et al., 2020* of counting the numbers of neurons in myenteric ganglia and comparing them between animals for statistically significant differences. Identification of myenteric ganglia was performed according to our pre-determined method published earlier *Kulkarni et al., 2017*. Briefly, contiguous clusters of neurons were defined as a ganglion and the total numbers of neurons within these clusters were enumerated as numbers of myenteric neurons per ganglion. As a rule, clusters of three neurons or more were deemed to consist of a ganglion and our enumeration strategy did not count extra-ganglionic neurons. We imaged ~10 ganglia per tissue for our enumeration and each group studied had n≥3 mice. Identification and enumeration of neurons and detection of co-localization was performed manually by trained laboratory personnel. *Figure 1d* comprises of the data from

six P60 mice, which includes data from the three mice that were used for the development experiment presented in *Figure 8c*.

## Protein isolation and detection

After the LM-MP tissue was isolated, it was weighed and placed in a sterile 1.5-ml microfuge tube. 1 X RIPA buffer (Cell Signaling Technology) with Halt Protease Inhibitor Cocktail (Thermo Scientific) at 5 X concentration, Phosphatase Inhibitor Cocktails II and III (Sigma-Aldrich) at 2 X concentrations were added to the tissue lysate buffer. Tissue was disrupted using 1.0 mm silica beads in Bullet Blender 24 (Next Advance) for 5 min at highest setting. The lysate was incubated at 4 °C with shaking for 30 min, centrifuged at 14,000 rpm for 20 min and the supernatant was taken and stored in –80 °C in aliquots. Protein concentration was estimated using Bradford assay solution (Biorad) following the manufacturer's protocol. For immunoblotting, 40 µg of protein was loaded per well of 4–20% gradient denaturing gel (Biorad). Protein marker used was Precision Plus Dual Color standards (Bio-Rad). After fractionating the proteins, they were blotted onto ImmunBlot PVDF membrane (Bio-Rad) overnight at 4 °C at 17 V for 12–16 hr. After blotting, membrane was blocked with Odyssey TBS blocking buffer (Li-Cor) for 1 hr at room temperature with shaking. Incubation with primary antibodies were carried out at 4 °C with shaking for 24 hr. Following binding, the blot was washed 4 times with TBS-T (Tris Buffered Saline with 0.5% Tween) for 15 min each with shaking at room temperature. Secondary antibody incubation was carried out in dark at room temperature for 1.5 hr with shaking. The blot was then washed four times for 15 min each and imaged on Odyssey CLx system (Li-Cor).

Part of the aforementioned protocol was also followed for detecting HuB protein (recombinant Human ELAVL2, 1-346aa, from Biosource; Catalogue number MBS205995). The protein was loaded in two amounts (0.5 µg and 1.25 µg) in separate lanes of a 4–20% gradient denaturing gel (Biorad) along with protein marker Precision Plus Dual Color standards (Biorad). After fractionating the proteins, they were blotted onto ImmunBlot PVDF membrane (Biorad) overnight at 4 °C at 17 V for 12–16 hr. After blotting, membrane was blocked with Odyssey TBS blocking buffer (Li-Cor) for 1 hr at room temperature with shaking. Incubation with primary antibodies were carried out at 4 °C with shaking for 24 hr. Following binding, the blot was washed four times with TBS-T (Tris Buffered Saline with 0.5% Tween) for 15 min each with shaking at room temperature. Secondary antibody incubation was carried out in dark at room tempera-ture for 1.5 hr with shaking. The blot was then washed four times for 15 min each and imaged on Odyssey CLx system (Li-Cor).

Antibody used to detect RET was a well validated antibody used previously for detecting RET in ENS (*Qiu et al., 2016*) and was further validated by us using a western blot on total proteins isolated from murine small intestinal LM-MP (*Figure 9—figure supplement 1e*). Antibodies used are detailed in the *Supplementary file 1*.

## RNA isolation and quantitative detection of specific transcripts

The isolated tissue was stored in RNALater Solution (Ambion). RNA was isolated using RNeasy Mini Kit (Qiagen) following manufacturer's protocol. RNA quantification was carried out using Epoch Microplate Spectrophotometer (BioTek). cDNA synthesis was carried by SuperScript IV VILO Master Mix (Invitrogen). Quantitative Real-time PCR was carried out using Taqman Gene Expression Master Mix (Applied Biosystems) and Roto-Gene Q (Qiagen). The probes used are listed in *Supplementary file 1*.

## Single-cell RNA sequencing and analyses of LM-MP tissues

Single-cell preparation from murine ileal LM-MP tissues: Ileal LM-MP tissues from male littermate C57/BL6 wildtype mice were isolated by peeling as previously described (*Kulkarni et al., 2017*). Succinctly, mice were anesthetized with isoflurane and sacrificed by cervical dislocation. A laparotomy was performed and the small intestine was removed and lavaged with PBS containing penicillin-streptomycin (PS; Invitrogen). The ileum was then isolated which was then cut into 2-cm-long segments and placed over a sterile plastic rod. A superficial longitudinal incision was made along the serosal surface and the LM-MP was peeled off from the underlying tissue using a wet sterile cotton swab and placed in Opti-MEM medium (Invitrogen) containing PenStrep (Invitrogen). The tissues were then dissociated in Digestion Buffer containing 1 mg/ml Liberase (Sigma-Aldrich) in OptiMEM. Tissues from mouse 1 were dissociated in the Digestion buffer containing Liberase TH and tissues from mouse 2 were dissociated in the Digestion buffer containing Liberase TL. Dissociation was performed at

37 °C for 30 min on a rotary shaker, after which the cells were centrifuged at 200 *g* for 7 min, and the pellet was resuspended in ice cold sterile PBS. The cell suspension was passed through a 40μm cell sieve and the resulting filtered cell suspension was again centrifuged at 200 *g* for 7 min. This process of cell centrifugation and filtration was repeated two more times, after which the cells were resuspended in 1 ml ice cold sterile PBS. The repeated steps of serial cell washes and filtration removed clumps and debris and the viability of the resulting cell suspension was estimated to be >90% using Trypan Blue dye test. The cells were then processed through 10 X Genomics Chromium V2.0 system (for the 6-month-old murine tissues) and 10 X Genomics Chromium V3.1 system (for the P21 murine tissues) according to the manufacturer's suggested workflow. The processing was done at the GRCF Core Facility at the Johns Hopkins University. The pooled libraries were sequenced on an Illumina HiSeq 2500 (for 6-month-old tissues) and Illumina Novaseq (for P21 tissues) to an average depth of $3.125 \times 10^8$ reads per sample library. The sequencing was performed at the CIDR core facility at the Johns Hopkins University.

For processing data from scRNAseq on 6-month-old murine LM-MP tissues: Pre-processing of FASTQs to Expression Matrices: FASTQ sequence files were processed following a Kallisto Bustools workflow compatible with downstream RNA velocity (*Bray et al., 2016*). References required for pseudo-alignment of transcripts were obtained using the get_velocity_files (functionality of BUSpaRSE (https://github.com/BUStools/BUSpaRse); *Moses and Pachter, 2023*; *Melsted et al., 2021*), with 'L=98' for 10 X Genomics v2.0 sequencing chemistry. Reads were pseudo-aligned to an index built from Ensembl 97 transcriptome annotation (Gencode vM22; GRCm38). Across two samples processed, a total of 578,529,125 reads were successfully pseudo-aligned. Barcodes within a Hamming distance of one to known 10 X Genomics v2.0 barcodes were corrected. Reads were classified as 'spliced' or 'unspliced' by their complement to the target list of intronic sequences and exonic sequences, respectively, and subsequently quantified separately into expression count matrices. Spliced counts are used for all analyses.

Single-cell gene expression analysis: scRNA-seq count matrices were analyzed using Monocle3. 11,123 high-quality cells were identified as meeting a 200 UMI minimum threshold with a mitochondrial read ratio of less than 20%; droplets that did not meet these criteria were excluded from the analysis. Mitochondrial counts were determined as the sum of reads mapping to 37 genes annotated to the mitochondrial genome. All genes with non-zero expression were included for downstream analysis. Raw counts were first scaled by a cell-size scaling factor and subsequently log10 transformed with a pseudo-count of 1. Normalized values are used in place of raw counts unless otherwise noted.

Prior to UMAP dimensionality reduction, batch effects between the two biological replicates were assessed and corrected via the mutual nearest neighbors (MNN) algorithm as implemented by Batchelor in Monocle3[156]. 15 clusters of cells in the UMAP embedding were identified by Leiden community detection. 30 marker genes for each cluster were identified based on greatest pseudo $R^2$ values and used for supervised annotation of cell types by searching UniProt, Allen Cell Atlas and through literature search with Pubmed.

NC-derived cell clusters were identified by expression of NC markers *Ret* and *Sox10*. MEN cluster was identified by its expression of CGRP-coding *Calcb, Met*, and *Cdh3*. The pan-MENs protein marker MHCst was identified by labeling with an antibody S46 which labels all members of the MHCst family (*Stockdale and Miller, 1987*). Since the antibody does not identify a single gene product, MHCst immunostaining could not be used to identify a specific gene marker for use in the annotation of the MEN cluster. For further analysis into the MENs population, the full LM-MP dataset was subset to include only the 2223 cells annotated as MENs. These cells were re-processed as above, but with a reduced PCA dimensionality of k=20 as input for the UMAP embedding. Five clusters of cells in the UMAP embedding were identified by Leiden community detection (k=10, resolution = 5e –4).

P20 LM-MP scRNA processing and analysis: Pre-processing of FASTQs to Expression Matrices: FASTQ sequence files were processed following a Kallisto Bustools workflow compatible with downstream RNA velocity. References required for pseudo-alignment of transcripts were obtained using the get_velocity_files (functionality of BUSpaRSE https://github.com/BUStools/BUSpaRse; *BUStools, 2022*), with 'L=91' for 10 X Genomics v3.1 sequencing chemistry. Reads were processed following the same steps as the 6-month LMMP dataset. Spliced counts are used for all analyses.

Single-cell gene expression analysis: scRNA-seq count matrices were analyzed using Monocle3. A total of 11,264 high-quality cells were identified as meeting a 600 UMI minimum threshold with a

mitochondrial read ratio of less than 20%; droplets that did not meet these criteria were excluded from the analysis. Mitochondrial counts were determined as the sum of reads mapping to 37 genes annotated to the mitochondrial genome. All genes with non-zero expression were included for downstream analysis. Raw counts were first scaled by a cell-size scaling factor and subsequently log10 transformed with a pseudo-count of 1. Normalized values are used in place of raw counts unless otherwise noted.

NC-derived and MENs cell clusters were identified by expression of established marker genes. Further, the identity of MENs was confirmed by a projection analysis using MENs-specific patterns learned in the six-month dataset. To investigate differences in division potential within the ENS, the data was subset to NENs, neuroglia, and MENs for computational cell cycle analysis (NENs: n=526; neuroglia: n=844; MENs: n=510). Continuous scores for cell cycle position (0-2π) were calculated via Tricycle by projecting each cell into the default reference space. The relevance of these scores was confirmed by profiling the scores' correlation with expression of genes with known variation over the cell cycle. Lastly, each cell was unambiguously characterized as 'non-cycling' (0–0.5π or 1.5-2π) or 'cycling' (0.5–1.5π) by binarizing Tricycle scores.

External datasets: Spliced and unspliced count matrices for mesenchymal subset of the Gut Cell Atlas were obtained from https://www.gutcellatlas.org/, *Elementaite et al., 2021* and existing annotations were used when available. SnRNAseq counts matrices from *May-Zhang et al., 2021* were downloaded from GEO GSE153192. Bulk RNAseq dataset from the Human Obstructed defecation study by *Kim et al., 2019* was obtained from GEO GSE101968.

Pattern discovery and ProjectR analyses: Pattern discovery was utilized to identify sets of co-expressed genes that define cell-type specific transcriptional signatures. The normalized expression matrix was decomposed via non-negative matrix factorization (NMF) as implemented in the R package *NNLM*, with k=50 and default parameters. Cell weights for each pattern were grouped by assigned cell-type and represented by heatmap. Pattern vectors were hierarchically clustered by a Euclidian distance metric, implemented in ComplexHeatmap (*Gu et al., 2016*). These patterns were then tested on the bulk RNA-Seq expression matrix for the Human Obstructed Defecation study (*Kim et al., 2019*). The log2 expression (log2(rpkm +1)) from this study was projected onto the NMF patterns using projectR (*Stein-O'Brien et al., 2019*). Students' t tests were performed on the projection weights from the Control and OD groups to test for differences between them. For the projection of datasets generated by *May-Zhang et al., 2021* and *Elementaite et al., 2021*, log10-transformed normalized counts were projected.

## Analysis of May-Zhang et al. dataset

Raw UMI count matrices for the single-nucleus RNA sequencing of enteric neurons performed by *May-Zhang et al., 2021* from the ileum of adult mice were obtained from GEO (GSE153192). The DropletUtils package (v1.10.3) (*Lun et al., 2019*; *Griffiths et al., 2018*) was used to compute barcode ranks, knee, and inflection point statistics for all the samples. All barcodes in the 10 X genomics libraries with total UMI greater than the inflection point were considered to be cell-containing droplets and retained for further analysis. The inDrop libraries were filtered to retain all barcodes with UMI greater than 200. The filtered matrices were combined to form a SingleCellExperiment object.

Cell-specific scaling factors accounting for composition biases were computed using computeSumFactors (scran v1.18.7) (*Lun et al., 2016*). The raw UMI counts were then scaled using the scaling factors and log 2 transformed after addition of a pseudo count of 1 using logNormCounts (scater v1.18.6) (*McCarthy et al., 2017*). Batch effects between the samples were corrected using fastMNN (batchelor v1.6.3) (*Haghverdi et al., 2018*).

Unsupervised clustering was performed using a graph-based clustering technique. The graph was obtained using buildSNNgraph (scran v1.18.7) (*Lun et al., 2016*) and the louvain algorithm implemented in the igraph package (v1.2.7)(*Csardi and Nepusz, 2006*) was used to identify communities. Differential expression testing was performed between all pairs of clusters using findMarkers (scran v1.18.7) (*Lun et al., 2016*). One-sided Welch's t-test was used to identify genes that were upregulated in a cluster as compared to any others in the dataset. Benjamini-Hochberg method was applied (to combined p-values) to correct for multiple testing and the genes were ranked by significance. The top ranked genes were used as marker genes for the clusters. The log fold change from the pairwise comparison with the lowest p-value was used as the summary log fold change.

The log normalized counts from this dataset were then projected into the NMF patterns learned on our scRNAseq data. This was performed using transfer learning implemented in projectR (v1.6.0). The projection weights corresponding to the four MEN-specific patterns were then clus-tered using graph-based clustering and visualized by UMAP (uwot v0.1.10) (*Melville, 2020*; https://github.com/jlmelville/uwot; *Melville, 2023*).

## Whole-gut transit time analyses

Whole-gut transit time (WGTT) for every mouse was analyzed by the method using the carmine red protocol (*Kulkarni et al., 2017*). Mice were placed in individual cages and deprived of food for 1 hr before receiving 0.3 mL 6% (wt/vol) carmine solution in 0.5% methylcellulose by oral gavage into the animal's stomach. The time taken for each mouse to produce a red fecal pellet after the administration of carmine dye was recorded in minutes. The experiment was terminated at 210 min post-gavage and the WGTT of any mice that did not expel the red dye at the termination was marked at the value of 210 min. The mean difference in whole gut transit time (in minutes) between both the *Ret+/+* and *Ret +/-*mice cohorts, and the GDNF and Saline-treated Control cohorts were analyzed statistically.

## In vivo injections

GDNF injection: Similar to prior report that gave sub-cutaneous injections of GDNF to post-natal mice (*Wang et al., 2010*), we took six littermate 10-day-old (P10) male Wnt1-Cre:*Rosa26Isl-tdTomato* mice and divided into two subgroups, GDNF and Control. Each mouse in the GDNF group was injected sub-cutaneously with 50 µl of 2 mg/ml of GDNF (Peprotech Catalogue #: 450–44) every other day, whereas the Control group was injected with 50 µl of sterile saline. The mice were given five doses and then sacrificed on P20, after which their LM-MP tissues were isolated as detailed above. The tissues were then immunostained with antibodies against Hu and imaged. In a separate experiment, adult (P60) mice were also injected sub-cutaneously with GDNF (100 µl of 100 µg/ml of GDNF). The mice were given five doses over a course of 10 days and then sacrificed on P70, after which their LM-MP tissues were isolated as detailed above. For studying the effect of GDNF on aging mice, two cohorts of 17-month-old male C57BL/6 mice (n=5 mice/cohort) were obtained from the aging colony of the National Institute of Aging. Before the start of dosing, the whole gut transit time was assayed. The animals were then injected daily sub-cutaneously either with 100 µl of saline (Control) or 100 µl of GDNF (500 µg/ml) for 10 consecutive days, after which the mice were sacrificed, and their LM-MP tissues were isolated as detailed above. The tissues were then immunostained with antibodies against Hu and imaged.

HGF injection: Similar to prior report that gave sub-cutaneous injections of HGF to post-natal mice, we took six littermate 10-day-old (P10) male Wnt1-Cre:*Rosa26Isl-tdTomato* mice and divided into two subgroups, HGF and Control. Each mouse in the HGF group was injected sub-cutaneously with 100 µl of 2 mg/ml of HGF (Peprotech Catalogue #:315–23) every other day, whereas the saline group was injected with 100 µl of sterile saline. The mice were given five doses and then sacrificed on P20, after which their LM-MP tissues were isolated as detailed above. The tissues were then immunostained with antibodies against Hu and imaged.

## Statistics

Data was analyzed using Graphpad Prism 8.3.1 and R using Unpaired Students t-test, Simple Linear Regression, and Ordinary One-Way ANOVA.

## Data

All raw data are provided in *Supplementary files 2 and 3*. We imaged atleast 10 ganglia per tissue for our enumeration and each group studied had n≥3 mice. Raw single-cell RNA sequencing data is archived on the NCBI GEO server and can be accessed under the accession numbers GSE156146 and GSE213604.

## Code

Code generated for the scRNA/snRNA analyses is available at *Slosberg, 2023b*; https://github.com/jaredslosberg/6month_ENS/.

## Acknowledgements

We would like to thank Dr. Chulan Kwon (JHU) for his kind gift of the Mesp1-cre mice, Dr. Jeremy Nathans (JHU) for his kind gift of the Tek-EGFP, Tek-Cre and $Hprt^{lsl-tdTomato}$ lineage-traced mice and his support, and Dr. Vanda Lennon (Mayo Clinic) for her kind gift of the ANNA1 anti-Hu antisera. We thank Dr. Akira Sawa (JHU) for his help with the $Rosa26^{lsl-EGFP}$ mouse line, and Dr Meera Murgai (NCI) for her help with the Myh11$^{CreERT2}$:$Rosa26^{lsl-YFP}$ mouse tissues. We thank Dr. Xinzhong Dong (JHU) and Dr. Mark Donowitz (JHU) for their support. The microscopy was performed on the Ross Imaging Core at the Hopkins Conte Digestive Disease Center at the Johns Hopkins University (P30DK089502) using the Olympus FV 3000rs (procured with the NIH-NIDDK S10 OD025244 grant) and we thank Dr. George McNamara for his help with confocal training and imaging. The 10 X Genomics Chromium processing for scRNAseq was performed at the GRCF Core and the sequencing was performed at the CIDR core at the Johns Hopkins University. SK was funded through a grant from the Ludwig Foundation, a grant from the NIA (R01AG066768), a pilot award from the Hopkins Digestive Diseases Basic & Translational Research Core Center grant (P30DK089502), and a pilot award from the Diacomp initiative through Augusta University. LAG was funded in part by a Johns Hopkins Catalyst Award and a grant from the NIA (R01AG066768). JS was funded through the Maryland Genetics, Epidemiology, and Medicine training program sponsored by the Burroughs Welcome Fund. PJP was funded through the Hopkins Conte Digestive Disease Center at the Johns Hopkins University (P30DK089502), NIDDK (R01DK080920), the Maryland Stem Cell Research Foundation (MSCRF130005), and a grant from the AMOS family.

We wish to thank Prof. Lior Pachter at Caltech for his help with responding to the reviewer's comments.

## Additional information

### Funding

| Funder | Grant reference number | Author |
| --- | --- | --- |
| National Institutes of Health | R01AG066768 | Subhash Kulkarni<br>Loyal A Goff |
| National Institutes of Health | R01DK080920 | Pankaj Jay Pasricha |
| Ludwig Family Foundation | Pilot and feasibility grant | Subhash Kulkarni |
| National Institutes of Health | P30DK089502 | Subhash Kulkarni<br>Pankaj Jay Pasricha |
| Augusta University | Diacomp program | Subhash Kulkarni |
| Burroughs Wellcome Fund | Maryland Genetics and Epidemiology and Medicine training program | Jared Slosberg |
| Johns Hopkins University | Catalyst Award | Loyal A Goff |
| Maryland Stem Cell Research Fund | MSCRF130005 | Pankaj Jay Pasricha |

The funders had no role in study design, data collection and interpretation, or the decision to submit the work for publication.

### Author contributions

Subhash Kulkarni, Conceptualization, Resources, Data curation, Formal analysis, Supervision, Funding acquisition, Validation, Investigation, Visualization, Methodology, Writing – original draft, Project administration, Writing – review and editing; Monalee Saha, Data curation, Validation, Investigation, Methodology; Jared Slosberg, Data curation, Software, Formal analysis, Investigation, Writing – review and editing; Alpana Singh, Data curation, Investigation; Sushma Nagaraj, Data curation, Formal analysis, Investigation; Laren Becker, Resources, Investigation; Chengxiu Zhang, Alicia Bukowski, Zhuolun Wang, Guosheng Liu, Jenna M Leser, Mithra Kumar, Shriya Bakhshi, Matthew J Anderson, Mark

Lewandoski, Investigation; Elizabeth Vincent, Resources; Loyal A Goff, Resources, Data curation, Software, Formal analysis, Supervision, Funding acquisition, Investigation, Methodology, Writing – original draft, Writing – review and editing; Pankaj Jay Pasricha, Conceptualization, Resources, Data curation, Formal analysis, Supervision, Funding acquisition, Visualization, Writing – original draft, Project administration, Writing – review and editing

**Author ORCIDs**
Subhash Kulkarni ⓘ https://orcid.org/0000-0002-2298-0623
Jared Slosberg ⓘ http://orcid.org/0000-0002-1803-2815
Alpana Singh ⓘ http://orcid.org/0000-0001-6532-8654
Sushma Nagaraj ⓘ http://orcid.org/0000-0001-5166-1309
Jenna M Leser ⓘ http://orcid.org/0000-0002-3037-0366
Matthew J Anderson ⓘ http://orcid.org/0000-0001-9387-5743
Mark Lewandoski ⓘ http://orcid.org/0000-0002-1066-3735
Loyal A Goff ⓘ http://orcid.org/0000-0003-2875-451X
Pankaj Jay Pasricha ⓘ http://orcid.org/0000-0002-8727-683X

## Ethics

Experimental protocols (Protocol number M021M105) were approved by The Johns Hopkins University's Animal Care and Use Committee in accordance with the guidelines provided by the National Institutes of Health. Presence of vaginal plug was ascertained as 0.5 days post-fertilization, and this metric was used to calculate age of mice. Only male mice were used for the studies detailed in this report. Animals were sacrificed post-anesthesia (using Isoflurane) by cervical dislocation, and every effort was made to minimize suffering.

Joint Public Review: https://doi.org/10.7554/eLife.88051.2.sa1
Reviewer #1 (Public Review): https://doi.org/10.7554/eLife.88051.2.sa2
Reviewer #2 (Public Review): https://doi.org/10.7554/eLife.88051.2.sa3
Reviewer #3 (Public Review): https://doi.org/10.7554/eLife.88051.2.sa4
Author Response: https://doi.org/10.7554/eLife.88051.2.sa5

# Additional files

## Supplementary files

• Supplementary file 1. List of all the primary and secondary antibodies and TaqMan probes used in this study.

• Supplementary file 2. Cluster-specific transcripts for the various clusters identified in the scRNAseq analyses of the P21 and the 6 month LMMP datasets.

• Supplementary file 3. Raw data of all experiments conducted and presented in this report.

• Supplementary file 4. scRNAseq data metrics for various detected cell populations present in the various clusters in the two single cell RNA sequencing experiments presented in this study.

## Data availability

Generated sequencing data has been deposited to GSE213604 (postnatal day 180) and GSE156146 (postnatal day 20). Code for scRNA-seq analysis has been made available at https://github.com/jared-slosberg/6month_ENS (copy archived at *Slosberg, 2023a*). All information on antibodies and probes used is in *Supplementary file 1* and the raw data is posted in *Supplementary file 3*.

The following dataset was generated:

| Author(s) | Year | Dataset title | Dataset URL | Database and Identifier |
|---|---|---|---|---|
| Kulkarni S, Saha M, Becker L, Singh A, Slosberg J, Zhang C, Bukowski A, Wang Z, Liu G, Leser J, Kumar M, Bakhshi S, Anderson M, Lewandowski M, Nagaraj S, Vincent E, Goff LA, Pasricha PJ | 2023 | Enteric nervous system: timecourse analysis of longitudinal muscle and myenteric plexus (LMMP) in mice | https://www.ncbi.nlm.nih.gov/geo/query/acc.cgi?acc=GSE213604 | NCBI Gene Expression Omnibus, GSE213604 |

The following previously published datasets were used:

| Author(s) | Year | Dataset title | Dataset URL | Database and Identifier |
|---|---|---|---|---|
| Kulkarni S, Saha M, Becker L, Wang Z, Leser J, Kumar M, Bakhshi S, Anderson M, Lewandoski M, Slosberg J, Nagaraj S, Vincent E, Goff LA, Pasricha PJ | 2021 | Age-dependent replacement of neural-crest derived enteric neurons by a novel mesodermal lineage and its functional consequences | https://www.ncbi.nlm.nih.gov/geo/query/acc.cgi?acc=GSE156146 | NCBI Gene Expression Omnibus, GSE156146 |
| May-Zhang AA, Tycksen Eric, Southard-Smith AN, Deal KK, Benthal JT, Buehler DP, Adam M, Simmons AJ, Monaghan JR, Matlock BK, Flaherty DK, Steven Potter S, Lau KS, Michelle Southard-Smith E | 2020 | Combinatorial Transcriptional Profiling of Mouse and Human Enteric Neurons Identifies Shared and Disparate Subtypes In Situ | https://www.ncbi.nlm.nih.gov/geo/query/acc.cgi?acc=GSE153192 | NCBI Gene Expression Omnibus, GSE153192 |
| Kim M, Metzger M | 2017 | Obstructed defecation - an enteric neuropathy? An exploratory study of patient samples | https://www.ncbi.nlm.nih.gov/geo/query/acc.cgi?acc=GSE101968 | NCBI Gene Expression Omnibus, GSE101968 |
| Marklund U | 2020 | Single Cell RNA-sequencing analysis of the juvenile and embryonic mouse enteric nervous system (ENS) of the small intestine | https://www.ncbi.nlm.nih.gov/geo/query/acc.cgi?acc=GSE149524 | NCBI Gene Expression Omnibus, GSE149524 |

*Continued*

| Author(s) | Year | Dataset title | Dataset URL | Database and Identifier |
|---|---|---|---|---|
| Elmentaite R, Kumasaka N, Roberts K, Fleming A, Dann E, King HW, Kleshchevnikov V, Dabrowska M, Pritchard S, Bolt L, Vieira SF, Mamanova L, Huang N, Perrone F, Kai'En IG, Lisgo SN, Katan M, Leonard S, Oliver TRW, Hook CE, Nayak K, Campos LS, Conde CD, Stephenson E, Engelbert J, Botting RA, Polanski K, Dongen SV, Patel M, Morgan MD, Marioni JC, Bayraktar OM, Meyer KB, He X, Barker RA, Uhlig HH, Mahbubani KT, Saeb-Parsy K, Zilbauer M, Clatworthy MR, Haniffa M, James KR, Teichmann SA | 2021 | Space-Time Gut Cell Atlas | https://www.gutcellatlas.org/ | Gut Cell Atlas, gutcellatlas |

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
