## [Editor Report · eLife assessment]

This paper identifies a subset of neurons within adult mouse myenteric ganglia that are not labeled via canonical neural-crest labeling, and argues, based on extensive lineage tracing, imaging and genomic data that these neurons are derived from mesoderm. There is **convincing** evidence for the existence of an unusual cell type in the gut that expresses neuronal markers, but which is derived from cells expressing markers of the mesoderm rather than the expected neural crest, which is an intriguing and **important** observation. While the data do not definitively establish the molecular taxonomy of this lineage, there is sufficient evidence to support the provocative and paradigm-shifting hypothesis of the non-ectodermal origin for enteric neurons to warrant further deeper investigation.

---

## [Referee Report · Joint Public Review]

In this manuscript, the authors challenge the fundamental concept that all neurons are derived from ectoderm. The key points of the authors argument are as follows:

1. Roughly half of the cells in the small intestinal longitudinal muscle-myenteric plexus (LM-MP) that express a pan-neuronal marker do not, by lineage tracing, appear to be derived from the neural crest.

2. Lineage tracing and marker gene imaging suggest that these non-neural crest derived neurons originate in the mesoderm, leading to their designation as mesodermal-derived enteric neurons (MENs).

3. Single-cell sequencing of LM-MP tissues confirms the mesodermal origin of MENs.

4. MENs progressively replace neural crest derived enteric neurons as mice age, eventually representing the bulk of the EN population.

There is broad agreement among the reviewers that the identification and description of this cell population is important, and that the failure of these cells to be labeled by neural crest lineage tracers is not artifactual. The work with transgenic lines is convincing that some presumptive neurons in the enteric nervous system (ENS) likely originate from an alternative source in the postnatal intestine and that this population increases in aging mice.

There is, however, ongoing disagreement between the authors and reviewers about whether the authors' provocative and potentially paradigm-changing proposal that these are neurons of mesodermal origin has been established. While the authors believe they have addressed the reviewers' concerns in multiple rounds of review (much of this prior to submission), the reviewers remain unconvinced and continue to request additional data and analyses.

A key premise of the preprint review system is that the best interests of science are not served by endlessly litigating disagreements around papers by either compelling the authors to do extensive and expensive additional experiments that they do not believe to be necessary or by treating the authors' claim as established in the face of continued skepticism. Accordingly the editor believes it is time to present this work, which everyone agrees contains important observations and valuable data, along with the following editor's synthesis of the reviewers' concerns and author responses about the question of these cells' origins. We encourage anyone interested in the details to review the already posted reviews and authors' response.

The following key issues have been raised during review:

* Is the lineage tracing and marker gene expression data definitive as to mesodermal origin?

* Are the cells analyzed in the genomic experiments the same as those identified in the lineage tracing experiments?

* Does the genomic data establish that the sub-population of cells the authors focus on are of mesodermal origin?

* Are there alternative explanations for the lineage tracing and genomic observations than a mesodermal origin?

* Is the lineage tracing and marker gene expression data definitive as to mesodermal origin? *

The proximal evidence that the authors present for a mesodermal origin of the non-NC derived cells is presented in Figure 2, which establishes the presence, via lineage tracing of Tek+ and Mesp1+ (and therefore mesoderm derived) and Hu+ (and therefore neuronal) cells. The fraction of lineage labeled cells in each case (~50%) corresponds roughly to the fraction of cells that do not appear to be NC derived.

The reviewers raise several technical questions about the lineage tracing experiments, including issues of incomplete labeling, ectopic labeling and toxicity. The authors have addressed each of these with data and/or citations, and the editor believes they have demonstrated, subject to the broader limits of lineage tracing experiments, that there are Hu+ cells in the tissue that are derived from cells that do not express NC markers and that do express mesodermal markers.

One reviewer raised the question of whether these cells are neurons. This appears to the editor to be a valid question, in that specific neuronal activity of these cells has not been established. But the authors' argument is persuasive that their Hu+ state would have led them to be designated neurons and that changing that designation based on not being derived from NC is circular. However the possibility that, despite this accepted designation, these cells are not functionally neurons should be noted by readers.

* Are the cells analyzed in the genomic experiments the same as those identified in the lineage tracing experiments, and does this data establish mesodermal origin? *

To provide orthogonal evidence for the presence of mesodermally derived enteric neurons, the authors carried out single-cell sequencing of dissociated cells from hand-dissected longitudinal muscle - myenteric plexus (LM-MP) tissue. They use standard methods to identify clusters of cells with similar transcriptomes, and designate, based on marker gene expression, two clusters to be neural crest derived enteric neurons (NENs) and mesoderm derived enteric neurons (MENs). However the reviewers raised several issues about the designation of the cells MENs, and therefore their equation with the cells identified in lineage tracing.

While the logic behind specific choices made in the single-cell analysis is not always clear in the manuscript, such as why genes not-specific to MENs were used to identify the MEN cluster and how genes were selected for subsequent analysis (although both issues are explained better in the authors' response to reviewers), they in the end identify a single large cluster that has the characteristics of MENs (it expresses both neuronal and mesodermal markers) that is (by immunohistochemistry) broadly associated with the previously described tissue MENs.

The standard methods for the delineation of clusters in single-cell sequencing data (which the authors use) are stochastic and defy statistical interpretation, and the way these data and analyses are used is often subjective. The editor shares the reviewers' confusion about aspects of the analysis, but also finds the authors' assertions that they have described a cluster of cells that express both neuronal and mesodermal genes, and that this cluster corresponds to the tissue MENs described in lineage tracing, to be broadly sound.

The biggest weakness in the single-cell data and analysis - identified by all reviewers - is the massive overrepresentation of MENs relative to NENs. The authors' explanation - that some cells are more sensitive to manipulations required to prepare cells for sequencing - is certainly well-represented in the literature and is therefore plausible. But it isn't fully satisfactory, especially because it undermines the notion that the MENs and NENs are functionally equivalent (though one could argue in response that increased fragility of NENs is why they are progressively replaced by MENs).

There are many additional questions about the single cell analysis that are difficult to resolve with the data in hand. I think everyone would agree that an ideal analysis would have more cells, deeper sequencing, and comprehensive validation of the identity of each cluster of cells. But given the time and expense required to carry out such experiments, we cannot demand them, and must take the data for what they are rather than what they could be. And in the end, it is the editors' view that these data and analyses bolster the authors' claims, without conclusively establishing them. That is, these data should neither be dismissed nor, on their own, considered definitive.

* Are there alternative explanations for the data than that they are mesodermally derived neurons? *

As discussed above, the reviewers generally agree that the lineage tracing experiments are careful and well-executed, and the authors have provided data that demonstrates that the data are highly unlikely to be due to either incomplete or ectopic lineage marking. The reviewers raise several possible alternative hypotheses, some based on the literature and some based on the genomic data. The authors discuss each in detail in their response. The editor would note that, at this stage in the history of single-cell analysis, the criteria for using single cell sequencing data to establish cell type and cell origin is are not well established, and that neither the presence nor absence of specific sets of genes in single cells should not, for both technical and biological reasons, be considered dispositive as to identity.

* Additional aspects of paper: *

There are additional intriguing aspects of the paper, especially the increase in the number of MENs relative to NENs over time, suggesting functional replacement of one population with the other, and some evidence for and speculation about what might be regulating this evolution. However these are somewhat secondary points relative to the central question at hand of whether the authors have discovered a population of mesodermally derived neurons.

* Editor's summary and comment: *

The editor believes it is a fair summary to say that the authors believe they have gone to great lengths to provide multiple lines of evidence that support their hypothesis, but that these reviewers, while appreciating the potential importance of the authors' discovery of an unusual cell type, are not yet convinced of its origin.

In an ideal world, the authors, reviewers and editor would all ultimately agree on what claims the data presented in a paper supports, and indeed this is what the traditional journal publishing system tries to achieve. But the system fails in cases like this where no consensus between authors and reviewers can be reached, as it neither makes sense to "accept" the paper and imply that it has been endorsed by the reviewers, nor to "reject" it and keep the work in peer review limbo.

There is certainly enough here to warrant the idea and the data and arguments behind it being digested and considered by people in the field. It may very well be that the authors - who have spent years working on this problem and likely know more about this population of cells than anyone on Earth - are right that they have discovered something that changes how we think about the development of the nervous system. To the extent the reviewers are representative, people are likely to need additional data to be convinced. But it is time to put that to the test.

---

## [Referee Report · Reviewer #1 (Public Review)]

The manuscript by Kulkarni et al proposes a new cellular origin of ENS, which is increased with age and therefore may be associated with the gradual decline of gut function. The study is based on an initial observation that many enteric neurons do not seem to retain tdTomato expression in Wnt1Cre-R26-Tom mice, suggesting a loss of neurons that are replaced by a non-neural crest source. Further detection of reporter expression within the ENS of Tek and Mesp Cre-lines indicated a mesodermal origin of the new enteric neurons. Mesodermally derived neurons (MENS) were associated with Met, while neural crest derived neurons (NENS) expressed Ret. GDNF could decrease occurrence of MENS (defined as tdTomato-negative cells), while HGF had the opposite effect. Age-associated decline in gut transit was alleviated with GDNF treatment, while Ret heterozygote mutants had an increase of MENS. Overall, the study suggests that neural crest derived neurons are replaced by mesodermal-derived neurons that lead to an overall reduction in GI-physiology and that manipulation of the balance between the two types of neurons could have beneficial effects of age-associated gut malfunction. Generation of neurons from non-ectodermal sources would be a paradigm shift not only in the ENS, but in the Neuroscience field as a whole. The presence of mesenchymal marker genes in subsets of cells of the ENS in native gut tissue is convincing and the lack of retained fluorescent reporter expression in ENS from the many neural and Cre drivers used is indeed clear.

The current state of the manuscript is though not conceivable as it has unsound interpretation of data at many places, most importantly there is no firm connection between the MENs identified in tissue and the scRNA cluster annotated as MENs. "scRNA-seq-MENs" show very little expression of the bona fide neuron markers used to detect "tissue-MENs" including Elavl4 and the overall proportions of "scRNA-seq-MENs" in the tissue is very far from that of "tissue-MENs". Hence, the claims that "tissue-MENs" equals "scRNA-seq MENs" could be excluded or their interpretation discussed in an unbiased manner. Marker expression of "scRNA-seq MENs" are suggestive of mesothelial cell identities, not ENS cells. Even the annotation of scRNA-seq profiles denoted as neural-crest derived enteric neurons (NENs) is highly questionable as 25% of the cells display bona fide lympathic epithelial cell markers and no neuronal markers.

---

## [Referee Report · Reviewer #2 (Public Review)]

In this study, the authors propose the possibility that some neurons in the enteric nervous system (ENS) originate postnatally from a non-ectodermal source. This possibility is investigated using a combination of transgenic lines, single cell RNA-sequencing (scRNA-seq), and immunofluorescence. Initially the authors identify a subset of neurons within myenteric enteric ganglia that are not lineage-labeled by canonical neural-crest derived cre-LoxP strategies. In their analysis, the group seeks to show that these neurons have an origin distinct from neural crest-derived progenitors that are known to initially colonize the developing gut. The team uses multiple cre lines (both Wnt1-cre and Pax3-cre) as well as several distinct reporter lines (ROSA-tdTomato, ROSA-EGFP, Hprt-tdTomato) to demonstrate that the lack of labeling by neural crest cre transgenes is consistent across several tools and not due to any transgene or reporter line artifact. Based on prior analysis that suggests some neurons in the ENS might be arising from a mesodermal lineage, the authors evaluate the possibility that mesoderm could contribute neurons to the ENS by evaluating expression of Tek-cre and Mesp1-cre tagged cell types in myenteric ganglia. The work with transgenic lines is convincing that some ENS neurons originate from an alternative source in the postnatal intestine and that this population increases in aging mice.

The authors apply single cell RNA-sequencing to identify additional markers of these non-neural crest enteric neurons. They rely on dissociation of laminar gut muscle preparations, stripped from the outside of the adult intestine, that contain many cell types including smooth muscle, vasculature, and enteric ganglia. In the analysis of this scRNA-seq data, the authors focus on a cluster of cells in the resulting UMAP plots as being the MENs cluster based on labeling of this cluster with three genes (Calcb (CGRP), Met, and Cdh3). Based on expression of these marker genes there are a very large number of MENs and very few neural crest-derived enteric neurons (NENs) seen in the UMAPs. It is not clear why this difference in cell numbers has occurred. The early lineage tracing data shown with cre transgenes (Figures 1 and 2) shows relatively equal numbers of NENs and MENs in confocal imaging studies, yet in the RNA-seq UMAPs thousands of MENs are displayed while very few NENs are present. There is the possibility that the authors have identified a cell cluster as MENs that does not coincide with the Mesp1-cre or Tek-cre lineage labeled neurons observed within enteric ganglia of the laminar gut muscle preparations. The authors state that they have "used the single cell transcriptomics to both confirm the presence of MENs and identify more MEN-specific markers", however there is not a direct relationship made in this study between the MENs imaged and the cells profiled by single cell RNA-sequencing.

In their analysis the authors note a difference in the percentage of enteric neurons labeled by the neural crest lineage tracer line, Wnt1-cre, relative to the total neurons labeled by the pan-neuronal marker HuC/D with age of the mice studied. They undertake a temporal analysis of the percentage of Wnt1-cre labeled neurons over total HuC/D neurons over the lifespan and note a decrease of Wnt1-cre labeled neurons with age. Further, the team assessed levels of growth factors that are known to promote proliferation and survival of NENs (GDNF-Ret signaling) versus factors known to promote growth of mesoderm (HGF) with age and document a decrease in GDNF-Ret signaling while HGF levels increase with age. The authors propose that the balance between these two signaling pathways is responsible for the shift in proportions of NENs versus MENs in aging animals.

Some of the conclusions of this paper are supported, but several additional analyses are needed to reach the outcomes that the authors infer:

1. Because the scRNA-seq data generated in this study derives from mixed cell populations present in laminar gut muscle preparations, there is a gap between the image data shown for the mesodermal cre lineage tracing and the MENs clusters the authors have selected in their single cell RNA-seq analysis. The absence of direct transcriptional profiling of cells labeled by Mesp1-cre or Tek1-cre expression prevents the authors from definitively connecting their in situ lineage labeling with specific clusters in the single cell RNA-seq analysis.

2. Differential gene expression is the standard approach for identifying markers of a particular cluster and yet this is lacking in this study, and the rationale for why some genes were prioritized as markers of MENs is missing from the manuscript. Reanalysis of the authors posted single cell RNA-seq data found that genes integral to calling MENs (marker genes) were detectable in the data. Met, Cdh3, Calcb, Elavl2, Hand2, Pde10a, Vsnl1, Tubb2b, Stmn2, Stx3, and Gpr88 were all expressed in very few cells and at low levels. Given this, how were these genes chosen to be marker genes for MENs, especially given the low sequencing depth utilized?

3. The authors rely on Phox2b as a marker for all ENS cells, including MENs. However, reprocessing of the authors posted single cell RNA-seq data finds that Phox2b is not detected in any of the cells in the MENs cluster and it's only expressed in very few cells of the neuroglia cluster. This discrepancy between the data the authors have generated and what is widely known about Phox2b expression in the ENS field must be explained as the absence of Phox2b message suggests there is an issue with reliance on low-depth scRNA-seq data for reaching the stated conclusions.

4. The authors have not considered potential similarities between their MENs and other developing ENS lineages, like enteric mesothelial fibroblasts reported by Zeisel et al. 2018, and further analysis is needed to show that MENs are indeed a distinct cell type. Top marker genes of the author's MENs clusters were expressed more often in the clusters that were left out of Morarach et al 2021's E15.5 and E18.5 datasets because those clusters were mostly Phox2b-negative on UMAPs. This lack of Phox2b expression matches the characteristic of the MENs clusters' Phox2b-negative status in the authors single cell dataset. It is important to note that the Morarach dataset consists of Wnt1-cre lineage labeled (originating from neural crest) flow sorted cells. This is of import as it implies that Phox2b-negative cells ARE present within the Wnt1-cre lineage labeled population, an aspect that is relevant to this study's data analysis.

5. Upon reprocessing of the authors MENs-genesis dataset with integration by sample as the authors describe, Met expression is evident within the cluster of NENs on the resulting UMAP plot and yet the authors rely on this gene as a marker of MENs. Whether Met expression is restricted to MENs should be resolved because the authors state it is exclusive to MENs and they subsequently investigate this gene across lifespan. Because it is not clear that Met is absent from neural crest derived enteric neurons this caveat complicates the interpretations of the present study.

6. The authors apply MHCst immunofluorescence to mark MENs, but do not show any RNA expression for the MHCst transcripts in their single cell data. How did the authors come to the conclusion that MHCst IHC would be an appropriate marker for MENs? This rationale is missing from the text.

---

## [Referee Report · Reviewer #3 (Public Review)]

In this manuscript, the authors challenge the fundamental concept that all neurons are derived from ectoderm. Specifically, they aim to show that while the early ENS arises embryologically from neural crest (NENs), with age it is slowly replaced by mesoderm-derived neurons (MENs). This claim is based on an array of transgenic reporter mice, immunofluorescence, and transcriptomics. They further propose that the transition from NENs to MENs is regulated by a changing balance in GDNF-RET versus HGF-MET signaling, respectively.

This is a provocative and potentially paradigm-changing proposal, but the data presented and the interpretation of that data fall short of establishing it.

1. MENs share more common characteristics with fibroblasts. The authors interpret this as representing neurons with fibroblast characteristics. Why not fibroblasts with neuronal characteristics? The ability to express neurotransmitter receptors and calcium channels is common in fibroblasts, but that isn't sufficient to characterize a neuron. For example, many cell types express neurotransmitters (CGRP in ILCs, Penk in fibroblasts). Expressing one of the Hu proteins (Elavl2) probably isn't enough to call these "neurons," especially when neurons usually express Elavl3-4 (HuC/D). Including calcium imaging and showing presence of action potentials would strengthen the argument that these are in fact neurons.

2. The scRNA-seq is unconvincing. There are several technical issues and the analysis omits important information required to make an unbiased assessment.

a. One issue in the interpretation is that MENs are shown by IHC to constitute half the neuronal population, with NENs making up the other half. The authors state that they performed an unbiased approach, sequencing all cells in the muscularis. If it were truly unbiased, then why do they detect a 28-fold increase in MENs in the single cell data? This does not reflect the IHC findings and points to an issue in technique that needs to be addressed.

b. Cell populations annotated by the author are confusing. The "unknown" population expresses many genes that are epithelial markers. This is puzzling because the authors state that they only sequenced the muscularis. This leads to questions regarding the initial samples and whether they were dissected appropriately or contaminated by another population.

c. The authors report a population of ICCs at P21 which is not identified at 6-months. Closer inspection of their data shows bona fide ICC markers, Ano1 and Kit, in their SMC cluster at 6-months, with failure to identify ICC clusters, raising questions about whether they have identified a new cell type.

d. While the authors critically examine other scRNA-seq datasets and claim that those groups mislabeled their populations, the above does not instill confidence in their ability to counter the unified literature.

1. MENs are identified based on genes that could be related to neurons, including calcium channels, neurotransmitter receptors, etc. It is worth noting that mesenchymal cells, ICCs, and smooth muscle also possess these characteristics. Therefore, it hard to justify why these MENs are considered "neurons." The authors should perform an analysis to examine homology between clusters in order to show which clusters the MENs are more similar to, neurons or otherwise.

2. Several issues raise questions about the quality of the scRNA-seq data, making interpretations very difficult:

a. MENs are identified to have higher UMI counts than other cells, which the authors interpret as the cells being bigger than others. If this is the case, why is this only observed in the P21 dataset and not at 6 months. Notably, high UMIs are also a sign of doublet contamination.

b. Authors include data from RBCs. As they do not have a nucleus, RNA abundance is low as expected. However, markers for RBCs include smooth muscle specific markers, MYH11 (an MEN marker) and Acta2. The presence of these markers can indicate high levels of "ambient RNA" which enters droplets from other cells lysed during digestion. Interestingly, MENs appear to cluster close to RBCs.

c. In light of the above possible evidence of doublet contamination and high levels of ambient RNA, the markers of MENs need to be reconsidered. MENs are stated to express markers that were previously (up until this manuscript) accepted markers of intestinal mesothelium (Ukp3b Krt19, WT1), smooth muscle cells (Myh11), and fibroblasts (Dcn, C3, Col6a1), raising the possibility that MENs are an erroneous cluster containing RNA from all these cell types.

1. The MEN population appears to be the largest cell population in the gut, which is unprecedented. The authors compare their scRNA-seq data to several other studies that have not made similar observations. Such analysis of other datasets is used to inform on the new data being generated. In the current manuscript, however, this takes the reverse approach and the authors analyze other data based on the assumption that they all mislabeled the MEN population.

a. In their assessment of Drokhlyansky et al., the authors claim that their mesothelium annotation is wrong despite expressing known mesothelial markers. This includes the gene Upk3b which is a bona fide mesothelial marker in the gut but is also expressed by "MENs." They proceed to analyze the Elmentaite et al. dataset and state that their "transitional fibroblast" population are actually MENs. That paper also has a population of Upk3b+ mesothelial cells and it is unclear why those are not actually MENs like in the Drokhlyansky et al. study.

b. The authors often refer to the study of May-Zhang et al. and their cluster annotated as "mesenchymal neurons" in the gut. It should be known that the original authors never made this claim. Rather, they acknowledge that the clusters in their study with poor correlation to neuronal profiles exhibit strong predictions for mesenchymal and vascular/immune cell types. They state: "We considered the possibility that these clusters might be non-neuronal." If these are "mesenchymal neurons" then the same logic would indicate that there are vascular neurons and immune cell neurons, and therefore this does not make a very compelling case.

1. A weakness of this study is that a lot of the data relies on reporter gene expression. The authors need to acknowledge several weaknesses of this approach. First, Wnt1-tdT recombination may be incomplete or one can have "Cre mosaicism" and therefore the lack of tdT is not sufficient evidence to say that those neurons are not neural crest-derived. Second, one can have off-target or leaky Cre expression, leading to low-level tdT expression, as seen in many of the images in this study. Third, Cre can exhibit toxicity and this may be more problematic in older mice given the long-term continuous expression of Cre (He et al, Am J Pathology, 2014;184:1660; Loonstra et al, PNAS, 2001;98:9209; Forni et al, J Neurosci, 2006;26:9593; Rehmani et al, Molecules, 2019;24:1189; Gillet et al, Sci Rep, 2019;9:19422; Stifter and Greter, Eur J Immunol, 2020;50:338).

---

## [Author Response]

We thank eLife for carrying out the peer review of our preprint. In this letter, we will provide a response to the eLife assessment, and the editor’s public review, and will also address the major points raised in the peer-review of our study.

First, we wish to inform the readers that including this review, our manuscript has now been reviewed 5 times. These have included three reviews at an earlier journal, a review at eLife under the older model, and the current review at eLife under the new model. In an effort to provide transparency and increase the reader’s confidence in our study, all the prior reviews and our rebuttals to them have been uploaded to Biorxiv and are publicly available for all readers to peruse [1]. These reviews will show that we have responded comprehensively with additional data, and analyses over the last 3 years. Of the current reviewers, Reviewer #1 (who was also Reviewer #1 at the earlier journal) has reviewed our manuscript all 5 times. At the prior journal, an additional Reviewer (#2) carried out 3 cycles of review – and we responded fully and comprehensively to all the issues and comments of that Reviewer. It is our understanding that the prior Reviewer #2 did not respond to the review request from eLife, after which eLife recruited two new Reviewers (current Reviewers #2 and #3), who have now reviewed our work twice – once under the older model and now again under the newer model.

Next, to ease readability, we will respond to the review in three parts. Part A will be dedicated to the editors’ public review. Part B will be dedicated to the response to eLife assessment, and we will respond to the reviewers’ comments in Part C.

Part A: Response to editor’s public review: We thank the editor for his nuanced and fair read of our data and our inferences, and of the multiple back-and-forth cycles of reviews and rebuttals. The editor’s public review highlights key points put forth in our data, and succinctly discusses the evidence provided for our claims. Here, we respond to each of these highlights.

(i) The editor agrees that subject to the broader limits of lineage fate-mapping experiments, which are universal for every prior and current study of vertebrate development, we have provided sufficient evidence for the presence of a population of cells within the myenteric ganglia, which shows mesodermal and not neural crest derivation, and which expresses the pan-neuronal marker Hu among other neuronal and mesenchymal/mesodermal markers.

Given that the current accepted annotation for enteric neurons depends on their expression of pan-neuronal markers (which we show are expressed by MENs), expression of neurotransmitter-encoding genes and proteins (such as CGRP, NOS1, ChAT, etc, which we show are expressed by MENs), and their localization within the enteric plexuses (we show evidence of intra-ganglionic localization of MENs in the myenteric plexus), our data suggests that in describing MENs, ours is the first report describing the presence of a mesoderm-derived neuronal population in a significant neural tissue. By virtue of the continual expansion of the MENs population with maturation and aging, we show evidence that MENs contributes to the post-natal maturation and aging of the enteric nervous system (ENS), and by reducing the proportions of MENs in aging tissue, we can rejuvenate the ENS to normalize gut function in aging mice.

(ii) The editor comments on whether beyond the accepted norm of their intraganglionic localization and expression of pan-neuronal markers, MENs can be described as functional neurons. We agree that in our manuscript, we did not test how MENs function. This is expressly because the current report is the first step in the study of MENs and does not aim to understand how MENs regulate various gut functions. In this response however, we wish to put forth a few arguments that would clarify some of the existing evidence on the functional nature of MENs as well as the current state of knowledge on ENS functions. These would help the readers understand the current evidence on the functional nature of MENs, and in addition, why it would be premature to expect MENs to exhibit canonical neuronal behavior.

a. MENs generate neurotransmitters and neuropeptides: Enteric neurons release various neurotransmitters, and their ability to generate important neurotransmitters such as nitric oxide (NO) and acetylcholine depends on their expression of enzymes Nitric Oxide Synthase 1 (NOS1) and choline acetyltransferase (ChAT). Our work shows that sub-populations of MENs express these important neurotransmitter-generating enzymes (Fig 3). Further, our data also shows that MENs express CGRP, which is an important neuropeptide for regulating various gut functions (Fig 3). These important data show that at the protein level, many MENs have the same cellular machinery as that of NENs that can help carry out regulation of important gut functions.

b. MENs have been shown to be functional in a prior study: Recently, enteric neurons have been shown to carry out significant immunomodulatory functions. These have included the expression of cytokines such as IL-18, which regulates intestinal barrier (as shown by Jarret et al. [2]), and CSF1, which regulates macrophage recruitment [3]. Jarret et al shows that the enteric neuron-derived IL-18 regulates immunity at the mucosal barrier. We show that the IL-18 – expressing enteric neurons are MENS (Fig 4), and thus, the data from Jarret et al [2] provides evidence that MENs are indeed functional in the in vivo environment.

c. We do not quite know how many enteric neurons work at the electrophysiological level: Canonical vertebrate neurons exhibit resting membrane potentials (RMP) in the range of -70 to -80 mV, and during neuronal activation, an increase in membrane potential beyond the threshold of -55 mV activates their action potential [4]. By contrast, past and recent studies have shown that the average RMP of rodent and human enteric neurons is significantly more positive than -70 mV (for human ENS: -48 ± 8 mV, for mouse ENS: -46 ± 6 mV for S neurons, -56 ± 5 mV for AH neurons) [5, 6]. These data suggest that enteric neurons show significant departures from canonical neuronal behaviors and thus, expecting MENs to adhere to canonical neuronal behavior – when most of the ENS does not adhere to expected norms - would be incorrect.

d. A neuron is not defined by its ability to generate an action potential: Neuronal behavior does not require the presence of action potentials, as observed in the neurons in *C. elegans* [7], much in the same way that the presence of action potentials is not restricted to neurons as it occurs in nonneuronal cells, including in enteroendocrine cells of the mammalian gut [8]. Thus, the presence or absence of action potentials cannot be the basis for adjudicating whether or not a neurotransmitter-expressing cell in a neural tissue is a functional neuron.

(iii) The Editor, after reading the extensive prior and recent correspondence between the authors and the reviewers on whether the cells analyzed in the transcriptomic experiments are the same as those observed in tissues (called tissue MENs by a reviewer), opined that he found “the authors' assertions that they have described a cluster of cells that express both neuronal and mesodermal genes, and that this cluster corresponds to the tissue MENs described in lineage tracing, to be broadly sound”.

We are enthused by the Editor’s opinion, as we had previously argued that our data connecting the transcriptomic data to tissue MENs is robust on the basis of extensive immunohistochemical validations of marker genes found in our single cell transcriptomic analyses. The Editor notes some confusion on why some marker genes not specific to MENs were used for the analyses and further points to the prior rebuttals we have posted on Biorxiv [1], where detailed clarifications on the choice of marker genes have been made. In the interest of readability, we direct the readers to these prior rebuttals at Biorxiv for more details. Succinctly, we initially tested canonical neuronal genes by immunolabeling (such as NOS1, ChAT, CGRP, etc) in NENs and MENs before performing single cell transcriptomic experiments. After performing the transcriptomic experiment, we next chose to validate neuronal and mesenchymal genes that were found expressed in the MENs cluster (such as DCN, SLPI, IL-18, NT-3, etc). Finally, in previous cycles of review, on the reviewer’s insistence, we included data on the expression of a host of neuronal genes and their encoded proteins (including Vsnl1, Pde10a, etc) to provide further evidence of neuronal identity of MENs.

While without a significantly large cluster of NENs, it is impossible to know in our transcriptomic data, whether a gene expressed by MENs would be similarly expressed by NENs, it is important to note that lack of detection of a gene in the single cell experiments cannot be inferred as lack of its expression in those cells, and hence, our inferences on whether any marker gene was exclusively expressed by neurons of a particular lineage were determined by immunohistochemistry.Additionally, we wish to reiterate and inform the readers that our study provides detailed analysis of prior work by May-Zhang et al [9], where they have described a small cluster of Phox2b-expressing cells from the murine myenteric plexus that shows the expression of neuronal and mesenchymal markers. Our analyses shows that the transcriptomic profile of MENs matches the molecular signature of these cells. In the longitudinal muscle – myenteric plexus layer, only glial cells and neurons express Phox2b [10], suggesting that this cluster sequenced by May-Zhang et al are cells of the myenteric plexus. We provide evidence that the majority of the MENs were left unsequenced by MayZhang et al and that this minimized the representation of MENs in their data (Fig 5). These data together provide important confirmation of our argument that the transcriptomic MENs point to no other cell type but the tissue MENs.

(iv) The Editor opines that a weakness in our current data is the significant overrepresentation of MENs in the single cell experiment, while also noting that our “explanation - that some cells are more sensitive to manipulations required to prepare cells for sequencing - is certainly well-represented in the literature and is therefore plausible….But it isn't fully satisfactory”. In our prior arguments (as well as in Part C), we have provided explanations based on prior observations that the issues of disproportionate representation of cell types are a technical limitation of the single cell transcriptomic methodology, which is prevalent in other experimental conditions for ENS (including the gut cell atlas study by Elmentaite et al [11]), and for other cell types in various organs. Due to this limitation, proportions of cells in the single cell space should not be inferred as their proportions in tissues.We also agree with the Editor that owing to the low representation of NENs, our data does not allow for a detailed comparison of the similarities and differences between the neurons of the two lineages, and that “an ideal analysis would have more cells, deeper sequencing, and comprehensive validation of the identity of each cluster of cells.” While in this study our aim was to describe the existence of MENs and not to perform an in-depth characterization of their sub-populations, we agree that this is the logical next step in creating a better understanding of the true diversity of ENS neurons. To that, we are currently evolving the methodologies to allow for a deeper and a more comprehensive analyses and validation of the various MENs populations, and study how they differ from NENs. We aim to publish these data in our next study.

(v) We agree with the Editor’s assessment on our transcriptomic data that “these data and analyses bolster the authors' claims, without conclusively establishing them. That is, these data should neither be dismissed nor, on their own, considered definitive.” We have only used our single cell transcriptomic data to provide additional support for our claims (which are based on extensive lineage fate mapping and immunohistochemical analyses) and are not using these as a stand-alone definitive proof of a mesodermal origin. The data from the transcriptomic experiments were used to learn additional molecular markers, whose expression in MENs in tissue could be tested by immunohistochemistry. With this methodology, we provide data on the coexpression of neuronal and mesenchymal markers by MENs, and test by computational analyses whether similar neuronal population exists in other murine and human transcriptomic datasets.

In addition, we completely agree with the Editor that “at this stage in the history of single-cell analysis, the criteria for using single cell sequencing data to establish cell type and cell origin is are not well established, and that neither the presence nor absence of specific sets of genes in single cells should not, for both technical and biological reasons, be considered dispositive as to identity.” We are very mindful of this limitation of these analyses and hence have continually ensured that our study only uses transcriptomic data of postnatal MENs to define a preliminary molecular signature of MENs, and not to infer developmental origins of MENs.

(vi) We thank the Editor for his summary and for highlighting that despite using multiple lines of evidence to support our hypothesis, the current reviewers are not yet convinced of the mesodermal origin of MENs. Our study utilizes well established tools for lineage fate-mapping (which are the only tools that currently are widely disseminated and accepted in the field of developmental biology) to show that MENs are not derived from the (Wnt1-cre, Pax3-cre -expressing) neural crest and instead are derived from the (Mesp1-cre, Tek-cre -expressing) mesoderm. The reviewers agree that by using multiple lines of evidence, we have established that our results of lineage fate-mapping are real and not due to any artifact. With this rationale, the reviewers would agree that MENs observed in tissue do not show evidence of derivation from neural crest while showing evidence of derivation from the mesoderm. Despite this, we cannot ascertain the scientific rationale for why despite agreeing with our lineage fate-mapping methods and analyses, the reviewers remain unconvinced as to the developmental origins of MENs. We do not know what other experiment would pass the reviewers’ muster to definitively annotate the mesodermal origins of MENs.

We wish to highlight that a recent study in ctenophores, where the investigators show evidence of a syncytial neural net [12], shows that much of the dogmatic view of how neurons are supposed to work is being overturned and newer paradigms that support broader interpretations for the definitions of neurons and how they regulate functions are being established. Our work on the developmental origins of a large population of neurons of the ENS, which is regarded as a primordial and conserved neural tissue, should be viewed in a similar vein.

Part B: Response to eLife assessment: Ours is the first report on the mesodermal derivation of a large population of neurons in a significant nervous system in mammals. We show that this population of neurons, called MENs, is molecularly distinct from the canonical neural crest-derived lineage of neurons, and that the post-natal ENS shows evidence of increasing presence of MENs in the maturing and aging ENS. We show that the two neuronal lineages are sensitive to their own growth factors, which can be used to manipulate their proportions in tissue, and thereby provide a potential rejuvenating therapy for age-associated intestinal dysmotility. We also show that on the basis of MENs’ marker expression, MENs maybe present in the human ENS, and that disproportionate changes in their proportions are associated with chronic gut dysmotility disorders. Our work has profound implications in the multiple fields, including those of enteric and peripheral neurobiology, developmental biology, medicine, and aging. We are thankful that the eLife assessment found that we provide sufficient evidence for this important work.

Part C: Response to Reviewers: Here, we wish to note that all the comments of the reviewers have been sufficiently addressed in prior reviews. All prior reviews, and our extensive rebuttals are available at our preprint for the readers’ perusal [1]. In this response, we wish to succinctly address some comments that have continued to emerge in this round of peer-review.

(i) We wish to highlight that the Reviewers 1 and 2 agree that our lineage-fate mapping experiments are correct and the results are not a result of any artifact. In addition to the additional reviewer in the prior reviews at an earlier journal, whose comments were addressed in full, we have a total of three reviewers who agree that our results on lineage fate-mapping are robust. Reviewer 3 comments on the possibility of ‘cre mosaicism’ or the deleterious issues with long-term expression of cre. Our prior rebuttals have dealt with this comment at length, but succinctly, our results are (a) based on extensive cre and floxed reporter controls for both the lineages, and (b) replicate observations made by other labs – including the Pachnis, the Heuckeroth, and the Southard-Smith labs to provide confidence that these are not due to any artifacts in cre or reporter gene expression. Finally, cre in the two lineage fate mapping systems (Wnt1-cre and Mesp1-cre) is only developmentally expressed and thus, there is no reasonable possibility that our results would be impacted by long-term expression of cre. Thus, our results and inferences on lineage fate mapping, which is central to our annotation of the two distinct developmental lineages, correctly describe the developmental origin of MENs.

(ii) By using extensive immunolabeling for (~21) markers that were learnt from our transcriptomic experiments, we provide evidence of the firm connection between the cluster of cells we annotated as MENs in the single cell transcriptomic experiments and the MENs we observe in tissues. Thus, we have performed more validation for these neurons than any other studies that have traditionally used 2 - 3 markers to validate a cell cluster in the ENS.

In addition, by providing evidence of the expression of pan-neuronal marker Hu and other ENS markers that include NOS1, ChAT, CGRP, etc and ~40 neuronally significant genes, we have established the neuronal nature of MENs. With regards to annotation of MENs as neurons, we expected and understand the confusion in the field with our discovery of mesoderm-derived neurons that coexpress neuronal and mesenchymal markers. We wish to put forth the following arguments for the readers to consider.

a. The annotation of Hu-expressing cells within the myenteric ganglia has been traditionally accepted as an enteric neuron. In those terms, by virtue of their intra-ganglionic presence and expression of Hu (and our data shows that Hu antibodies do not discriminate between the three neuronal isoforms of Hu) and other neuronal markers such as NOS1, ChAT, and CGRP, MENs should be annotated as neurons. We had addressed the semantic nature of this question in our last rebuttal (review #3, reviewer 1), which is available on the preprint [1].

b. As the molecular data on MENs suggests that they have significantly different biology, it would not be unreasonable to expect that their neuronal behavior may be quite different. This is underscored by the fact that we observe many MENs to lack the expression the protein SNAP25, whose presence is thought to be central to canonical neuronal behavior. We also cite evidence that neurons without SNAP-25 expression occur in the CNS neurons as well. In light of these discoveries, gauging the biology and neuronal behavior of MENs is a significant undertaking as it cannot be assumed that the behavior of MENs will be similar to that of NENs.

c. It is not logical to say that “Expressing one of the Hu proteins (Elavl2) probably isn't enough to call these "neurons" especially when neurons usually express Elavl3-4 (HuC/D)” especially when there are currently no antibodies to discriminate between the three neuronal gene products.

d. While at the outset it maybe an easy proposition to suggest that we provide evidence of neuronal activity in MENs by calcium flux or by electrophysiological means, it is important to know that calcium flux exists in all cells of the gut wall, including in smooth muscles, enteric glia, neurons and thus studying calcium flux will not provide definitive proof of neuronal behavior in MENs. Further, we reiterate from Part A of this response letter that “neuronal behavior does not require the presence of action potentials, as observed in the neurons in *C. elegans* [7], much in the same way that the presence of action potentials is not restricted to neurons as it occurs in non-neuronal cells, including in enteroendocrine cells of the mammalian gut [8]. Thus, the presence or absence of action potentials cannot be the basis for adjudicating whether or not a neurotransmitter-expressing cell in a neural tissue is a functional neuron.”

(iii) Our identification and validation of the molecular identity MENs using single cell transcriptomic experiments helps us establish the congruency of our cell cluster with a similar cluster enteric neurons previously observed by the SouthardSmith lab in their analyses. Thus, similar to our observations on the lineage-fate mapping models, observations on our transcriptomic data are also in-line with the observations made by other labs in the field.

(iv) To address any remaining confusion in the minds of the reviewers and of the readers about the correct methodology for interpreting single cell transcriptomic data and the limitations of this technique, we wish to reiterate that:

a. Single cell or nucleus RNA sequencing methods are biased towards sequencing transcripts that are abundant relative to all other transcripts for that individual cell (detection and amplification bias). Thus, while the same transcript may be equally expressed at an absolute level in two different cells, it will be more readily sequenced and detected in the cell where the transcript is relatively more abundant.

b. Correct interpretation of single cell/nucleus transcriptomic data relies on an understanding that not all transcripts of a cell can be sequenced and detected, and thus absence of the expression of transcripts in a cell does not imply absent gene expression. Together this shows the fallacy of an argument often put-forth by the reviewers that a lack of detection of a gene transcript (for e.g. Phox2b) in MENs in a scRNAseq experiment should be inferred as a lack of expression of this transcript, even though we provide evidence of the expression of PHOX2B protein in MENs, and the expression of this transcript in the MENs in the data from the Southard-Smith lab.

c. scRNAseq is not a technique where annotation of a previously unknown cluster should be biased by the detection of expression of one or two genes, and instead establishing identity or conferring novel annotation of that cluster is defined by co-expression of several genes which must be validated in tissue.

d. It is well known that enzyme-based dissociation methods are unequally tolerated by diverse cell types, which is known to cause over- or underrepresentation of several cell types in scRNAseq (Uniken Venema et al.[13], who showed that dissociation method drives detection and abundance of cells sequenced; Wu et al.[14], showed the existence of similar dissociation bias in the kidney; Tiklova et al.[15] showed that specific subpopulations of Dat-expressing neurons in the developing mammalian brain were underrepresented in scRNAseq). The Gut Cell Atlas study (Elmentaite et al.[16]) was not able to detect NENs in the adult intestinal tissue. The lack of detectable canonical enteric neurons (NENs) in the adult tissue in their study should not be viewed as an absence of NENs in those tissues, and with the same logic, a restricted abundance of NENs and a larger abundance of MENs in our dataset cannot and should not be viewed as a reliable indicator of their actual proportions in tissues. The aim of our study is not to provide a comprehensive molecular atlas for all cells that reside in the LM-MP tissue layer, but to use the information in this atlas to identify a cell cluster that best describes MENs, and then use additional tools to validate this information.

e. Without extensive validation by immunohistochemical or other means, detection of transcripts of a particular gene ‘Z’ (which is known to be expressed in cell type ‘X’) in a particular cell cluster ‘A’ of a single cell transcriptomic dataset does not directly imply that cell cluster ‘A’ points to cell type ‘X’. Thus, the detection of transcripts of the gene Wt1 (which is known to be expressed in mesothelial cells) in MENs, in itself does not mean that the MENs cluster comprises of mesothelial cells. It simply suggests that in addition to its expression in mesothelial cells, Wt1 gene is also expressed by MENs – an inference which is supported by data that show the expression of LacZ in myenteric ganglia cells in the WT1-cre transgenic mouse (Wilms et al 2005 [17]).

(v) Our study has performed two scRNAseq studies, first to establish the distinct molecular signature of MENs, and second to provide transcriptomic evidence of MENs-genesis. In the last and current review, Reviewer 2 opines that we should perform an additional single cell RNA sequencing experiment just to show that the MENs cluster is represented in the mesoderm-enriched transcriptomic data. There is no doubt that owing to the expression of various mesodermal-markers that we show are expressed by MENS (both transcriptomically in scRNAseq and at the level of proteins in tissues), the cluster of MENs is mesodermal in origin. Thus, we have already provided evidence and met a higher burden of proof on the mesodermal identity of MENs, and thus, we do not consider the costly scRNAseq experiment proposed by the reviewer a definitive experiment that would justify the time or the cost.

(vi) Our prior rebuttals have provided the reviewers with evidence that shows that our study has used standard bioinformatic pipelines to analyze our data, and our inferences of the transcriptomic data are sound and well validated by additional methods.

(vii) Many comments of the reviewers that required textual edits were already carried out after the prior review at eLife. While a revised version of our manuscript was submitted to eLife for the current review, it is unfortunate that the reviewers have not updated many of their comments. For the sake of brevity, we will not be responding further to the comments that we have already addressed at length in prior rebuttals or in form of textual edits.

References

1. Kulkarni, S., et al., Age-associated changes in lineage composition of the enteric nervous system regulate gut health and disease. bioRxiv, 2022: p. 2020.08.25.262832.

2. Jarret, A., et al., Enteric Nervous System-Derived IL-18 Orchestrates Mucosal Barrier Immunity. Cell, 2020. 180(1): p. 50-63 e12.

3. Muller, P.A., et al., Crosstalk between muscularis macrophages and enteric neurons regulates gastrointestinal motility. Cell, 2014. 158(2): p. 300--13.

4. Chrysafides, S.M., S.J. Bordes, and S. Sharma, Physiology, Resting Potential, inStatPearls. 2023: Treasure Island (FL) ineligible companies. Disclosure: Stephen Bordes declares no relevant financial relationships with ineligible companies. Disclosure: Sandeep Sharma declares no relevant financial relationships with ineligible companies.

5. Yew, W.P., et al., Electrophysiological and morphological features of myenteric neurons of human colon revealed by intracellular recording and dye fills. Neurogastroenterol Motil, 2023. 35(4): p. e14538.

6. Furukawa, K., G.S. Taylor, and R.A. Bywater, An intracellular study of myenteric neurons in the mouse colon. J Neurophysiol, 1986. 55(6): p. 1395-406.

7. Liu, Q., G. Hollopeter, and E.M. Jorgensen, Graded synaptic transmission at the *Caenorhabditis elegans* neuromuscular junction. Proc Natl Acad Sci U S A, 2009. 106(26): p. 10823-8.

8. Gribble, F.M. and F. Reimann, Enteroendocrine Cells: Chemosensors in the Intestinal Epithelium. Annu Rev Physiol, 2016. 78: p. 277-99.

9. May-Zhang, A.A., et al., Combinatorial Transcriptional Profiling of Mouse and Human Enteric Neurons Identifies Shared and Disparate Subtypes In Situ. Gastroenterology, 2021. 160(3): p. 755-770 e26.

10. Corpening, J.C., et al., A Histone2BCerulean BAC transgene identifies differential expression of Phox2b in migrating enteric neural crest derivatives and enteric glia. Dev Dyn, 2008. 237(4): p. 1119-32.

11. Elmentaite, R., et al., Cells of the human intestinal tract mapped across space and time. Nature, 2021. 597(7875): p. 250-255.

12. Burkhardt, P., et al., Syncytial nerve net in a ctenophore adds insights on the evolution of nervous systems. Science, 2023. 380(6642): p. 293-297.

13. Uniken Venema, W.T.C., et al., Gut mucosa dissociation protocols influence cell type proportions and single-cell gene expression levels. Sci Rep, 2022. 12(1): p. 9897.

14. Wu, H., et al., Comparative Analysis and Refinement of Human PSC-Derived Kidney Organoid Differentiation with Single-Cell Transcriptomics. Cell Stem Cell, 2018. 23(6): p. 869-881 e8.

15. Tiklova, K., et al., Single-cell RNA sequencing reveals midbrain dopamine neuron diversity emerging during mouse brain development. Nat Commun, 2019. 10(1): p. 581.

16. Elmentaite, R., et al., Single-Cell Sequencing of Developing Human Gut Reveals Transcriptional Links to Childhood Crohn's Disease. Dev Cell, 2020. 55(6): p. 771783 e5.

17. Wilm, B., et al., The serosal mesothelium is a major source of smooth muscle cells of the gut vasculature. Development, 2005. 132(23): p. 5317-28.